# Spectral Gradient Descent Mitigates Anisotropy-Driven Misalignment: A Case Study in Phase Retrieval

Guillaume Braun [1]   Han Bao [2 3 1]   Wei Huang [1 2]   Masaaki Imaizumi [4 1 5]

## Abstract

Spectral gradient methods, such as the Muon optimizer, modify gradient updates by preserving directional information while discarding scale, and have shown strong empirical performance in deep learning. We investigate the mechanisms underlying these gains through a dynamical analysis of a nonlinear phase retrieval model with anisotropic Gaussian inputs, equivalent to training a two-layer neural network with quadratic activation and fixed second-layer weights. Focusing on a spiked covariance setting where the dominant variance direction is orthogonal to the signal, we show that gradient descent (GD) suffers from a variance-induced misalignment: during the early escaping stage, the high-variance but uninformative spike direction is multiplicatively amplified, degrading alignment with the true signal under strong anisotropy. In contrast, spectral gradient descent (SpecGD) removes this spike amplification effect, leading to stable alignment and accelerated noise contraction. Numerical experiments confirm the theory and show that these phenomena persist under broader anisotropic covariances.

## 1. Introduction

Modern deep neural networks are predominantly trained by first-order optimization algorithms, most notably Stochastic Gradient Descent (Robbins & Monro, 1951) and its adaptive variants. These methods, including AdaGrad (Duchi et al., 2011), RMSProp (Tieleman & Hinton, 2012), Adam (Kingma & Ba, 2015), and their extensions, can be interpreted as Euclidean gradient descent equipped with preconditioning mechanisms based on past gradients or cur-

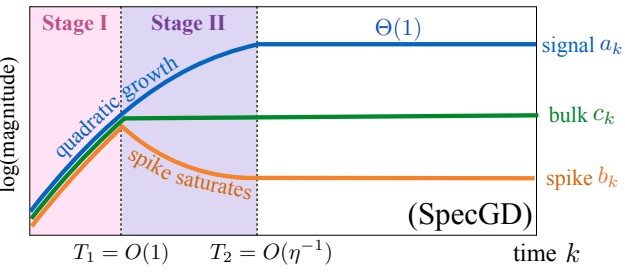

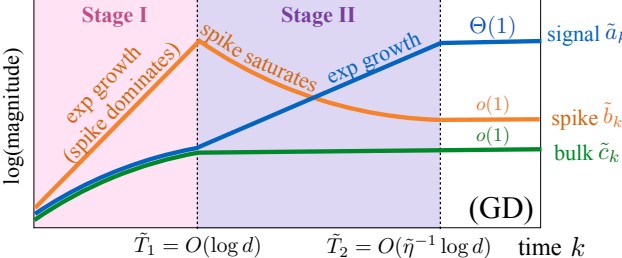

*Figure 1.* Qualitative summary of the learning dynamics of SpecGD (**top**) and GD (**bottom**). The parameter matrix is projected to the teacher *signal*, spurious *spike*, and isotropic *bulk* components, and the respective magnitudes (or coefficients) are denoted by $a_k$, $b_k$, and $c_k$, at time $k$, formally defined in Section 3. $\eta$ and $\tilde{\eta}$ are learning rates for SpecGD and GD, and $d$ is data dimension. In our theoretical analysis, we track how these three coefficients behave dynamically. **(i) Two stages:** all coefficients grow in Stage I and the signal coefficient approaches to $\Theta(1)$ (alignment) in Stage II; **(ii) Spike dominance:** spike $b_k$ dominates the others without the scale invariance of SpecGD; **(iii) Spike saturation:** after $b_k$ stops growing, the dynamics enter Stage II and $a_k$ begins to dominate (alignment); **(iv) Transition time:** SpecGD has no dimensional dependency in its transition time $T_1$ and hence exits both stages faster thanks to the synchronous growth in Stage I.

vature information. The simplest instances, such as Adam, rely on diagonal, coordinate-wise rescaling of the gradient, while more structured approaches such as K-FAC (Martens & Grosse, 2015) and Shampoo (Gupta et al., 2018) exploit matrix-valued second-moment information to construct Kronecker-factored preconditioners. Despite differences in implementation and per-iteration cost, they share a common design principle: the update remains linear in the gradient, and optimization proceeds along a Euclidean descent direction whose scale is adjusted by the preconditioner.

The recently introduced optimizer Muon (Jordan et al.,

[1]RIKEN AIP [2]The Institute of Statistical Mathematics [3]Tohoku University [4]The University of Tokyo [5]Kyoto University. Correspondence to: Guillaume Braun <guillaume.braun@riken.jp>.

*Proceedings of the 43rd International Conference on Machine Learning*, Seoul, South Korea. PMLR 306, 2026. Copyright 2026 by the author(s).

2024) breaks with the prevailing paradigm of first-order optimization in deep learning. Rather than treating model parameters as vectors and updating them by rescaling the Euclidean gradient, Muon explicitly exploits the matrix structure of layerwise parameters. In each layer, Muon replaces the raw gradient by an orthogonalized matrix direction obtained from its polar decomposition, preserving the left and right singular subspaces of the gradient matrix while discarding its singular values.

Remarkably, this conceptually simple update, which departs from classical gradient preconditioning, has been shown to match or outperform widely used adaptive optimizers such as Adam in large-scale neural network training, including Transformer architectures, while remaining computationally efficient and often reaching a given training loss in substantially fewer optimization steps (Liu et al., 2025). These empirical results suggest that treating parameters as matrices and modifying the geometry of the update, rather than merely rescaling gradients, can fundamentally alter learning dynamics.

Despite these striking empirical successes, the mechanisms underlying Muon's performance advantages remain only partially understood. Existing explanations can be broadly grouped into three categories.

First, Muon has been studied from an optimization perspective and admits deterministic or stochastic convergence guarantees (Lau et al., 2026; Shen et al., 2025; Chang et al., 2025; Sfyraki & Wang, 2025; Sato et al., 2025). While these results establish optimization guarantees, such as convergence to stationary points under appropriate learning rates, the statistical perspective is additionally necessary to explain the improved representation learning performance of Muon.

Second, several works geometrically characterize when spectral gradient updates can be beneficial. For instance, Fan et al. (2026) show that Muon converges to a max-margin solution in the spectral norm in linear classification, whereas GD favors the $\ell_2$-norm max-margin solution; similarly, Chen et al. (2026) show that Muon converges to a minimum spectral-norm solution characterized through KKT conditions. Related analyses identify conditions under which gradient orthogonalization is advantageous, such as low stable-rank features or rapidly growing curvature (Su, 2025; Davis & Drusvyatskiy, 2025). These approaches typically require intricate per-iteration conditions whose implications for long-term behavior remain unclear.

Finally, the end-to-end training dynamics of SpecGD have gained attention. Vasudeva et al. (2026) study SpecGD in a linear classification model under class imbalance and show that it learns all principal components of the data at the same rate, whereas GD prioritizes dominant components. A closely related insight appears in the Transformer setting, where Wang et al. (2025b) combine empirical evidence and a linear associative-memory analysis to show that Muon yields more balanced updates under heavy-tailed data. Closest to our setting, Wang et al. (2025a) analyze Muon in an isotropic linear regression setting and show that, in the large-batch regime, Muon effectively behaves as a linear method, suggesting that anisotropy or nonlinear feature learning are necessary to theoretically describe the advantages of Muon. This result motivates our study to focus on nonlinear regression under anisotropic inputs.

To see a fundamental difference between SpecGD and GD, we consider an anisotropic and nonlinear setting and show that SpecGD induces qualitatively different training dynamics from GD. Unlike Ma et al. (2026), which study a simplified Muon dynamics with decoupled spectral components and condition-number-free convergence, our analysis reveals a different mechanism in phase retrieval. SpecGD suppresses anisotropy-driven misalignment by controlling variance amplification in a coupled dynamics that evolves through successive stages.

## 1.1. Contributions

We study the training dynamics of SpecGD in an anisotropic phase retrieval setting and compare it with GD. Our goal is to understand the mechanism by which SpecGD interacts with anisotropic data and how this affects the global learning trajectory. Below, we summarize our findings.

- **SpecGD as a sign-based update in an adaptive basis.** We show that the learning dynamics of both SpecGD and GD admit an exact reduction to a three-dimensional invariant manifold capturing the evolution of the *signal*, spurious *spike*, and isotropic *bulk* components. On this reduced manifold, SpecGD performs a sign-based (scale-invariant) gradient update, not in the ambient Euclidean basis. This adaptive normalization fundamentally alters how the algorithm interacts with anisotropy, preventing amplification of uninformative directions that dominate GD dynamics. Compare the spike coefficient $b_k$ in Figure 1 to see the difference in their responses to anisotropy.

- **Stage-wise learning dynamics.** We show that the learning dynamics exhibit a two-stage structure. During the *growth stage* (Stage I) of SpecGD, all of the {signal, spike, bulk} coefficients grow simultaneously. Then, the *alignment stage* (Stage II) follows, where the signal coefficient continues to grow and align with the signal direction eventually, while the {spike, bulk} coefficients saturate.

  In the growth stage, SpecGD amplifies all {signal, spike, bulk} coefficients at comparable rates; GD heterogeneously amplifies so that the spike coefficient quickly dominates the {signal, bulk} coefficients, driven by

the anisotropic data covariance. Compare the Stage I $(a_k, b_k, c_k)$ curves in Figure 1.

This difference forces GD to use a smaller learning rate for stability and delays the onset of the alignment stage. In contrast, SpecGD avoids spike dominance, leading to earlier and more robust alignment. Figure 1 compare the transition time $T_1$ of Stage I.

- **Empirical validation beyond the stylized model.** Through numerical experiments, we demonstrate that SpecGD continues to outperform GD under more general covariance structures, such as power-law spectra, and in finite-sample regimes. These results suggest that the proposed mechanism captures a robust feature of spectral gradient methods rather than an artifact of the stylized spiked covariance model.

## 1.2. Other Related Work

**Non-convex optimization and feature learning.** A broad line of work studies gradient-based training dynamics in neural networks, with the goal of characterizing feature learning, convergence, and computational–statistical trade-offs in nonconvex models such as the multi-index model (Saad & Solla, 1995a;b; Goldt et al., 2019; Veiga et al., 2022; Ben Arous et al., 2022; Abbe et al., 2022; 2023; Bietti et al., 2025; Arnaboldi et al., 2023b; Dandi et al., 2024; Arnaboldi et al., 2024; Collins-Woodfin et al., 2024; Bruna & Hsu, 2025). These analyses provide a detailed understanding of how gradient-based methods transition from lazy or kernel-like regimes to genuine feature learning, and how this transition depends on model architecture, initialization, and complexity. However, most existing analyses rely on isotropic or weakly structured inputs, which allow the dynamics to collapse onto a finite set of order parameters.

**Learning from anisotropic data.** Beyond the linear setting, where anisotropy and power-law spectra have been extensively studied (Maloney et al., 2022; Bahri et al., 2024; Atanasov et al., 2026; Bordelon et al., 2024; Worschech & Rosenow, 2025; Paquette et al., 2024; Lin et al., 2024), the impact of anisotropic inputs on nonlinear regression and feature learning is less well understood. Existing works have primarily focused on restricted settings: Ba et al. (2023) analyze single-index models under a spiked covariance structure; Braun et al. (2025) consider learning under a more general but mild form of anisotropy. More recently, Wortsman & Loureiro (2025) study kernel ridge regression with power-law covariates, showing that in certain regimes the spectral decay of the inputs is transferred to the learned features. Closest to our work, Braun et al. (2026) characterize the population dynamics of phase retrieval under power-law data, revealing that—unlike in the isotropic case—the learning dynamics are governed by an infinite hierarchy

of coupled ordinary differential equations. Their analysis, however, is restricted to gradient flow and a single-neuron estimator.

**Quadratic models.** Quadratic observation models, with phase retrieval as a canonical example, are standard testbeds for studying nonconvex learning dynamics (Candès et al., 2015; Tan & Vershynin, 2019; Davis et al., 2020; Ma et al., 2021; Dong et al., 2023; Tan & Vershynin, 2023). From a learning perspective, they correspond to shallow neural networks with quadratic activations and represent the simplest nonlinear extension of linear prediction. Recent work has analyzed their training dynamics under gradient-based methods (Sarao Mannelli et al., 2020; Arnaboldi et al., 2023a; Martin et al., 2024; Erba et al., 2026; Arous et al., 2026; Defilippis et al., 2026; Braun et al., 2026). Our contribution builds on this line of work by studying a multi-neuron quadratic model with spiked input covariance and matrix-aware spectral updates, leading to qualitatively different and faster dynamics.

## 1.3. Notations

We write $a_n \lesssim b_n$ (resp. $a_n \gtrsim b_n$) if there exists a constant $C > 0$ such that $a_n \leq Cb_n$ (resp. $a_n \geq Cb_n$) for all $n$. We write $a_n = \Theta(b_n)$ if there exist constants $0 < c \leq C < \infty$ such that $c\, b_n \leq a_n \leq C\, b_n$ for all $n$. Standard asymptotic notations $O(\cdot)$ and $\Omega(\cdot)$ are used when the inequalities hold for sufficiently large $n$. For vectors $x, y \in \mathbb{R}^d$, we denote by $\langle x, y \rangle$ and $\|x\|$ the Euclidean inner product and norm, respectively. For matrices $A, B \in \mathbb{R}^{d \times d}$, we use the Frobenius inner product $\langle A, B \rangle_F := \mathrm{Tr}(A^\top B)$, with associated Frobenius norm $\|A\|_F$. The operator norm of a matrix $A$ is denoted by $\|A\|$. The $d \times d$ identity matrix is denoted by $I_d$, and the unit sphere in $\mathbb{R}^d$ by $\mathbb{S}^{d-1}$.

## 2. Statistical Framework

**Data generation.** We consider a phase retrieval model of the form
$$y = (x^\top w_\star)^2 + \nu,$$
where $w_\star \in \mathbb{S}^{d-1}$ is the unknown target signal. An input $x \sim \mathcal{N}(0, Q)$ is drawn from a Gaussian distribution with the covariance matrix $Q \in \mathbb{R}^{d \times d}$, and $\nu \sim \mathcal{N}(0, \sigma^2)$ is an additive noise term independent of $x$.

Throughout the theoretical analysis, we focus on a *spiked covariance* model of the form
$$Q = I_d + \lambda vv^\top, \qquad \lambda > 0, \ v \in \mathbb{S}^{d-1},$$
where the spike direction $v$ is assumed to be orthogonal to the signal, i.e., $v^\top w_\star = 0$.

*Remark* 2.1. Our goal is to study the effect of an uninformative covariance direction on the learning dynamics. Unlike

prior work, e.g., Ba et al. (2023), that exploits alignment between the spike and the signal to accelerate learning, we consider a more challenging regime in which the spike is orthogonal to the signal.

When the spike intensity $\lambda$ is large, most samples concentrate along an uninformative covariance direction orthogonal to the signal, while informative fluctuations along the signal direction become comparatively rare. As a result, gradient updates are dominated by variance-driven components, making signal recovery more challenging. We make the following assumption on $\lambda$.

**Assumption A.** We assume that $\log d \lesssim \lambda \lesssim d$.

**Model.** We consider a one-hidden-layer neural network with the squared activation function. Given an input $x \in \mathbb{R}^d$, the model output is

$$f_W(x) = \sum_{j=1}^{m} (w_j^\top x)^2,$$

where $W = (w_1, \ldots, w_m) \in \mathbb{R}^{d \times m}$ denotes the hidden-layer weights.

This model admits an equivalent matrix formulation. By defining the positive semidefinite matrix $M := WW^\top \in \mathbb{R}^{d \times d}$, the network output can be written as

$$f_W(x) = x^\top M x.$$

Thus, training a one-layer neural network with the squared activations is equivalent to learning a positive semidefinite matrix $M$ of rank at most $m$.

**Loss.** We consider the population squared loss

$$\mathcal{L}(W) = \mathbb{E}\big(y - f_W(x)\big)^2 = \mathbb{E}\big(x^\top M x - x^\top M_\star x\big)^2 + \sigma^2,$$

where $M = WW^\top$ and $M_\star := w_\star w_\star^\top$. Note that the loss depends on $W$ only through the matrix $M$.

**Signal alignment.** We quantify alignment with the signal by using the Frobenius cosine between $M$ and $M_\star$,

$$\text{Align}(M) := \frac{\langle M, M_\star \rangle}{\|M\|_F \|M_\star\|_F} = \frac{w_\star^\top M w_\star}{\|M\|_F}, \qquad (1)$$

which takes values in $[0, 1]$.

**Initialization.** We assume $m = d$ together with the on-manifold initialization by sampling $W(0)^\top \in \mathbb{R}^{m \times d}$ uniformly from the Stiefel manifold $\text{St}(m, d)$, so that $W(0)W(0)^\top = I_d$. We then rescale the initialization by a factor $\theta > 0$, which yields $M(0) = \theta^2 I_d$.

**Assumption B.** We set the initialization scale as

$$\theta^2 = \rho_0 d^{-1} \quad \text{with } \rho_0 = \log^{-1} d.$$

This small, isotropic initialization places the dynamics in the feature-learning regime. More generally, any choice $\rho_0 = o(1)$ would lead to the same qualitative behavior; the specific scaling is fixed for clarity.

*Remark* 2.2. The Stiefel initialization is used for analytical convenience. In practice, a qualitatively similar behavior is observed with a small Gaussian initialization $w_j \overset{\text{i.i.d.}}{\sim} \mathcal{N}(0, \theta^2 I_d)$ with $\theta^2 = O(\rho_0(md)^{-1})$ in our experiments in Section 6.

**Optimization.** We study the training dynamics induced by gradient-based methods at the population level.

Spectral gradient descent (SpecGD) updates the gradient by its polar factor, yielding the population update at time $k = 0, 1, 2, \ldots$ as

$$W_{k+1} = W_k - \eta \,\text{polar}\big(\nabla_W \mathcal{L}(W_k)\big), \qquad (2)$$

where $\eta > 0$ denotes a learning rate and $\text{polar}(A) := A(A^\top A)^{-1/2}$ the polar factor of $A \in \mathbb{R}^{d \times d}$, with $(\cdot)^{-1/2}$ understood in the Moore–Penrose sense. By construction, SpecGD preserves the principal directions of the gradient while discarding its singular values, suppressing anisotropic scaling effects.

*Remark* 2.3. SpecGD serves as an idealized proxy for Muon, in which momentum is removed and the Newton–Schulz approximation is replaced by the exact polar decomposition. This type of idealization is common in theoretical studies of normalization-based optimizers, as it isolates the essential geometric bias while enabling precise analysis. The computational cost of computing the polar decomposition is $O(m^2 d)$ (if $m \le d$).

We compare the above result with the gradient descent (GD), which updates the parameter matrix $\tilde{W}_k$ for $k = 0, 1, 2, \ldots$ as

$$\tilde{W}_{k+1} = \tilde{W}_k - \tilde{\eta} \nabla_W \mathcal{L}(\tilde{W}_k), \qquad (3)$$

with a learning rate $\tilde{\eta} > 0$.

## 3. Reduced ODE Dynamics on Manifold

We show that the dynamics reduce to a three-dimensional manifold (Section 3.1). We then introduce a common stage decomposition (Section 3.2). Proofs are deferred to Section A.

### 3.1. Three-dimensional Manifold

We can reduce the dynamics to a *three-dimensional invariant manifold*, by exploiting the symmetries of the spiked covariance model. In particular, the ambient space is naturally decomposed into three components: (i) the signal direction $w_\star$, (ii) the spike direction $v$, and (iii) the subspace

orthogonal to both. To this end, we define the manifold

$$\mathcal{M} := \left\{ a\, w_\star w_\star^\top + b\, vv^\top + c\, P_\perp \; : \; a, b, c \in \mathbb{R}^+ \right\},$$

where $P_\perp := I_d - w_\star w_\star^\top - vv^\top$ denotes the projection onto $\text{span}\{w_\star, v\}^\perp$. We refer to the noise directions $\{P_\perp w_i : i = 1, \ldots, m\}$ as the *bulk*.

Owing to the covariance symmetry, the dynamics can be restricted to the invariant manifold $\mathcal{M}$: if the initial alignment with all bulk directions is identical, this property is preserved by the gradient dynamics. With $M_k := W_k W_k^\top$ and $\tilde{M}_k := \tilde{W}_k \tilde{W}_k^\top$, the following lemma makes this precise.

**Lemma 3.1.** *If $M_0 \in \mathcal{M}$, then $\mathcal{M}$ is invariant under SpecGD and GD: $M_k, \tilde{M}_k \in \mathcal{M}$ for all $k \geq 0$.*

This property follows from the invariance of the population loss under orthogonal transformations that fix the signal direction $w_\star$ and the spike direction $v$.

For any trajectory $\{M_k\}_{k \in \mathbb{N}}$ by SpecGD evolving in $\mathcal{M}$, there exist unique coefficient scalars $a_k, b_k$, and $c_k$ such that

$$M_k = a_k\, w_\star w_\star^\top + b_k\, vv^\top + c_k\, P_\perp. \tag{4}$$

Similarly, for $\{\tilde{M}_k\}_{k \in \mathbb{N}}$ yielded by GD, we define coefficient scalars $\tilde{a}_k, \tilde{b}_k$ and $\tilde{c}_k$ as satisfying

$$\tilde{M}_k = \tilde{a}_k\, w_\star w_\star^\top + \tilde{b}_k\, vv^\top + \tilde{c}_k\, P_\perp.$$

### 3.2. Two Stages of Training Dynamics

Motivated by the reduced dynamics shown in Figure 1 (see also Figure 3 for numerical simulations), we introduce two training stages and their associated transition times.

**Stage I (growth):** For SpecGD, we define $T_1$ as the first time at which the spike coefficient stops growing,

$$T_1 := \inf\{k \geq 0 : b_{k+1} \leq b_k\}.$$

This time marks the end of the initial growth regime in which all coefficients increase, after which the spike saturates while the signal coefficient continues to grow. For the GD case, we similarly define $\tilde{T}_1 := \inf\{k \geq 0 : \tilde{b}_{k+1} \leq \tilde{b}_k\}$.

**Stage II (spike saturation and alignment):** For SpecGD, we further define $T_2$ as the time at which the signal coefficient reaches a fixed constant level $\delta > 0$ small enough,

$$T_2 := \inf\{k \geq T_1 : a_k \geq \delta\}.$$

Beyond $T_2$, the signal dominates the nuisance coefficients $(b_k, c_k)$, leading to the alignment close to one. We similarly define $\tilde{T}_2 := \inf\{k \geq \tilde{T}_1 : \tilde{a}_k \geq \delta\}$ for GD.

## 4. Main Results

We now present our main theoretical results. We first characterize the population dynamics induced by SpecGD (Section 4.1), and then compare them with the corresponding dynamics of GD (Section 4.2), highlighting the qualitative and quantitative advantages of SpecGD.

### 4.1. Dynamics of SpecGD

We now summarize the discrete-time population dynamics of SpecGD on the invariant manifold $\mathcal{M}$. The resulting behavior follows the stage structure previously introduced and is illustrated by Figure 1.

**Theorem 4.1** (Stage I: uniform growth). *Consider the discrete-time population SpecGD dynamics on $\mathcal{M}$ with the on-manifold initialization described in Section 2. Choose $\eta = \kappa/\sqrt{d + \lambda}$ for a sufficiently small absolute constant $\kappa > 0$. Then, for $d$ large enough, the following holds:*

*(i) the stage time length satisfies $T_1 = O(1)$,*
*(ii) the coefficients scale as $a_k = (\sqrt{a_0} + k\eta)^2, b_k = (\sqrt{b_0} + k\eta)^2$, and $c_k = (\sqrt{c_0} + k\eta)^2$ for all $k \leq T_1$.*

From this, we see that the initial growth stage ends in constant time independent of dimension $d$. Furthermore, all coefficients grow quadratically with the number of updates at identical rates, in contrast with GD. This behavior stems from the sign-based update induced by the polar decomposition; see Section 5 and Proposition 5.3.

We next summarize Stage II describing the alignment phase.

**Theorem 4.2** (Stage II: alignment). *Assume the same set of the assumptions as in Theorem 4.1. Then, for $d$ large enough, the following holds:*

*(i) the stage time length satisfies $T_2 = O(\eta^{-1})$,*
*(ii) the coefficients scale as $a_k = (\sqrt{a_0} + k\eta)^2, b_k = O((1 + \lambda)^{-1})$, and $c_k = O(d^{-1})$ for all $k \geq T_1$.*

*Furthermore, for all $k \geq T_2$, we have*

$$a_k = \Theta(1) \quad \text{and} \quad \text{Align}(k) = 1 - o(1). \tag{5}$$

This result indicates that in Stage II, only $a_k$ continues to increase until it reaches constant order at time $T_2$, while the increase in the nuisance coefficients $(b_k, c_k)$ are uniformly bounded at their natural scales. As a result, after Stage II, the alignment converges to one.

### 4.2. Comparison with GD

We characterize the population GD dynamics on $\mathcal{M}$ through two successive training stages.

**Theorem 4.3.** *Consider the GD dynamics on $\mathcal{M}$. Choose a learning rate $\tilde{\eta} \leq 1/(16(1 + \lambda))$. Then, for $d$ large enough, the following holds:*

- *Stage I (growth)*
    - *(i) the time length satisfies $\tilde{T}_1 = O((\tilde{\eta}\lambda)^{-1} \log d)$,*
    - *(ii) the spike coefficient $\tilde{b}_k$ evolves as $\tilde{b}_{k+1} \gtrsim (1 + \Omega(\tilde{\eta}))^k \tilde{b}_0$ and*

    $$\max\left\{\frac{\tilde{a}_k}{\tilde{b}_k}, \frac{\tilde{c}_k}{\tilde{b}_k}\right\} = O((1 - \delta)^k),$$

    *with some $\delta \in (0, 1)$.*
- *Stage II (noise saturation and alignment).*
    - *(i) the time length satisfies $\tilde{T}_2 = O(\tilde{\eta}^{-1} \log d)$,*
    - *(ii) the signal coefficient $\tilde{a}_k$ evolves as $\tilde{a}_{k+1} \gtrsim (1 + \Omega(\tilde{\eta}))^k \tilde{a}_0$, while the other coefficients satisfy $\max\{\tilde{b}_k, \tilde{c}_k\} = O(1)$.*

*Furthermore, for all $k \geq \tilde{T}_2$, we have $\tilde{a}_k = \Theta(1)$ and $\max\{\tilde{b}_k, \tilde{c}_k\} = o(1)$.*

Based on the estimates for $\tilde{T}_1$ and $\tilde{T}_2$, GD requires more time than SpecGD to progress through both Stage I and Stage II, with this gap becoming more pronounced as the dimension $d$ increases. This slowdown stems from the stage-wise dynamics of $(\tilde{a}_k, \tilde{b}_k, \tilde{c}_k)$. During Stage I, GD is dominated by spike-driven growth of $\tilde{b}_k$, which increases geometrically faster than the other coefficients (see Proposition D.1). As a result, the signal alignment can deteriorate in this phase, as quantified in Corollary D.2 in the appendix. In Stage II, the signal component $\tilde{a}_k$ grows only after a longer delay, leading to a delayed improvement in alignment. Overall, both stages last longer under GD than under SpecGD.

## 5. Proof Outline

In this section, we outline the proofs of the results presented in Section 4. For clarity, we focus on the *continuous-time* population dynamics, where the underlying mechanisms are more transparent. We begin by introducing the corresponding continuous-time gradient flows (GFs):

$$\dot{W}(t) = -\text{polar}\big(\nabla_W \mathcal{L}(W(t))\big), \qquad \text{(SpecGF)}$$
$$\dot{\tilde{W}}(t) = -\nabla_W \mathcal{L}(\tilde{W}(t)). \qquad \text{(GF)}$$

When $M(t) \in \mathcal{M}$ we write

$$M(t) := W(t)W(t)^\top = a(t)\,w_\star w_\star^\top + b(t)\,vv^\top + c(t)\,P_\perp.$$

The same decomposition applies to GF, with $\tilde{M}(t) := \tilde{W}(t)\tilde{W}(t)^\top$ and coefficients $\tilde{a}(t)$, $\tilde{b}(t)$, and $\tilde{c}(t)$.

### 5.1. Reduced ODE on $\mathcal{M}$

Lemma 3.1 is also valid in continuous time (see the proof in Section A), hence the population dynamics generated by either GF or SpecGF remain confined to $\mathcal{M}$ for all $t \geq 0$ whenever $M(0) \in \mathcal{M}$. Consequently, the evolution of $M(t)$ is fully characterized by the three scalar coefficients $a(t)$, $b(t)$, and $c(t)$. We now make this reduction explicit for GF by deriving closed systems of ordinary differential equations governing the dynamics of $(\tilde{a}(t), \tilde{b}(t), \tilde{c}(t))$. The detailed proofs are deferred to Section A.

**Proposition 5.1.** *The dynamics of GF is described by the following ODEs:*

$$\dot{\tilde{a}}(t) = -2g_{w_\star}(t)\tilde{a}(t), \qquad (6)$$
$$\dot{\tilde{b}}(t) = -2g_v(t)\tilde{b}(t), \qquad (7)$$
$$\dot{\tilde{c}}(t) = -2g_\perp(t)\tilde{c}(t), \qquad (8)$$

*where*

$$g_{w_\star}(t) := 8(\tilde{a}(t) - 1) + 4(\tilde{r}(t) - 1),$$
$$g_v(t) := 8(1 + \lambda)^2\tilde{b}(t) + 4(1 + \lambda)(\tilde{r}(t) - 1), \quad (9)$$
$$g_\perp(t) := 8\tilde{c}(t) + 4(\tilde{r}(t) - 1),$$

*and $\tilde{r}(t) := \tilde{a}(t) + (1 + \lambda)\tilde{b}(t) + (d - 2)\tilde{c}(t)$.*

*Remark* 5.2. The quantity $\tilde{r}$ admits the form $\text{Tr}(\tilde{M}Q)$, which represents the "mass" of the network and directly controls the coupling between the three coefficients. In the small random initialization regime, $\tilde{r}(0) = o(1)$ by choice of the scaling $\theta$. This ensures that the ODE system is decoupled during early training.

We next derive the corresponding reduction for SpecGF. In contrast to GF, the resulting dynamics are scale-invariant and depend only on the signs of the pre-gradients $g_{w_\star}(t)$, $g_v(t)$, and $g_\perp(t)$.

**Proposition 5.3.** *The dynamics of SpecGF is described by the following ODEs:*

$$\dot{a}(t) = -2\,\text{Sign}(g_{w_\star}(t))\,\sqrt{a(t)},$$
$$\dot{b}(t) = -2\,\text{Sign}(g_v(t))\,\sqrt{b(t)}, \qquad (10)$$
$$\dot{c}(t) = -2\,\text{Sign}(g_\perp(t))\,\sqrt{c(t)},$$

*with the convention $\text{Sign}(0) := 0$. Equivalently, in square-root variables $\alpha(t) := \sqrt{a(t)}$, $\beta(t) := \sqrt{b(t)}$, and $\gamma(t) := \sqrt{c(t)}$,*

$$\dot{\alpha}(t) = -\text{Sign}(g_{w_\star}(t)), \qquad (11)$$
$$\dot{\beta}(t) = -\text{Sign}(g_v(t)), \qquad (12)$$
$$\dot{\gamma}(t) = -\text{Sign}(g_\perp(t)). \qquad (13)$$

*Remark* 5.4. The reduced dynamics of SpecGF differ qualitatively from those of GF. While GF induces multiplicative,

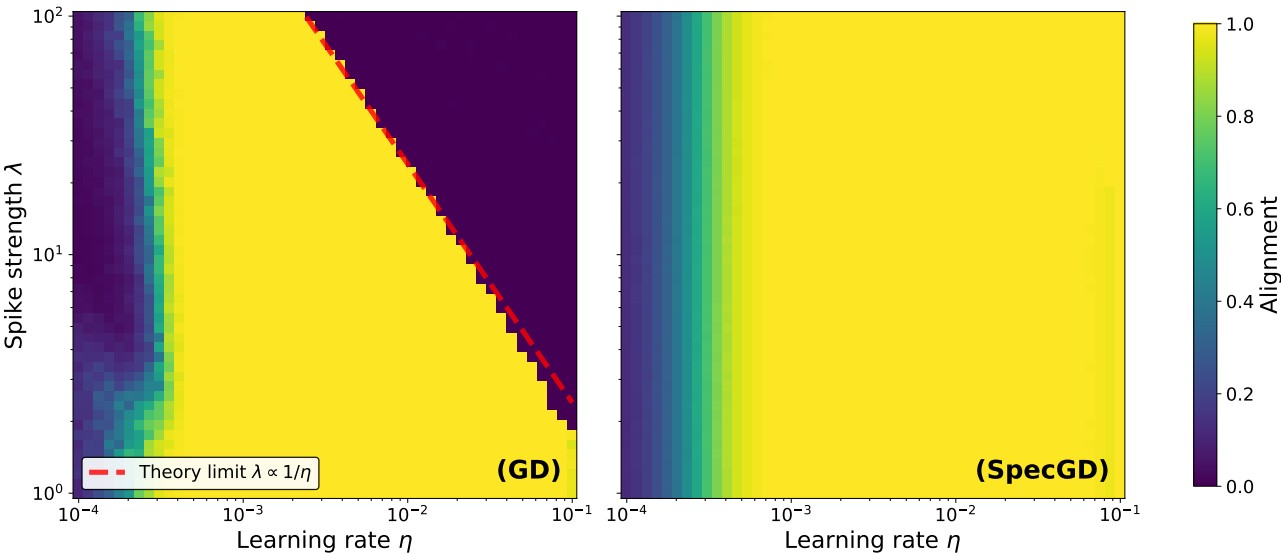

*Figure 2.* Final Frobenius alignment for GD (**left**) and SpecGD (**right**). Parameters: $d = 300$, $m = 100$, $T = 1000$, and $\rho_0 = 10^{-2}$.

scale-dependent updates driven by data anisotropy, SpecGF depends only on the sign of the reduced coefficients, yielding scale-invariant dynamics on $\mathcal{M}$. This mechanism should not be confused with signed gradient descent (often used as a proxy for Adam), where the sign operator is applied in the canonical Euclidean basis. In contrast, SpecGF applies the sign operation in a basis that adapts to the data geometry and the current iterate.

We now describe the main proof ideas. The detailed proofs for GF (resp. SpecGF) are deferred to Section B (resp. Section C). The discrete-time cases are treated in Sections D and E, respectively.

### 5.2. Stage I: escape from the small random initialization regime

For GF, our initialization $\tilde{W}(0)/\theta \in \text{St}(d, d)$ yields

$$\tilde{a}(0) = \tilde{b}(0) = \tilde{c}(0) = \theta^2 \text{ and } \tilde{r}(0) = (d + \lambda)\theta^2 = O(\rho_0).$$

Consequently, as long as these quantities remain small, the pre-gradient in (9) are approximately constant

$$g_{w_\star}(t) \approx -12,\ g_v(t) \approx -4(1 + \lambda),\ \text{ and } g_\perp(t) \approx -4.$$

Plugging these into the reduced GF system (6) and the SpecGF system (10) yields the exponential growth in Stage I described in Theorem 4.3 and the quadratic growth described in Theorem 4.1, respectively. This is illustrated by Figure 3 in the appendix, and see also Figure 1. Since SpecGF only depends on the signs of the (rescaled) spike coefficient $g_v$, the influence of $\lambda$ is mild, whereas GF is directly affected by $\lambda$ to amplify the spike coefficient $\tilde{b}(t)$ exponentially.

For GF, however, this approximation, obtained by freezing the coefficients $g_{w_\star}(t)$, $g_v(t)$, and $g_\perp(t)$ at their initial values, is not sufficient to describe the entire escape stage. Due to the much faster growth of the spike coefficient, the coupling through $\tilde{r}(t)$ becomes relevant before time $T_1$. As a consequence, the spike no longer evolves independently of the bulk, and its growth eventually saturates at a turning point. Capturing this behavior requires a finer decomposition of Stage I into two sub-stages, analyzed in Sections B.1 and B.2 of the appendix.

### 5.3. Stage II: noise saturation and signal growth

We focus on the analysis of GF, and the analysis of SpecGF following from similar arguments. Recall that $g_v(t) = 4(1 + \lambda)(\tilde{r}(t) + 2(1 + \lambda)\tilde{b}(t) - 1)$, so the sign of $g_v$ is determined by the network mass $\tilde{r}(t) + 2(1 + \lambda)\tilde{b}(t) - 1$. By definition of $\tilde{T}_1$ we have $g_v < 0$ for all $t \leq \tilde{T}_1$. Define $\tilde{C}(t) := (d - 2)\tilde{c}(t)$ and $\tilde{B}(t) := (1 + \lambda)\tilde{b}(t)$. By using a barrier argument that prevents the spike from growing indefinitely, we can show that these quantities remain bounded.

**Lemma 5.5.** *For all $t \geq 0$, one has*

$$0 \leq \tilde{a}(t) \leq 1,\ 0 \leq \tilde{B}(t) \leq \frac{1}{3},\ 0 \leq \tilde{C}(t) \leq 1.$$

We have $\tilde{B}(\tilde{T}_1) \approx 1/3 \approx \tilde{r}(\tilde{T}_1)$ since $\tilde{B}$ dominates $\tilde{a}$ and $\tilde{C}$ during Stage I and $g_v(\tilde{T}_1) = 0$ by definition. Then the spike can no longer grow further. On the other hand, the signal coefficient satisfies

$$\dot{\tilde{a}}(t) = \tilde{a}(t)(24 - 16\tilde{a}(t) - 8\tilde{r}(t)) \geq \kappa\tilde{a}(t),$$

for some positive constant $\kappa > 0$ since $\tilde{r}(t) \leq \frac{7}{3}$ by Lemma 5.5 and $\tilde{a}(t) \leq \delta$ during Stage II for $\delta > 0$ small

enough. As a consequence, the signal coefficient continues to grow exponentially, but at a slower rate than Stage I. Similarly, the bulk coefficient satisfies

$$\dot{\tilde{c}}(t) = 8(1 - \tilde{r}(t))\tilde{c}(t) - 16\tilde{c}(t)^2.$$

Fix any small $\varepsilon > 0$. As long as $\tilde{r}(t) \leq 1 - \varepsilon$, we have $1 - \tilde{r}(t) \geq \varepsilon$ and hence

$$\dot{\tilde{c}}(t) \geq 8\varepsilon\,\tilde{c}(t) - 16\tilde{c}(t)^2.$$

Moreover, while $\tilde{r}(t) \leq 1-\varepsilon$ we also have $\tilde{c}(t) \leq \tilde{r}(t)/(d-2) \leq 1/(d-2)$, so for $d$ large enough the quadratic term is negligible and

$$\dot{\tilde{c}}(t) \geq 4\varepsilon\,\tilde{c}(t).$$

Thus the bulk mass $\tilde{C}(t)$ grows exponentially until $\tilde{r}(t)$ enters the band of $\tilde{r}(t) \simeq 1$. Once $\tilde{r}(t)$ becomes close to 1, the prefactor $(1 - \tilde{r}(t))$ vanishes and suppresses further bulk growth, leading to saturation. Since the growth of $\tilde{r}(t)$ is mainly driven by $\tilde{C}(t)$ during most of Stage II, we have $\tilde{r}(t) < 1 - \varepsilon$ during this stage. After that, $\tilde{a}(t)$ will drive the convergence of $\tilde{r}(t)$ toward 1.

### 5.4. Discretization

The discrete-time analysis is more delicate than in continuous time and cannot be obtained by a direct Euler discretization of the gradient flow. The main technical difficulties and their resolution are addressed in the appendix; see Section D for GD and Section E for SpecGD.

**Quadratic learning rate effects.** In discrete time, the population update induces the exact multiplicative recursion

$$\tilde{M}_{k+1} = (I - \tilde{\eta}G_k)\,\tilde{M}_k\,(I - \tilde{\eta}G_k),$$

where $G_k$ denotes the population gradient matrix evaluated at $\tilde{M}_k$. The formal definition and derivation are given in Appendix D. Even when restricted to the invariant manifold $\mathcal{M}$, this produces squared factors in the scalar dynamics, leading to a genuine $O(\tilde{\eta}^2)$ contribution. While negligible away from critical regimes, this term becomes relevant near turning points where the leading drift vanishes, and may affect monotonicity if left uncontrolled. The discrete analysis therefore relies on explicit one-step bounds and a sufficiently small learning rate to ensure that the qualitative phase structure of the continuous dynamics is preserved.

**Discrete barriers and non-smooth updates.** Continuous-time barrier arguments do not directly extend to discrete time, since a single update may overshoot a threshold. We therefore rely on discrete trapping arguments, showing that once an iterate enters a contracting region it cannot escape, and that any overshoot is uniformly controlled at scale $O(\tilde{\eta})$. In SpecGD, this issue is compounded by the non-smooth

sign-based update, which may cross sign-change thresholds and induce small oscillations. We show that such oscillations remain confined to an $O(\eta)$ neighborhood (Sections D and E).

## 6. Numerical Experiments

We present numerical experiments illustrating and extending our theoretical findings on learning-rate effects, alignment dynamics, and robustness to random initialization and power-law covariance matrices.

**Influence of the learning rate.** We set $d = 300$, $m = 100$, $\rho_0 = 0.01$ and use batches of size 5000. We compare the alignment obtained after $T = 1000$ iterations for GD and SpecGD across various choices of the learning rate $\eta$ and spike strength $\lambda$. We use small Gaussian random initialization with scaling $\theta^2 = \rho_0(md)^{-1}$. Figure 2 shows that when $\lambda$ is large, GD fails to align for moderately large learning rates $\eta$, whereas SpecGD remains robust. A possible explanation for this failure is that a large learning rate breaks the barrier mechanism that prevents the spike component from growing excessively. This behavior is further illustrated in Figures 6, 7, and 8 in the appendix. Figure 2 also suggests that SpecGD could tolerate a larger learning rate than $O((d + \lambda)^{-1/2})$.

**Population dynamics.** Figure 5 in the appendix complements Figure 3 and shows that under GD the alignment decreases during Phase I and improves only late in Phase II. Moreover, even after the alignment has converged to one, there remains a significant difference in the corresponding loss scale, suggesting that GD and SpecGD may induce different scaling laws.

**Minibatch training with random initialization and power-law covariance.** Using minibatch training with a small random Gaussian initialization, we consider a more general covariance matrix $Q$ with a power-law spectrum, where the signal direction $w_\star$ is aligned with the tail of $Q$.

Figure 4 reports the evolution of the empirical loss and the Frobenius alignment with the signal. In this strongly anisotropic regime, the loss remains highly erratic for both GD and SpecGD does not exhibit a clear monotone decrease over the time horizon considered. Nevertheless, the behavior of the representations is markedly different. Under GD, the alignment stays close to zero throughout training, indicating that the learned representation remains essentially orthogonal to the signal direction. In contrast, SpecGD displays a characteristic two-stage behavior: after an initial decrease of alignment during the early iterations, the algorithm progressively aligns with the signal, achieving a significantly higher alignment than GD. These results highlight that SpecGD is

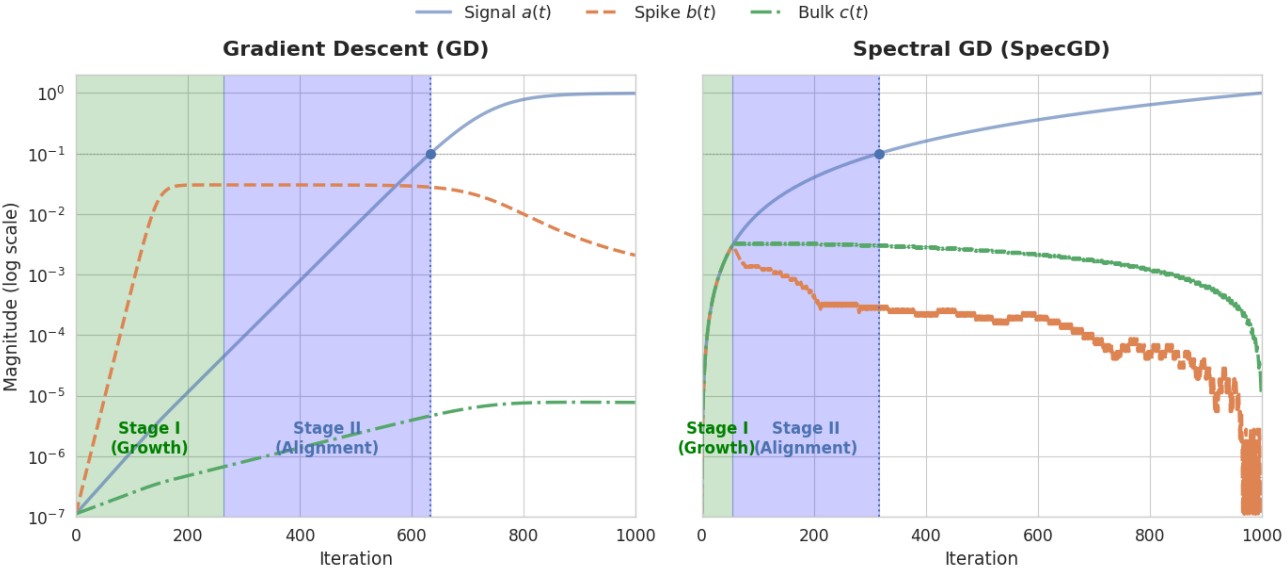

*Figure 3.* Log-scale plots of the dynamics ($d = m = 300$, $\eta = 10^{-3}$, $\rho_0 = 10^{-2}$, $\lambda = 10$). **Left:** GD. **Right:** SpecGD.

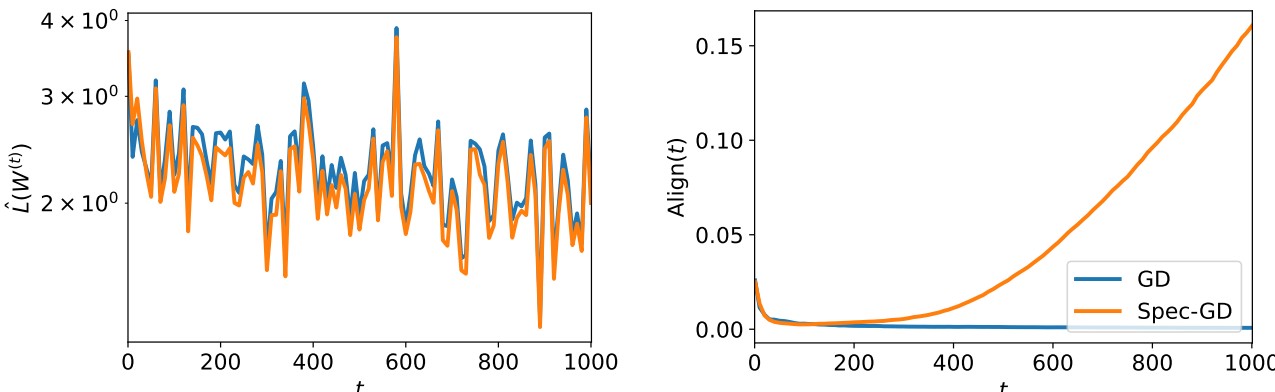

*Figure 4.* Training dynamics under a power-law covariance. **Left:** Empirical loss as a function of the iteration. **Right:** Frobenius alignment with the signal direction. While the loss remains highly irregular for both algorithms, SpecGD ultimately aligns with the signal, whereas GD remains nearly orthogonal.

still able to extract the relevant low-variance structure of the data, whereas standard GD fails to do so in this setting.

**MNIST dataset.** In Section F, we study the behavior of GD and SpecGD in a more realistic neural network setting. We train a two-layer ReLU network on a binary MNIST task (digits 4 vs. 9), where the inputs are perturbed by a rank-one Gaussian spike introducing a label-independent anisotropic direction in the data. As shown in Figure 10, SpecGD remains more stable than GD under this perturbation.

## 7. Discussion and Future Work

We have shown that, under anisotropic data, gradient descent can be misled toward an uninformative direction, resulting in delayed signal alignment and a need for small learning rates in strongly anisotropic regimes. In contrast, spectral gradient descent is largely insensitive to the magnitude of this spurious spike: its sign-based updates limit spike-driven dynamics and enable more direct signal alignment. Our analysis rigorously explains this separation through a detailed phase-wise decomposition of the training dynamics.

A natural first extension is to analyze the dynamics in the online noisy setting. Beyond this, extending the analysis to more general covariance structures, such as power-law spectra, and deriving the associated scaling laws is an interesting direction for future work. Finally, studying richer nonlinear models, including multi-index settings, is another promising direction.

## Acknowledgment

We would like to thank Bruno Loureiro for insightful discussions. HB is supported by JST PRESTO (JPMJPR24K6). WH is supported by JSPS KAKENHI (24K20848) and JST BOOST (JPMJBY24G6). MI is supported by JSPS KAKENHI (24K02904), JST CREST (JPMJCR21D2), JST FOREST (JPMJFR216I), and JST BOOST (JPMJBY24A9).

## Impact Statement

This paper presents work whose goal is to advance the theory of machine learning. There are many potential societal consequences of our work, none of which we feel must be specifically highlighted here.

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

The appendix is organized as follows. In Section A, we provide the detailed derivations and proofs underlying the reduced dynamical system presented in Section 3. Sections B and C analyze the continuous-time population dynamics of GD and SpecGD, respectively. Sections D and E are devoted to the corresponding discrete-time population dynamics. Finally, Section F contains additional experimental results. Throughout this appendix, we simplify notation by denoting the weight matrix by $W$ and the learning rate by $\eta$ for all algorithms, as the specific algorithm under consideration will be clear from context.

## A. Derivation of the Reduced Dynamical System on the Invariant Manifold

In this section, we provide the detailed derivation of the closed-form expression of the gradient and explain how it leads to the system of ODEs presented in Section 3.

### A.1. Closed-form expression of the loss and the gradient

We start by deriving a closed-form expression for the loss.

**Lemma A.1.** *Let us write $C = WW^\top - w_\star w_\star^\top$. The population loss can be rewritten as*

$$\mathcal{L}(W) = 2\,\mathrm{Tr}\big((QC)^2\big) + \big(\mathrm{Tr}(CQ)\big)^2.$$

*Proof.* Let $z \sim \mathcal{N}(0, I_d)$ and write $x = Q^{1/2}z$. Then

$$x^\top C x = z^\top \big(Q^{1/2}CQ^{1/2}\big)z = z^\top Bz,$$

where $B := Q^{1/2}CQ^{1/2}$.

**Step 1: Expansion.** We expand

$$z^\top Bz = \sum_{i,j} B_{ij}z_i z_j,$$

and therefore obtain

$$(z^\top Bz)^2 = \sum_{i,j,k,\ell} B_{ij}B_{k\ell}\,z_i z_j z_k z_\ell.$$

Taking expectations yields

$$\mathbb{E}(z^\top Bz)^2 = \sum_{i,j,k,\ell} B_{ij}B_{k\ell}\,\mathbb{E}[z_i z_j z_k z_\ell].$$

**Step 2: Wick's formula.** Since $z \sim \mathcal{N}(0, I_d)$, Wick's formula yields

$$\mathbb{E}[z_i z_j z_k z_\ell] = \delta_{ij}\delta_{k\ell} + \delta_{ik}\delta_{j\ell} + \delta_{i\ell}\delta_{jk},$$

where $\delta_{ab}$ denotes the Kronecker delta.

Substituting this identity into the sum gives

$$\sum_{i,j,k,\ell} B_{ij}B_{k\ell}\,\delta_{ij}\delta_{k\ell} = \sum_i B_{ii}\sum_k B_{kk} = (\mathrm{Tr}\,B)^2,$$

$$\sum_{i,j,k,\ell} B_{ij}B_{k\ell}\,\delta_{ik}\delta_{j\ell} = \sum_{i,j} B_{ij}^2 = \|B\|_F^2 = \mathrm{Tr}(B^2),$$

$$\sum_{i,j,k,\ell} B_{ij}B_{k\ell}\,\delta_{i\ell}\delta_{jk} = \sum_{i,j} B_{ij}B_{ji} = \mathrm{Tr}(B^2).$$

Since $B$ is symmetric, $\mathrm{Tr}(B^\top B) = \mathrm{Tr}(B^2)$, so the last two terms coincide. Collecting all terms, we obtain

$$\mathbb{E}(z^\top Bz)^2 = (\mathrm{Tr}\,B)^2 + 2\,\mathrm{Tr}(B^2).$$

**Step 3: Rewriting in terms of $Q$ and $C$.** By cyclicity of the trace,

$$\mathrm{Tr}(B) = \mathrm{Tr}(Q^{1/2}CQ^{1/2}) = \mathrm{Tr}(CQ),$$

and

$$\mathrm{Tr}(B^2) = \mathrm{Tr}(Q^{1/2}CQCQ^{1/2}) = \mathrm{Tr}(CQCQ) = \mathrm{Tr}\big((CQ)^2\big).$$

Therefore,

$$\mathbb{E}(x^\top Cx)^2 = 2\,\mathrm{Tr}\big((CQ)^2\big) + \big(\mathrm{Tr}(CQ)\big)^2.$$

$\square$

As a consequence of Lemma A.1 and the chain rule, we obtain an explicit characterization of the population gradient of $W$ and $M = WW^\top$.

We now compute the population gradient for GF.

**Lemma A.2.** *Let $M := WW^\top$, $C := M - w_\star w_\star^\top$, and*

$$G(M) := 8\,QCQ + 4\,\mathrm{Tr}(CQ)\,Q.$$

*Then, $\nabla_W \mathcal{L}(W(t)) = G(M(t))W(t)$, and under GF*

$$\dot{M}(t) = -G(M(t))M(t) - M(t)G(M(t)).$$

*Proof.* By Lemma A.1, the population loss can be written as

$$\mathcal{L}(W) = 2\,\mathrm{Tr}(CQCQ) + \big(\mathrm{Tr}(CQ)\big)^2.$$

We first compute the gradient with respect to $C$. We have

$$\mathrm{d}\,\mathrm{Tr}(CQCQ) = \mathrm{Tr}(\mathrm{d}C\,QCQ) + \mathrm{Tr}(CQ\,\mathrm{d}C\,Q) = 2\,\mathrm{Tr}(\mathrm{d}C\,QCQ),$$

where we used the cyclicity of the trace and the symmetry of $Q$ and $C$. Moreover,

$$\mathrm{d}\big(\mathrm{Tr}(CQ)\big)^2 = 2\,\mathrm{Tr}(CQ)\,\mathrm{d}\,\mathrm{Tr}(CQ) = 2\,\mathrm{Tr}(CQ)\,\mathrm{Tr}(\mathrm{d}C\,Q).$$

Combining these identities yields

$$\mathrm{d}\mathcal{L} = 4\,\mathrm{Tr}(\mathrm{d}C\,QCQ) + 2\,\mathrm{Tr}(CQ)\,\mathrm{Tr}(\mathrm{d}C\,Q) = \big\langle 4\,QCQ + 2\,\mathrm{Tr}(CQ)\,Q,\ \mathrm{d}C\big\rangle.$$

This shows that

$$\nabla_C \mathcal{L} = 4\,QCQ + 2\,\mathrm{Tr}(CQ)\,Q.$$

Next, since $C = WW^\top - w_\star w_\star^\top$, its differential satisfies

$$\mathrm{d}C = \mathrm{d}W\,W^\top + W\,\mathrm{d}W^\top.$$

Applying the chain rule, we obtain

$$\mathrm{d}\mathcal{L} = \langle \nabla_C\mathcal{L},\ \mathrm{d}C\rangle = \langle \nabla_C\mathcal{L},\ \mathrm{d}W\,W^\top + W\,\mathrm{d}W^\top\rangle.$$

Using the symmetry of $\nabla_C\mathcal{L}$, this simplifies to

$$\mathrm{d}\mathcal{L} = 2\,\langle \nabla_C\mathcal{L}\,W,\ \mathrm{d}W\rangle,$$

which implies

$$\nabla_W \mathcal{L}(W) = 2\,\nabla_C\mathcal{L}\,W.$$

Substituting the expression of $\nabla_C\mathcal{L}$ yields the claimed formula for the population gradient. $\square$

### A.2. Three-dimensional reduction of the matrix-valued dynamics

The spiked covariance model induces a natural decomposition of the ambient space into three components: the signal direction $w_\star$, the spike direction $v$, and the subspace orthogonal to both, which we refer to as the *bulk*. At the population level, the loss is invariant under rotations that fix $w_\star$ and $v$. Therefore, if at initialization the alignment with all bulk directions are equal, this property will be kept through gradient updates (see Lemma 3.1 for a formal statement). This motivates the definition of the following manifold $\mathcal{M}$.

Let $P_\perp := I_d - w_\star w_\star^\top - vv^\top$ be the projection onto $\{w_\star, v\}^\perp$. We introduce the three-dimensional manifold

$$\mathcal{M} := \left\{ M \in \mathbb{R}^{d \times d}_{\mathrm{sym}} : M = a\,w_\star w_\star^\top + b\,vv^\top + c\,P_\perp \ \text{ for some } (a,b,c) \in \mathbb{R}^3 \right\}. \tag{14}$$

On $\mathcal{M}$, we use the shorthand

$$r(t) := \mathbb{E}(x^\top M(t)x) = \mathrm{Tr}(M(t)Q) = a(t) + (1+\lambda)b(t) + (d-2)c(t). \tag{15}$$

The quantity $r(t)$ represents the unnormalized alignment of the network $M$ at time $t$ with the covariance $Q$. In the feature learning regime, it is common to choose a small initialization scaling $\theta$ such that $r(0) = o(1)$ in order to neglect the network scaling effect on the dynamics and approximate the square loss by the correlation loss. Note that ideally $r(t)$ should converge to $\mathrm{Tr}(w_\star w_\star^\top Q) = 1$. To simplify the notation, we will often drop the dependence in $t$ of the functions $r(t), a(t), b(t)$ and $c(t)$ when it is clear from the context.

**Lemma 3.1.** *If $M_0 \in \mathcal{M}$, then $\mathcal{M}$ is invariant under SpecGD and GD: $M_k, \tilde{M}_k \in \mathcal{M}$ for all $k \geq 0$.*

*Proof.* We prove invariance for the discrete-time iterates.

**Step 1: algebraic structure of $\mathcal{M}$.** Let

$$P_\star := w_\star w_\star^\top, \qquad P_v := vv^\top, \qquad P_\perp := I - P_\star - P_v.$$

Since $w_\star \perp v$ and $\|w_\star\| = \|v\| = 1$, these are orthogonal projectors satisfying $P_i P_j = \mathbf{1}_{i=j} P_i$ and $P_\star + P_v + P_\perp = I$. By definition,

$$\mathcal{M} = \big\{ aP_\star + bP_v + cP_\perp : a, b, c \in \mathbb{R} \big\},$$

which is a three-dimensional commutative subalgebra of $\mathbb{R}^{d \times d}$. In particular, $\mathcal{M}$ is closed under addition and matrix multiplication, and all elements of $\mathcal{M}$ commute with each other.

**Step 2: structure of $G(M)$.** Recall that

$$G(M) = 8QCQ + 4\,\mathrm{Tr}(CQ)\,Q, \qquad C = M - P_\star, \qquad Q = I + \lambda P_v.$$

If $M \in \mathcal{M}$, then $C \in \mathcal{M}$ and $Q \in \mathcal{M}$. Since $\mathcal{M}$ is a commutative algebra, it follows that $QCQ \in \mathcal{M}$. Moreover, $\mathrm{Tr}(CQ)$ is a scalar, hence $\mathrm{Tr}(CQ)Q \in \mathcal{M}$. Therefore

$$G(M) \in \mathcal{M}, \qquad \text{and} \quad G(M)M = MG(M).$$

**Step 3: invariance under GD.** Recall that the GD iterate is

$$W_{k+1} = W_k - \eta\,G(M_k)W_k, \qquad M_k := W_k W_k^\top.$$

Assume that $M_k \in \mathcal{M}$. By Step 2, $G(M_k) \in \mathcal{M}$, hence $I - \eta G(M_k) \in \mathcal{M}$. Using the update and $M_k = W_k W_k^\top$, we obtain

$$M_{k+1} = W_{k+1}W_{k+1}^\top = \big(W_k - \eta\,G(M_k)W_k\big)\big(W_k - \eta\,G(M_k)W_k\big)^\top$$
$$= \big(I - \eta G(M_k)\big)\,M_k\,\big(I - \eta G(M_k)\big)^\top.$$

Since $G(M_k)$ is symmetric, $\big(I - \eta G(M_k)\big)^\top = \big(I - \eta G(M_k)\big)$. Because $\mathcal{M}$ is a commutative subalgebra (Step 1), it is closed under multiplication and addition; therefore

$$\big(I - \eta G(M_k)\big) \in \mathcal{M}, \quad M_k \in \mathcal{M} \implies M_{k+1} \in \mathcal{M}.$$

By induction, $M_k \in \mathcal{M}$ for all $k \geq 0$ whenever $M_0 \in \mathcal{M}$.

**Step 4: invariance under SpecGD.** The SpecGD iterate is

$$W_{k+1} = W_k - \eta \, \mathrm{polar}\big(G(M_k)W_k\big), \qquad M_k := W_k W_k^\top.$$

Assume that $M_k \in \mathcal{M}$. By Step 2, $G(M_k) \in \mathcal{M}$, and in particular $M_k$ and $G(M_k)$ commute. Hence there exists an orthogonal matrix $U$ whose first two columns are $w_\star$ and $v$ such that

$$M_k = U \, \mathrm{diag}(a_k, b_k, c_k I_{d-2}) \, U^\top, \qquad G(M_k) = U \, \mathrm{diag}(g_{\star,k}, g_{v,k}, g_{\perp,k} I_{d-2}) \, U^\top$$

for some scalars $a_k, b_k, c_k \geq 0$ and $g_{\star,k}, g_{v,k}, g_{\perp,k} \in \mathbb{R}$. Since $M_k = W_k W_k^\top \succeq 0$, there exists an orthogonal matrix $V$ such that

$$W_k = U \, \mathrm{diag}(\sqrt{a_k}, \sqrt{b_k}, \sqrt{c_k} I_{d-2}) \, V^\top.$$

Consequently,

$$G(M_k)W_k = U \, \mathrm{diag}(g_{\star,k}\sqrt{a_k}, \, g_{v,k}\sqrt{b_k}, \, g_{\perp,k}\sqrt{c_k} I_{d-2}) \, V^\top.$$

Thus $G(M_k)W_k$ has the same left/right singular spaces $U, V$ and the same $(1,1)$, $(2,2)$ and bulk block structure. Taking the polar factor, therefore, preserves this block structure:

$$\mathrm{polar}\big(G(M_k)W_k\big) = U \, \mathrm{diag}(s_{\star,k}, \, s_{v,k}, \, s_{\perp,k} I_{d-2}) \, V^\top,$$

where $s_{\star,k} = \mathrm{Sign}(g_{\star,k}\sqrt{a_k})$, $s_{v,k} = \mathrm{Sign}(g_{v,k}\sqrt{b_k})$, $s_{\perp,k} = \mathrm{Sign}(g_{\perp,k}\sqrt{c_k})$. Plugging this into the SpecGD update yields

$$W_{k+1} = U \, \mathrm{diag}\big(\sqrt{a_k} - \eta s_{\star,k}, \, \sqrt{b_k} - \eta s_{v,k}, \, (\sqrt{c_k} - \eta s_{\perp,k})I_{d-2}\big) \, V^\top,$$

and hence

$$M_{k+1} = W_{k+1}W_{k+1}^\top = U \, \mathrm{diag}\Big((\sqrt{a_k} - \eta s_{\star,k})^2, \, (\sqrt{b_k} - \eta s_{v,k})^2, \, (\sqrt{c_k} - \eta s_{\perp,k})^2 I_{d-2}\Big) U^\top \in \mathcal{M}.$$

By induction, $M_k \in \mathcal{M}$ for all $k \geq 0$ whenever $M_0 \in \mathcal{M}$.

$\square$

*Remark* A.3 (Extension to continuous time). The invariance argument above is stated and proved in discrete time, which is the main setting of this work. For GD, the extension to continuous time is straightforward: the gradient flow $\dot{W} = -G(M)W$ (and the induced flow $\dot{M} = -(G(M)M + MG(M))$) has a locally Lipschitz right-hand side, hence admits a unique solution, and tangency of the vector field to $\mathcal{M}$ implies invariance.

For SpecGF, it is less immediate, since the polar operator is non-smooth and the corresponding ODE is not globally Lipschitz. A clean way to define SpecGF is therefore as a vanishing-step-size limit of SpecGD: one considers the piecewise-linear interpolation of the SpecGD iterates and extracts convergent subsequences as the step size tends to zero. Since SpecGD preserves $\mathcal{M}$ at every step and $\mathcal{M}$ is closed, any such limit trajectory remains in $\mathcal{M}$. This yields invariance for SpecGF without invoking uniqueness or smoothness of the continuous-time dynamics.

Lemma 3.1 shows that if $M(0) \in \mathcal{M}$, then the matrix flow induced by either GD or SpecGD remains confined to $\mathcal{M}$ for all $t \geq 0$. As a consequence, the evolution of $M(t)$ is fully characterized by the three scalar coefficients $a(t)$, $b(t)$, and $c(t)$. The following propositions make this reduction explicit by deriving the corresponding ODEs governing the dynamics of these quantities.

**Proposition 5.1.** *The dynamics of GF is described by the following ODEs:*

$$\dot{\tilde{a}}(t) = -2g_{w_\star}(t)\tilde{a}(t), \tag{6}$$

$$\dot{\tilde{b}}(t) = -2g_v(t)\tilde{b}(t), \tag{7}$$

$$\dot{\tilde{c}}(t) = -2g_\perp(t)\tilde{c}(t), \tag{8}$$

*where*

$$g_{w_\star}(t) := 8(\tilde{a}(t) - 1) + 4(\tilde{r}(t) - 1),$$
$$g_v(t) := 8(1 + \lambda)^2\tilde{b}(t) + 4(1 + \lambda)(\tilde{r}(t) - 1), \tag{9}$$
$$g_\perp(t) := 8\tilde{c}(t) + 4(\tilde{r}(t) - 1),$$

*and* $\tilde{r}(t) := \tilde{a}(t) + (1 + \lambda)\tilde{b}(t) + (d - 2)\tilde{c}(t).$

*Proof.* A direct computation then yields

$$C = (a - 1)\, w_\star w_\star^\top + b\, vv^\top + c\, P_\perp.$$

Since $w_\star \perp v$ and $P_\perp$ projects onto their orthogonal complement, the projectors $w_\star w_\star^\top$, $vv^\top$, and $P_\perp$ are mutually orthogonal. Using this decomposition and the fact that $Q$ acts as multiplication by $1 + \lambda$ on $\mathrm{span}\{v\}$ and as the identity on its orthogonal complement, we obtain

$$QCQ = (a - 1)\, w_\star w_\star^\top + (1 + \lambda)^2 b\, vv^\top + c\, P_\perp.$$

Moreover, taking the trace gives

$$\mathrm{Tr}(CQ) = (a - 1) + (1 + \lambda)b + (d - 2)c = r - 1,$$

where $r = a + (1 + \lambda)b + (d - 2)c$. Substituting these expressions into the definition of $G(M)$ yields the eigenvalues stated in (9).

Next, observe that both $M$ and $G(M)$ are diagonal in the same orthogonal decomposition

$$\mathbb{R}^d = \mathrm{span}\{w_\star\}\ \oplus\ \mathrm{span}\{v\}\ \oplus\ \mathrm{span}\{w_\star, v\}^\perp.$$

In particular, they commute. As a consequence, the GD flow,

$$\dot{M} = -\big(G(M)M + MG(M)\big),$$

simplifies on $\mathcal{M}$ to

$$\dot{M} = -2G(M)M.$$

Finally, since $G(M)$ acts by scalar multiplication on each of the three invariant subspaces, the evolution of $M$ preserves this decomposition. Projecting the above equation onto $\mathrm{span}\{w_\star\}$, $\mathrm{span}\{v\}$, and $\mathrm{span}\{w_\star, v\}^\perp$ respectively yields the scalar ODEs (6) for $a(t)$, $b(t)$, and $c(t)$. $\qquad\square$

**Proposition 5.3.** *The dynamics of SpecGF is described by the following ODEs:*

$$\begin{aligned}
\dot{a}(t) &= -2\, \mathrm{Sign}(g_{w_\star}(t))\, \sqrt{a(t)}, \\
\dot{b}(t) &= -2\, \mathrm{Sign}(g_v(t))\, \sqrt{b(t)}, \\
\dot{c}(t) &= -2\, \mathrm{Sign}(g_\perp(t))\, \sqrt{c(t)},
\end{aligned} \tag{10}$$

*with the convention* $\mathrm{Sign}(0) := 0$. *Equivalently, in square-root variables* $\alpha(t) := \sqrt{a(t)}$, $\beta(t) := \sqrt{b(t)}$, *and* $\gamma(t) := \sqrt{c(t)}$,

$$\dot{\alpha}(t) = -\,\mathrm{Sign}(g_{w_\star}(t)), \tag{11}$$
$$\dot{\beta}(t) = -\,\mathrm{Sign}(g_v(t)), \tag{12}$$
$$\dot{\gamma}(t) = -\,\mathrm{Sign}(g_\perp(t)). \tag{13}$$

*Proof.* Fix a time $t$ such that $M(t) \in \mathcal{M}$, and write

$$M = a\, w_\star w_\star^\top + b\, vv^\top + c\, P_\perp.$$

Since $w_\star \perp v$, there exists an orthogonal matrix

$$U = \big[w_\star,\ v,\ U_\perp\big] \in \mathbb{R}^{d \times d}$$

such that

$$M = U\, \mathrm{diag}(a, b, c, \ldots, c)\, U^\top.$$

On $\mathcal{M}$, the matrix $G(M)$ is diagonal in the same decomposition, hence

$$G(M) = U \ \mathrm{diag}(g_{w_\star}, g_v, g_\perp, \ldots, g_\perp) \, U^\top.$$

Let $W$ be any factor of $M$, i.e. $M = WW^\top$. Choose an SVD of $W$ whose left singular vectors are $U$:

$$W = U \begin{bmatrix} D & 0 \end{bmatrix} V^\top, \qquad D = \mathrm{diag}(\sqrt{a}, \sqrt{b}, \sqrt{c}, \ldots, \sqrt{c}),$$

where $V \in \mathbb{R}^{m \times m}$ is orthogonal and the block $[D\ 0]$ is interpreted in the natural way when $m \neq d$ (in particular, if some of $a, b, c$ vanish then the corresponding diagonal entries of $D$ are 0).

Set

$$\Gamma := \mathrm{diag}(g_{w_\star}, g_v, g_\perp, \ldots, g_\perp).$$

Then

$$G(M)W = U\Gamma U^\top U \begin{bmatrix} D & 0 \end{bmatrix} V^\top = U \begin{bmatrix} \Gamma D & 0 \end{bmatrix} V^\top.$$

We now compute the polar factor. We use the orthogonal equivariance of the polar map: for orthogonal $U, V$, $\mathrm{polar}(UAV^\top) = U \, \mathrm{polar}(A) \, V^\top$. Hence

$$\mathrm{polar}\big(G(M)W\big) = U \, \mathrm{polar}\Big( \begin{bmatrix} \Gamma D & 0 \end{bmatrix} \Big) V^\top.$$

Since $\Gamma D$ is diagonal, the rows of $[\Gamma D\ 0]$ are mutually orthogonal. A direct calculation from $\mathrm{polar}(A) = A(A^\top A)^{-1/2}$ then gives

$$\mathrm{polar}\Big( \begin{bmatrix} \Gamma D & 0 \end{bmatrix} \Big) = \begin{bmatrix} S & 0 \end{bmatrix}, \qquad S = \mathrm{diag}\big(\mathrm{Sign}(g_{w_\star}), \mathrm{Sign}(g_v), \mathrm{Sign}(g_\perp), \ldots, \mathrm{Sign}(g_\perp)\big),$$

with the convention $\mathrm{Sign}(0) = 0$ (note that if a diagonal entry of $D$ equals 0, the corresponding row of $[\Gamma D\ 0]$ is zero and remains zero after applying $\mathrm{polar}$).

Therefore the SpecGD flow $\dot{W} = -\mathrm{polar}(G(M)W)$ satisfies

$$\dot{W} = -U \begin{bmatrix} S & 0 \end{bmatrix} V^\top.$$

Using $\dot{M} = \dot{W}W^\top + W\dot{W}^\top$ and the above decompositions, we obtain

$$\dot{M} = -U\Big(SD + DS\Big)U^\top, \qquad D = \mathrm{diag}(\sqrt{a}, \sqrt{b}, \sqrt{c}, \ldots, \sqrt{c}).$$

Since $S$ and $D$ are diagonal, $SD + DS = 2SD$, and thus

$$\dot{M} = -2\,U \ \mathrm{diag}\big(\mathrm{Sign}(g_{w_\star})\sqrt{a}, \ \mathrm{Sign}(g_v)\sqrt{b}, \ \mathrm{Sign}(g_\perp)\sqrt{c}, \ldots, \mathrm{Sign}(g_\perp)\sqrt{c}\big) U^\top.$$

Projecting onto $\mathrm{span}\{w_\star\}$, $\mathrm{span}\{v\}$, and $\mathrm{span}\{w_\star, v\}^\perp$ yields

$$\dot{a} = -2\ \mathrm{Sign}(g_{w_\star})\ \sqrt{a}, \qquad \dot{b} = -2\ \mathrm{Sign}(g_v)\ \sqrt{b}, \qquad \dot{c} = -2\ \mathrm{Sign}(g_\perp)\ \sqrt{c},$$

which is (10). Finally, writing $\alpha = \sqrt{a}, \beta = \sqrt{b}, \gamma = \sqrt{c}$ and using $\dot{\alpha} = \dot{a}/(2\sqrt{a})$ (with the convention $\dot{\alpha} = 0$ when $a = 0$) gives (11). $\qquad\square$

## A.3. Values at initialization

We initialize

$$W(0) = \theta\,U, \qquad U \in \mathrm{St}(d, d) = \mathsf{O}(d).$$

Since $UU^\top = I$, the induced matrix initialization is

$$M(0) = \theta^2 I = \theta^2\,(P_\star + P_v + P_\perp) \in \mathcal{M}.$$

Therefore,
$$a_0 = b_0 = c_0 = \theta^2.$$

The corresponding weighted mass is
$$r_0 := a_0 + (1 + \lambda)b_0 + (d - 2)c_0 = (d + \lambda)\theta^2.$$

We choose $\theta^2 = \rho_0/(d + \lambda)$ with $\rho_0 = o(1)$, so that
$$r_0 = \rho_0, \qquad a_0 = b_0 = c_0 = \frac{\rho_0}{d + \lambda} =: \mu.$$

In particular, the initial alignment satisfies
$$\text{Align}(0) = \frac{a_0}{\sqrt{a_0^2 + b_0^2 + (d - 2)c_0^2}} = \frac{1}{\sqrt{d}}.$$

## B. Dynamics analysis of continuous GD

Throughout this subsection we work on the invariant manifold $\mathcal{M}$ guaranteed by Lemma 3.1. Hence $M(t) = a(t)\, w_\star w_\star^\top + b(t)\, vv^\top + c(t)\, P_\perp$ and
$$r(t) = a(t) + (1 + \lambda)b(t) + (d - 2)c(t).$$

Recall that the reduced GD dynamics is given by
$$\dot{a} = a\,(24 - 16a - 8r), \qquad \dot{b} = 8(1 + \lambda)(1 - r)\,b - 16(1 + \lambda)^2 b^2, \qquad \dot{c} = 8(1 - r)\,c - 16c^2. \tag{16}$$

We will also track the (Frobenius) alignment with the signal direction:
$$\text{Align}(t) := \frac{\langle M(t), w_\star w_\star^\top \rangle_F}{\|M(t)\|_F \cdot \|w_\star w_\star^\top\|_F} = \frac{a(t)}{\sqrt{a(t)^2 + b(t)^2 + (d - 2)c(t)^2}}.$$

Define the *spike mass* and *bulk mass*
$$B(t) := (1 + \lambda)b(t), \qquad C(t) := (d - 2)c(t), \qquad \text{so that} \qquad r(t) = a(t) + B(t) + C(t).$$

Multiplying the $b$-equation in (16) by $(1 + \lambda)$ yields
$$\dot{B} = 8(1 + \lambda)\,B\big((1 - r) - 2B\big) = 8(1 + \lambda)\,B\big(1 - a - C - 3B\big). \tag{17}$$

In particular,
$$\dot{b} \leq 0 \quad \Longleftrightarrow \quad \dot{B} \leq 0 \quad \Longleftrightarrow \quad B \geq \frac{1 - a - C}{3}. \tag{18}$$

We will use this to define the "turning" time when the spike starts contracting.

**Definition B.1** (Key stopping times). Let $\rho \in (0, 1/12)$ and $\varepsilon \in (0, 1/4]$ be fixed constants. Define
$$\begin{aligned}
T_{1a} &:= \inf\{t \geq 0 : r(t) \geq \rho\}, \\
T_1 &:= \inf\{t \geq T_{1a} : \dot{b}(t) \leq 0\}, \\
T_{2a} &:= \inf\{t \geq T_1 : r(t) \geq 1 - \varepsilon\}, \\
T_2 &:= \inf\{t \geq T_{2a} : a(t) \geq \rho\}.
\end{aligned} \tag{19}$$

**Four-phase picture.** With these stopping times, the GD trajectory admits the following four phases:

- **Phase Ia (fast spike escape):** For $t \in [0, T_{1a}]$, $r$ grows from $\rho_0 = o(1)$ to $\rho$ and the spike term drives the growth. Furthermore, during this phase $a(t), b(t)$ and $c(t)$ grow exponentially in a decoupled way.

- **Phase Ib (spike saturation to the turning point):** For $t \in [T_{1a}, T_1]$, $B$ rapidly approaches the point where $\dot{b}$ changes sign.

- **Phase IIa (bulk inflation to the $r \simeq 1$ band):** For $t \in [T_1, T_{2a}]$, $r$ grows from $\simeq 1/3$ to $\simeq 1$ thanks to the growth of the bulk mass $C(t) = (d-2)c(t)$, while the signal $a(t)$ remains negligible, and the alignment is still small.

- **Phase IIb (signal amplification and alignment):** $t \in [T_{2a}, T_2]$, where the signal coefficient $a(t)$ grows from a tiny value to a constant ($a \geq \rho$), and consequently the alignment becomes close to 1 (because $\lambda b(t)^2$ and $(d-2)c(t)^2$ are $o(1)$).

The following bounds will be used repeatedly.

**Lemma 5.5.** *For all $t \geq 0$, one has*

$$0 \leq \tilde{a}(t) \leq 1, \; 0 \leq \tilde{B}(t) \leq \frac{1}{3}, \; 0 \leq \tilde{C}(t) \leq 1.$$

*Proof.* First, note that positivity is a consequence of the definition of the functions $a$, $b$ and $c$.

The reduced population dynamics define a smooth autonomous ODE in $(a, B, C)$. Hence $a(t), B(t), C(t)$ are $C^1$ functions of time. As a result, if one of the variables were to exit a given interval, it would first have to reach the boundary of that interval, where we may evaluate the sign of the corresponding derivative.

**Barrier for $a$.** If $a(t) = 1$, then $r(t) \geq a(t) = 1$, so

$$\dot{a}(t) = a(t)\big(24 - 16a(t) - 8r(t)\big) = 24 - 16 - 8r(t) = 8(1 - r(t)) \leq 0.$$

Thus $a$ cannot cross above 1.

**Barrier for $B$.** Recall equation (17). If $B(t) = 1/3$, then since $a(t), C(t) \geq 0$,

$$1 - a(t) - C(t) - 3B(t) \leq 1 - 3B(t) \leq 0,$$

and (17) gives $\dot{B}(t) \leq 0$. Hence $B$ cannot cross above $1/3$.

**Barrier for $C$.** Write $\dot{C} = (d-2)\dot{c} = 8(1 - r)C - 16C^2/(d-2)$. If $C(t) = 1$, then $r(t) \geq C(t) = 1$, so

$$\dot{C}(t) \leq -\frac{16}{d-2} < 0,$$

and $C$ cannot cross above 1.

Finally $r = a + B + C \leq 1 + 1/3 + 1 = 7/3$. $\qquad\square$

### B.1. Phase Ia: fast spike-driven escape to $r = \rho$

We quantify the spike-driven escape to the macroscopic level $r = \rho$ and show that at time $T_{1a}$ the spurious spike component $B(t) = (1 + \lambda)b(t)$ already carries essentially all the mass, while the signal component $a(t)$ and the bulk component $C(t) = (d-2)c(t)$ remain negligible.

**Proposition B.2.** *Assume GD is initialized as in Section 2 with $\rho_0 = \log^{-c_0} d$ and Assumption A is satisfied. Then the following holds.*

1. **(Exponential growth while $r$ is small).** *For all $t \in [0, T_{1a}]$ one has $r(t) \leq \rho$, and*

$$a(0)\, e^{24(1-\rho)t} \; \leq \; a(t) \; \leq \; a(0)\, e^{24t}, \tag{20}$$

$$b(0)\, e^{8(1+\lambda)(1-3\rho)t} \; \leq \; b(t) \; \leq \; b(0)\, e^{8(1+\lambda)t}, \tag{21}$$

$$c(0)\, e^{\left(8(1-\rho)-16\rho/(d-2)\right)t} \; \leq \; c(t) \; \leq \; c(0)\, e^{8t}. \tag{22}$$

2. **(Escape time scale).** *The escape time satisfies*

$$T_{1a} = O\Big(\frac{1}{\lambda}\log d\Big). \tag{23}$$

3. **(Spike dominance at escape).** *Let $B(t) = (1+\lambda)b(t)$ and $C(t) = (d-2)c(t)$ so that $r(t) = a(t) + B(t) + C(t)$.*
   *Then*

$$a(T_{1a}) = O(\frac{\rho_0}{d}), \qquad C(T_{1a}) = O(\rho_0), \tag{24}$$

*and therefore, using $r(T_{1a}) = \rho$,*

$$B(T_{1a}) = \rho - a(T_{1a}) - C(T_{1a}) = \rho\,(1 + o(1)). \tag{25}$$

*Proof.* Fix $\rho \in (0, 1/12)$ and recall that by definition of $T_{1a}$, $r(t) \le \rho$ for all $t \in [0, T_{1a}]$.

**Step 1: exponential growth bounds for $a, b, c$ on $[0, T_{1a}]$.**

*Bounds for a.* For $t \in [0, T_{1a}]$, since $a(t) \le r(t) \le \rho$, we have

$$\frac{\dot{a}(t)}{a(t)} = 24 - 16a(t) - 8r(t) \le 24, \qquad \frac{\dot{a}(t)}{a(t)} \ge 24 - 16\rho - 8\rho = 24(1 - \rho).$$

Integrating yields (20).

*Bounds for b.* Writing $B(t) = (1 + \lambda)b(t)$, we have

$$\frac{\dot{b}(t)}{b(t)} = 8(1 + \lambda)\big(1 - a(t) - (d - 2)c(t) - 3B(t)\big).$$

Since $a(t) + (d - 2)c(t) + B(t) = r(t) \le \rho$ and $B(t) \le r(t) \le \rho$, it follows that

$$1 - 3\rho \le 1 - a(t) - (d - 2)c(t) - 3B(t) \le 1.$$

Integrating yields (21).

*Bounds for c.* We have

$$\frac{\dot{c}(t)}{c(t)} = 8(1 - r(t)) - 16c(t).$$

Since $r(t) \le \rho$ and $c(t) \le r(t)/(d - 2) \le \rho/(d - 2)$ on $[0, T_{1a}]$,

$$8(1 - \rho) - \frac{16\rho}{d - 2} \le \frac{\dot{c}(t)}{c(t)} \le 8.$$

Integrating yields (22).

**Step 2: escape time scale.** Since $r(t) \le \rho$ on $[0, T_{1a}]$, we have $B(t) \le \rho$. Using the lower bound on $b(t)$,

$$B(t) \ge (1 + \lambda)b(0) \exp\big(8(1 + \lambda)(1 - 3\rho)t\big).$$

At time $T_{1a}$, this gives

$$(1 + \lambda)b(0) \exp\big(8(1 + \lambda)(1 - 3\rho)T_{1a}\big) \le \rho,$$

and hence

$$T_{1a} \le \frac{1}{8(1 + \lambda)(1 - 3\rho)} \log\Big(\frac{\rho}{(1 + \lambda)b(0)}\Big).$$

Since $b(0) \asymp \rho_0/(d + \lambda)$, we have

$$\log\Big(\frac{\rho}{(1 + \lambda)b(0)}\Big) = \Theta(\log d),$$

which proves (23).

**Step 3: spike dominance at escape.** Using the upper bounds from Step 1 and $T_{1a} = O(\frac{\log d}{\lambda})$, we have

$$a(T_{1a}) \le a(0)e^{24T_{1a}}, \qquad c(T_{1a}) \le c(0)e^{8T_{1a}}.$$

With $a(0) \asymp c(0) \asymp \rho_0/(d+\lambda)$ and $\lambda \gtrsim \log d$,

$$e^{O(\log d/\lambda)} = O(1),$$

so

$$a(T_{1a}) \lesssim \frac{\rho_0}{d+\lambda} = o(1), \qquad C(T_{1a}) = (d-2)c(T_{1a}) \lesssim \rho_0 = o(1).$$

Since $r(T_{1a}) = \rho$, this implies

$$B(T_{1a}) = \rho - a(T_{1a}) - C(T_{1a}) = \rho(1 + o(1)),$$

which completes the proof. □

### B.2. Phase Ib: rapid saturation of the spike to its turning point ($r \simeq 1/3$)

After Stage I, the spurious spike mass $B(t) = (1+\lambda)b(t)$ is already of constant order, while the signal and bulk components remain negligible. Define

$$K(t) := \frac{1 - a(t) - C(t)}{3}.$$

Phase Ib corresponds to the short time interval during which $B(t)$ rapidly increases from level $\rho$ to its turning point, where $\dot{b}$ changes sign, i.e. where $B(t) = K(t)$.

**Proposition B.3.** *Let $T_{1a}, T_1$ be as in Definition B.1. Then the following holds.*

1. *(Fast time scale).* The duration of Phase Ib satisfies

$$T_1 - T_{1a} = O\left(\frac{1}{\lambda} \log \frac{1}{\rho_0}\right) = O\left(\frac{\log \log d}{\lambda}\right). \tag{26}$$

2. *(Stability of slow components).* The signal and bulk components remain essentially unchanged during Stage II:

$$a(T_1) = a(T_{1a})\,(1 + o(1)), \qquad C(T_1) = C(T_{1a})\,(1 + o(1)). \tag{27}$$

3. *(Turning location).* At time $T_1$,

$$B(T_1) = \frac{1}{3}\,(1 + o(1)), \qquad r(T_1) = \frac{1}{3}\,(1 + o(1)). \tag{28}$$

*Proof.* Recall that $B$ satisfies

$$\dot{B}(t) = 8(1+\lambda)\,B(t)\bigl(1 - a(t) - C(t) - 3B(t)\bigr). \tag{29}$$

Define $E(t) := K(t) - B(t)$. By definition of $T_1$ and (18), for $t \in [T_{1a}, T_1)$ we have $\dot{b}(t) > 0$, hence $\dot{B}(t) > 0$, which implies $B(t) < K(t)$. By continuity,

$$B(T_1) = K(T_1). \tag{30}$$

**Step 1: monotonicity and uniform bounds for $B$ on $[T_{1a}, T_1]$.** Since $\dot{B}(t) > 0$ on $[T_{1a}, T_1)$, $B$ is increasing and therefore

$$B(t) \ge B(T_{1a}) = \rho(1 + o(1)) \ge \rho/2, \qquad t \in [T_{1a}, T_1], \tag{31}$$

for $d$ large enough. Moreover, because $B(t) \le K(t) \le 1/3$ on $[T_{1a}, T_1]$,

$$0 \le B(t) \le \frac{1}{3}, \qquad t \in [T_{1a}, T_1]. \tag{32}$$

**Step 2: Duhamel formula.** Differentiating $E(t) = K(t) - B(t)$ and using (29) yields

$$\dot{E}(t) = \dot{K}(t) - 24(1+\lambda)B(t)E(t), \qquad \dot{K}(t) = -\frac{\dot{a}(t) + \dot{C}(t)}{3}. \tag{33}$$

Apply variation-of-constants on $[T_{1a}, T_1]$:

$$E(T_1) = E(T_{1a})e^{-\int_{T_{1a}}^{T_1} 24(1+\lambda)B(u)\,du} + \int_{T_{1a}}^{T_1} e^{-\int_s^{T_1} 24(1+\lambda)B(u)\,du}\,\dot{K}(s)\,ds. \tag{34}$$

We now upper bound the forcing term $\dot{K}$ using only crude growth bounds. Since $\dot{a} \le 24a$ and $\dot{C} \le 8C$, for all $t \ge T_{1a}$,

$$a(t) + C(t) \le (a(T_{1a}) + C(T_{1a}))e^{24(t - T_{1a})}. \tag{35}$$

Moreover, $|\dot{K}(t)| \le \frac{|\dot{a}(t)| + |\dot{C}(t)|}{3} \le 8(a(t) + C(t))$, hence

$$|\dot{K}(t)| \le 8(a(T_{1a}) + C(T_{1a}))e^{24(t - T_{1a})}. \tag{36}$$

Let $\Delta := T_1 - T_{1a}$. Using (31) and $(1 + \lambda) \ge \lambda$, we have

$$\int_{T_{1a}}^{T_1} 24(1+\lambda)B(u)\,du \ge 24\lambda \cdot \frac{\rho}{2}\Delta =: \mu_\rho \lambda \Delta, \qquad \mu_\rho := 12\rho,$$

and similarly $\int_s^{T_1} 24(1+\lambda)B(u)\,du \ge \mu_\rho\lambda(T_1 - s)$. Taking absolute values in (34) and using (30) gives

$$E(T_{1a})e^{-\mu_\rho\lambda\Delta} \le \int_{T_{1a}}^{T_1} e^{-\mu_\rho\lambda(T_1 - s)}\,|\dot{K}(s)|\,ds.$$

Plugging (36) and changing variables $u = T_1 - s$ yields

$$E(T_{1a})e^{-\mu_\rho\lambda\Delta} \le 8(a(T_{1a}) + C(T_{1a}))e^{24\Delta}\int_0^\Delta e^{-\mu_\rho\lambda u}\,du \le \frac{8}{\mu_\rho\lambda}(a(T_{1a}) + C(T_{1a}))e^{24\Delta}.$$

Since $E(T_{1a}) = \frac{1}{3} - \rho + o(1) = \Theta(1)$ and $a(T_{1a}) + C(T_{1a}) = O(\rho_0)$ (Phase Ia), we obtain

$$e^{-(\mu_\rho\lambda - 24)\Delta} \lesssim \frac{\rho_0}{\lambda}.$$

Because $\lambda \gtrsim \log d$ and $\rho$ is a fixed constant, $\mu_\rho\lambda \gg 24$ for $d$ large enough, hence

$$\Delta = T_1 - T_{1a} = O\left(\frac{1}{\lambda}\log\frac{1}{\rho_0}\right) = O\left(\frac{\log\log d}{\lambda}\right) = o(1). \tag{37}$$

**Step 3: bootstrap-free bound $r(t) \le 1/2$ on $[T_{1a}, T_1]$.** By (35) and (37),

$$\sup_{t \in [T_{1a}, T_1]} (a(t) + C(t)) \le (a(T_{1a}) + C(T_{1a}))e^{24\Delta} = O(\rho_0)(1 + o(1)) = o(1).$$

Together with $B(t) \le 1/3$ from (32), this yields for all $t \in [T_{1a}, T_1]$,

$$r(t) = a(t) + B(t) + C(t) \le \frac{1}{3} + o(1)$$

for $d$ large enough.

**Step 4: stability of the slow components.** Since $\Delta = o(1)$ and $\dot{a} \le 24a$, $\dot{C} \le 8C$,

$$a(T_1) = a(T_{1a})e^{O(\Delta)} = a(T_{1a})(1 + o(1)), \qquad C(T_1) = C(T_{1a})e^{O(\Delta)} = C(T_{1a})(1 + o(1)).$$

**Step 5: turning location.** Using $B(T_1) = K(T_1)$ from (30),

$$B(T_1) = \frac{1 - a(T_1) - C(T_1)}{3} = \frac{1}{3}(1 + o(1)),$$

and therefore

$$r(T_1) = a(T_1) + B(T_1) + C(T_1) = \frac{1}{3}(1 + o(1)). \qquad \square$$

**B.3. Phase IIa: bulk inflation to the $r \simeq 1$ band**

At the turning time $T_1$, the previous analysis yields

$$B(T_1) = \frac{1}{3}(1 + o(1)), \qquad r(T_1) = \frac{1}{3}(1 + o(1)), \qquad a(T_1) = \Theta(\mu), \qquad C(T_1) = \Theta(\rho_0). \tag{38}$$

Phase IIa analyzes the evolution on $[T_1, T_{2a}]$, where $T_{2a} := \inf\{t \geq T_1 : r(t) \geq 1 - \varepsilon\}$.

**Proposition B.4.** *Let $T_1, T_{2a}$ be as in Definition B.1. Then the following hold:*

1. *(**Time scale**). For fixed $\varepsilon \in (0, 1/4]$,*

$$T_{2a} - T_1 = \Theta\big(\log(1/\rho_0)\big) = \Theta(\log \log d). \tag{39}$$

2. *(**Bulk is order one at** $T_{2a}$). At time $T_{2a}$,*

$$C(T_{2a}) \geq \frac{2}{3} - \varepsilon - o(1), \qquad \text{and hence} \qquad c(T_{2a}) = \Theta\Big(\frac{1}{d}\Big). \tag{40}$$

3. *(**Signal is still negligible at** $T_{2a}$). One has*

$$a(T_{2a}) = \mu \, \rho_0^{-O(1)} = \frac{\log^{O(1)} d}{d + \lambda} = o(1). \tag{41}$$

4. *(**Alignment is still small at** $T_{2a}$). In fact,*

$$\mathrm{Align}(T_{2a}) \leq O\big(\sqrt{d}\, a(T_{2a})\big) = o(1). \tag{42}$$

*Proof.* **Step 1: $C(T_{2a})$ must be of constant order.** At time $T_{2a}$ we have $r(T_{2a}) = 1 - \varepsilon$ and $r = a + B + C$, hence

$$C(T_{2a}) = 1 - \varepsilon - a(T_{2a}) - B(T_{2a}).$$

By Lemma 5.5, $B(T_{2a}) \leq 1/3$, and by positivity $a(T_{2a}) \geq 0$, so

$$C(T_{2a}) \geq 1 - \varepsilon - \frac{1}{3} - a(T_{2a}) = \frac{2}{3} - \varepsilon - a(T_{2a}).$$

Since $a(T_{2a}) = o(1)$ (proved in Step 3 below), this gives (40).

**Step 2: time scale for $C$ to grow from $\Theta(\rho_0)$ to $\Theta(1)$.** For $t \in [T_1, T_{2a})$, by definition of $T_{2a}$ we have $r(t) \leq 1 - \varepsilon$, hence $1 - r(t) \geq \varepsilon$. Therefore from $\dot{c} = 8(1 - r)c - 16c^2$ we obtain the upper bound $\dot{c} \leq 8c$ and, using $c(t) \leq r(t)/(d - 2) \leq 1/(d - 2)$, for $d$ large enough also

$$-16c(t)^2 \geq -4\varepsilon \, c(t), \qquad \text{so that} \qquad \dot{c}(t) \geq 8\varepsilon c(t) - 16c(t)^2 \geq 4\varepsilon c(t), \qquad t \in [T_1, T_{2a}).$$

Multiplying by $(d - 2)$ yields for $C(t) = (d - 2)c(t)$

$$4\varepsilon \, C(t) \; \leq \; \dot{C}(t) \; \leq \; 8\, C(t), \qquad t \in [T_1, T_{2a}).$$

Integrating gives

$$C(T_1) \, e^{4\varepsilon(t - T_1)} \; \leq \; C(t) \; \leq \; C(T_1) \, e^{8(t - T_1)}.$$

Since $C(T_1) = \Theta(\rho_0)$ by (38) and $C(T_{2a}) = \Theta(1)$ by (40), it follows that $T_{2a} - T_1 = \Theta(\log(1/\rho_0))$, proving (39) (for fixed $\varepsilon$).

**Step 3: $a(T_{2a})$ remains small.** On $[T_1, T_{2a}]$, we always have $\dot{a} \leq 24a$, hence

$$a(T_{2a}) \leq a(T_1) \, e^{24(T_{2a} - T_1)}.$$

Using $a(T_1) = \Theta(\mu)$ from (38) and $T_{2a} - T_1 = \Theta(\log(1/\rho_0))$ from Step 2, we get $a(T_{2a}) = \mu\,\rho_0^{-O(1)}$, which is (41).

**Step 4: alignment bound.** From (40), we have $C(T_{2a}) \geq c_\varepsilon > 0$ for $d$ large enough. Then

$$(d-2)c(T_{2a})^2 = \frac{C(T_{2a})^2}{d-2} \geq \frac{c_\varepsilon^2}{d},$$

so the alignment denominator satisfies

$$\sqrt{a(T_{2a})^2 + b(T_{2a})^2 + (d-2)c(T_{2a})^2} \geq \sqrt{(d-2)c(T_{2a})^2} \geq \frac{c_\varepsilon}{\sqrt{d}}.$$

Therefore

$$\text{Align}(T_{2a}) = \frac{a(T_{2a})}{\sqrt{a(T_{2a})^2 + b(T_{2a})^2 + (d-2)c(T_{2a})^2}} \leq \frac{a(T_{2a})}{c_\varepsilon/\sqrt{d}} = O(\sqrt{d}\,a(T_{2a})),$$

which tends to 0 by (41). This proves (42). $\qquad\square$

### B.4. Phase IIb: signal amplification and alignment convergence

Stage IIb analyzes the interval $[T_{2a}, T_2]$, where $T_2 := \inf\{t \geq T_{2a} : a(t) \geq \rho\}$. The key point is that once $a$ reaches a constant, the alignment is close to 1 because $b = O(1/\lambda)$ and $(d-2)c^2 = O(1/d)$.

**Proposition B.5.** *Let $T_{2a}, T_2$ be as in Definition B.1. Then:*

1. *(**Time to macroscopic signal**). One has*

$$T_2 - T_{2a} = \Theta\left( \log \frac{1}{a(T_{2a})} \right) = \Theta(\log d). \tag{43}$$

2. *(**Alignment at time** $T_2$). At time $T_2$,*

$$\text{Align}(T_2) = 1 - O\left(\frac{1}{\lambda^2}\right) - o(1). \tag{44}$$

*Proof.* **Step 1: exponential growth bounds for $a$ on $[T_{2a}, T_2]$.** By definition of $T_2$, for all $t \in [T_{2a}, T_2]$ we have $a(t) \leq \rho$. Using $r = a + B + C$, we rewrite

$$\frac{\dot{a}(t)}{a(t)} = 24 - 16a(t) - 8r(t) = 24 - 16a(t) - 8(a(t) + B(t) + C(t)) = 24 - 24a(t) - 8B(t) - 8C(t).$$

By Lemma 5.5, for all $t \geq 0$ we have $B(t) \leq 1/3$ and $C(t) \leq 1$. Therefore, for all $t \in [T_{2a}, T_2]$,

$$\frac{\dot{a}(t)}{a(t)} \geq 24 - 24\rho - \frac{8}{3} - 8 = \frac{16}{3} - 24\rho =: \kappa_\rho, \tag{45}$$

where $\kappa_\rho > 0$ since $\rho < 1/12$. Also, since $r(t) \geq a(t)$ we have

$$\frac{\dot{a}(t)}{a(t)} = 24 - 16a(t) - 8r(t) \leq 24 - 24a(t) \leq 24.$$

Integrating these bounds on $[T_{2a}, t]$ gives, for all $t \in [T_{2a}, T_2]$,

$$a(T_{2a})\, e^{\kappa_\rho(t - T_{2a})} \leq a(t) \leq a(T_{2a})\, e^{24(t - T_{2a})}.$$

Setting $t = T_2$ and using $a(T_2) = \rho$ yields

$$\frac{1}{24} \log\left(\frac{\rho}{a(T_{2a})}\right) \leq T_2 - T_{2a} \leq \frac{1}{\kappa_\rho} \log\left(\frac{\rho}{a(T_{2a})}\right).$$

Since $\rho$ is an absolute constant, this proves

$$T_2 - T_{2a} = \Theta\left( \log \frac{1}{a(T_{2a})} \right).$$

Moreover, by Lemma B.4 we have $a(T_{2a}) = \frac{\log^{O(1)} d}{d+\lambda}$, hence $\log(1/a(T_{2a})) = \Theta(\log d)$, and (43) follows.

**Step 2: alignment at $T_2$.** At time $T_2$ we have $a(T_2) = \rho$. By Lemma 5.5, $B(T_2) \leq 1/3$, hence

$$b(T_2) = \frac{B(T_2)}{1+\lambda} \leq \frac{1}{3(1+\lambda)} = O\left(\frac{1}{\lambda}\right), \qquad \Rightarrow \qquad b(T_2)^2 = O\left(\frac{1}{\lambda^2}\right).$$

Also $C(T_2) \leq 1$ implies $c(T_2) = C(T_2)/(d-2) \leq 1/(d-2)$, and thus

$$(d-2)c(T_2)^2 \leq \frac{1}{d-2} = o(1).$$

Therefore,

$$\mathrm{Align}(T_2) = \frac{a(T_2)}{\sqrt{a(T_2)^2 + b(T_2)^2 + (d-2)c(T_2)^2}} = \frac{\rho}{\sqrt{\rho^2 + O(\lambda^{-2}) + o(1)}} = 1 - O\left(\frac{1}{\rho^2 \lambda^2}\right) - o(1).$$

Since $\rho$ is an absolute constant, this yields (44). $\qquad\square$

### B.5. Proof of Corollary D.2

Recall that $\mathrm{Align}(t) = \frac{a(t)}{\|M(t)\|_F}$ with

$$M(t) = a(t)\, w_\star w_\star^\top + b(t)\, vv^\top + c(t)\, P_\perp, \qquad \|M(t)\|_F^2 = a(t)^2 + b(t)^2 + (d-2)c(t)^2.$$

By Proposition B.3 (Phase Ia), with high probability we have at time $T_{1a}$

$$a(T_{1a}) \lesssim \mu = \frac{\rho_0}{d+\lambda}, \qquad B(T_{1a}) := (1+\lambda)b(T_{1a}) = \rho(1+o(1)), \qquad C(T_{1a}) := (d-2)c(T_{1a}) = o(1). \qquad (46)$$

We consider two regimes.

**Case 1: $\lambda \ll \sqrt{d}$ (spike-dominated Frobenius norm).** In this regime, the spike term dominates $\|M(T_{1a})\|_F$. Indeed,

$$b(T_{1a})^2 = \frac{B(T_{1a})^2}{(1+\lambda)^2} = \Theta\left(\frac{1}{(1+\lambda)^2}\right), \qquad (d-2)c(T_{1a})^2 = \frac{C(T_{1a})^2}{d-2} = o\left(\frac{1}{d}\right),$$

so $b(T_{1a})^2 \gg (d-2)c(T_{1a})^2$ and $b(T_{1a})$ controls the Frobenius norm. Therefore,

$$\mathrm{Align}(T_{1a}) = \frac{a(T_{1a})}{\|M(T_{1a})\|_F} \lesssim \frac{a(T_{1a})}{b(T_{1a})} = \frac{(1+\lambda)a(T_{1a})}{B(T_{1a})} \lesssim \frac{(1+\lambda)\mu}{\rho} = \frac{\rho_0}{\rho}\frac{1+\lambda}{d+\lambda} \lesssim \frac{\rho_0}{\rho}\frac{\lambda}{d},$$

where we used (46) and the bound $(1+\lambda)/(d+\lambda) \lesssim \lambda/d$ in this regime.

**Case 2: $\lambda \gtrsim \sqrt{d}$ (bulk-dominated Frobenius norm).** In this regime, the Frobenius norm is controlled by the bulk component. During Phase Ia we have $\dot{c}(t) > 0$ as long as $r(t) \leq \rho$, so $c(t)$ is nondecreasing on $[0, T_{1a}]$. Since $c(0) = \mu$, this yields the lower bound

$$c(T_{1a}) \geq c(0) = \mu.$$

Consequently,

$$\|M(T_{1a})\|_F \geq \sqrt{(d-2)}\, c(T_{1a}) \gtrsim \sqrt{d}\,\mu.$$

Combining with $a(T_{1a}) \lesssim \mu$ from (46), we obtain

$$\mathrm{Align}(T_{1a}) = \frac{a(T_{1a})}{\|M(T_{1a})\|_F} \lesssim \frac{\mu}{\sqrt{d}\,\mu} = \frac{1}{\sqrt{d}}.$$

# C. Dynamics analysis of continuous SpecGD

Throughout we work on the invariant manifold

$$\mathcal{M} = \left\{ M = a\, w_\star w_\star^\top + b\, v v^\top + c\, P_\perp \; : \; a, b, c \geq 0 \right\},$$

and write the trajectory as

$$M(t) = a(t)\, w_\star w_\star^\top + b(t)\, v v^\top + c(t)\, P_\perp, \qquad a(t), b(t), c(t) \geq 0.$$

Recall that from Proposition 5.3, the induced scalar dynamics is, for $t \geq 0$,

$$\dot{a}(t) = -2\, \mathrm{Sign}(g_{w_\star}(t))\, \sqrt{a(t)}, \quad \dot{b}(t) = -2\, \mathrm{Sign}(g_v(t))\, \sqrt{b(t)}, \quad \dot{c}(t) = -2\, \mathrm{Sign}(g_\perp(t))\, \sqrt{c(t)}, \tag{47}$$

where

$$g_{w_\star} = 4(r + 2a - 3), \qquad g_v = 4(1 + \lambda)(r + 2(1+\lambda)b - 1), \qquad g_\perp = 4(r + 2c - 1). \tag{48}$$

Equivalently, in square-root variables $\alpha = \sqrt{a}, \beta = \sqrt{b}, \gamma = \sqrt{c}$, whenever the coordinate is strictly positive one has

$$\dot{\alpha}(t) = -\mathrm{Sign}(g_{w_\star}(t)), \qquad \dot{\beta}(t) = -\mathrm{Sign}(g_v(t)), \qquad \dot{\gamma}(t) = -\mathrm{Sign}(g_\perp(t)). \tag{49}$$

**Stopping times and two-phase picture.** Fix a constant $\rho \in (0, 1/12)$ and define

$$T_1' := \inf\left\{ t \geq 0 : \; r(t) + 2(1+\lambda)b(t) \geq 1 \right\}, \qquad T_2' := \inf\left\{ t \geq 0 : \; a(t) \geq \rho \right\}. \tag{50}$$

We interpret these as:

- **Phase I** ($[0, T_1']$)**:** uniform growth $(\alpha, \beta, \gamma)$.

- **Phase II** ($[T_1', T_2']$ **and beyond):** the noise saturates, while the signal continues to grow until it reaches $a = \rho$, implying alignment.

## C.1. Phase I: isotropic expansion up to the first switch

Recall that at initialization we have $a(0) = b(0) = c(0) = \mu$ with $\mu = \rho_0/(d + \lambda)$.

**Lemma C.1.** *For all* $t \in [0, T_1']$,

$$\alpha(t) = \beta(t) = \gamma(t) = \sqrt{\mu} + t, \qquad \textit{equivalently} \qquad a(t) = b(t) = c(t) = (\sqrt{\mu} + t)^2.$$

*Moreover,*

$$T_1' = \Theta\big((d + \lambda)^{-1/2}\big). \tag{51}$$

*Proof.* At $t = 0$,

$$r(0) + 2(1+\lambda)b(0) = (d + 3\lambda + 2)\mu < 1, \qquad r(0) + 2c(0) = (d + \lambda + 2)\mu < 1,$$

and

$$r(0) + 2a(0) = (d + \lambda + 2)\mu < 3$$

for all large $d$ since $\rho_0 = o(1)$. Hence $g_{w_\star}(0) < 0$, $g_v(0) < 0$, and $g_\perp(0) < 0$.

As long as these three signs remain negative, (47) becomes $\dot{a} = 2\sqrt{a}, \dot{b} = 2\sqrt{b}, \dot{c} = 2\sqrt{c}$, so $\frac{d}{dt}\sqrt{a} = \frac{d}{dt}\sqrt{b} = \frac{d}{dt}\sqrt{c} = 1$ a.e. With $\sqrt{a}(0) = \sqrt{b}(0) = \sqrt{c}(0) = \sqrt{\mu}$, this gives $\alpha(t) = \beta(t) = \gamma(t) = \sqrt{\mu} + t$ on the maximal interval before the first switch (that corresponds to $T_1'$). By plugin the values of $a(t), b(t)$ and $c(t)$ we obtain

$$r(t) + 2(1+\lambda)b(t) = (d + 3\lambda + 2)(\sqrt{\mu} + t)^2.$$

Therefore the first time $r(t) + 2(1+\lambda)b(t)$ reaches 1 satisfies (51), which equals $T_1'$ by definition (50). $\qquad \square$

### C.2. Phase II: noise saturation and signal growth to $a = \rho$

**Lemma C.2.** *For all $t \geq 0$,*

$$(1 + \lambda)b(t) \leq \frac{1}{3}, \qquad c(t) \leq \frac{1}{d}. \tag{52}$$

*Consequently,*

$$b(t) \leq \frac{1}{3(1 + \lambda)}, \qquad (d - 2)c(t) \leq \frac{d - 2}{d} < 1. \tag{53}$$

*Proof. Spike trapping.* Suppose at some time $t$ we had $(1 + \lambda)b(t) > 1/3$. Then

$$r(t) + 2(1 + \lambda)b(t) = a(t) + 3(1 + \lambda)b(t) + (d - 2)c(t) > 1,$$

so $g_v(t) > 0$ by (48), and (47) gives $\dot{b}(t) = -2\sqrt{b(t)} < 0$. Thus $b$ cannot cross upward through the level $b = 1/(3(1 + \lambda))$, proving $(1 + \lambda)b(t) \leq 1/3$.

*Bulk trapping.* Suppose at some time $t$ we had $c(t) > 1/d$. Then

$$r(t) + 2c(t) = a(t) + (1 + \lambda)b(t) + d\,c(t) > 1,$$

so $g_\perp(t) > 0$ and (47) gives $\dot{c}(t) = -2\sqrt{c(t)} < 0$. Thus $c$ cannot cross upward through the level $c = 1/d$, proving $c(t) \leq 1/d$. The scaled bounds (53) follow immediately. $\square$

**Lemma C.3.** *Assume $\rho \in (0, 1/12)$ and $\mu \in (0, \rho)$. Then on the entire interval $\{t : a(t) \leq \rho\}$ one has $g_{w_\star}(t) < 0$ and hence*

$$\dot{a}(t) = 2\sqrt{a(t)}, \qquad equivalently \qquad \dot{\alpha}(t) = 1 \quad whenever\ a(t) \in (0, \rho].$$

*In particular,*

$$T_2' = \sqrt{\rho} - \sqrt{\mu} \leq \sqrt{\rho}. \tag{54}$$

*Moreover, $a(t) \geq \rho$ for all $t \geq T_2'$.*

*Proof.* Using Lemma C.2 we have for all $t$

$$(1 + \lambda)b(t) + (d - 2)c(t) \leq \frac{1}{3} + 1 = \frac{4}{3}.$$

Hence, whenever $a(t) \leq \rho$,

$$r(t) + 2a(t) = 3a(t) + (1 + \lambda)b(t) + (d - 2)c(t) \leq 3\rho + \frac{4}{3}.$$

Since $\rho < 1/12$, we have $3\rho + 4/3 < 3$, and therefore $g_{w_\star}(t) = 4(r(t) + 2a(t) - 3) < 0$. By (47), this implies $\dot{a}(t) = 2\sqrt{a(t)}$ as long as $a(t) \leq \rho$, and hence $\dot{\alpha}(t) = 1$ on that region.

Starting from $a(0) = \mu$, integrating $\dot{\alpha} = 1$ until $\alpha = \sqrt{\rho}$ yields $T_2' = \sqrt{\rho} - \sqrt{\mu}$, proving (54). Finally, observe that $g_{w_\star}(t) > 0$ implies $r(t) + 2a(t) > 3$, which in turn implies $3a(t) > 3 - ((1 + \lambda)b(t) + (d - 2)c(t)) \geq 3 - 4/3 = 5/3$ and hence $a(t) > 5/9$. Therefore, for all $t$ such that $a(t) \leq 5/9$, we must have $g_{w_\star}(t) \leq 0$ and thus $\dot{a}(t) \geq 0$. Since $\rho < 1/12 < 5/9$, once $a$ reaches $\rho$ it cannot decrease below $\rho$ afterwards. $\square$

**Proposition C.4** (Alignment after reaching $a = \rho$). *Fix $\rho \in (0, 1/12)$ and let $T_2'$ be as in (50). Then for all $t \geq T_2'$,*

$$a(t) \geq \rho, \qquad b(t) \leq \frac{1}{3(1 + \lambda)}, \qquad (d - 2)c(t)^2 \leq \frac{1}{d - 2}.$$

*Consequently,*

$$1 - \mathrm{Align}(t) \leq \frac{1}{2\rho^2}\left(\frac{1}{9(1 + \lambda)^2} + \frac{1}{d - 2}\right), \qquad t \geq T_2'. \tag{55}$$

*In particular, if $d \to \infty$ and $\lambda(d) \to \infty$, then $\sup_{t \geq T_2'}(1 - \mathrm{Align}(t)) \to 0$.*

*Proof.* The lower bound $a(t) \geq \rho$ for $t \geq T_2'$ is Lemma C.3. The bound $b(t) \leq 1/(3(1+\lambda))$ and $c(t) \leq 1/d$ are Lemma C.2, and $c(t) \leq 1/d$ implies

$$(d-2)c(t)^2 \leq (d-2) \cdot \frac{1}{d^2} \leq \frac{1}{d-2}.$$

Now write

$$\text{Align}(t) = \frac{a(t)}{\sqrt{a(t)^2 + b(t)^2 + (d-2)c(t)^2}} = \frac{1}{\sqrt{1+X(t)}}, \qquad X(t) := \frac{b(t)^2 + (d-2)c(t)^2}{a(t)^2}.$$

Using $1 - (1+x)^{-1/2} \leq x/2$ for $x \geq 0$ and $a(t) \geq \rho$ gives

$$1 - \text{Align}(t) \leq \frac{X(t)}{2} \leq \frac{1}{2\rho^2}\big(b(t)^2 + (d-2)c(t)^2\big).$$

With $b(t) \leq 1/(3(1+\lambda))$ and $(d-2)c(t)^2 \leq 1/(d-2)$ we obtain (55). The limit follows since $(1+\lambda)^{-2} \to 0$ and $(d-2)^{-1} \to 0$. □

## D. Discrete-time population dynamics of GD on $\mathcal{M}$

We analyze the population dynamics of GD restricted to the invariant manifold $\mathcal{M}$ in the noiseless population setting ($\sigma = 0$). First we summarize our main results from which result Theorem 4.3.

**Proposition D.1.** *Consider the discrete-time population GD dynamics on $\mathcal{M}$ with the on-manifold initialization of Section 2, and choose a learning rate $\eta \leq 1/(16(1+\lambda))$. Then, for $d$ large enough, the following holds.*

**Stage I (spike-driven growth, $k \leq T_1$).** *There exists an intermediate stopping time $T_{1a} < T_1$ such that the following behavior holds.*

*For $k \leq T_{1a}$, while the total mass $r_k$ remains small, all components grow geometrically:*

$$a_{k+1} = (1 + 24\eta + o(1))\, a_k,$$
$$b_{k+1} = \big(1 + 8\eta(1+\lambda) + o(1)\big)\, b_k,$$
$$c_{k+1} = (1 + 8\eta + o(1))\, c_k.$$

*At time $T_{1a}$, the total mass becomes of constant order, while $a_k + (d-2)c_k = o(1)$.*

*For $T_{1a} < k \leq T_1$, the spike component stops increasing, whereas the signal and bulk components remain essentially unchanged. Overall, Stage I lasts $T_1 = O((\eta\lambda)^{-1} \log d)$ iterations.*

**Stage II (signal amplification, $T_1 < k \leq T_2$).** *After spike saturation, the signal component grows geometrically at rate $a_{k+1} \simeq (1 + O(\eta))a_k$.*

*There exists a time $T_{2a}$ with $T_1 < T_{2a} < T_2$ such that for $T_1 < k \leq T_{2a}$ the bulk component continues to increase until the total mass approaches one. For $T_{2a} < k \leq T_2$, the bulk stabilizes, the spike component decays, and the signal continues to grow until it reaches constant order. In particular, after $T_2 = O(\eta^{-1} \log d)$ iterations, one has*

$$a_k = \Theta(1), \qquad b_k^2 + (d-2)c_k^2 = o(1), \qquad r_k = 1 - o(1).$$

As a corollary, we obtain the following result on the alignment of GD.

**Corollary D.2.** *At time $T_1$, the alignment satisfies*

$$\text{Align}(T_1) \lesssim \min\{\rho_0\lambda/d,\, d^{-1/2}\}.$$

*In particular, if $\lambda \ll \sqrt{d}$ then $\text{Align}(T_1) = o(d^{-1/2})$, whereas if $\lambda \gtrsim \sqrt{d}$ then $\text{Align}(T_1) \lesssim d^{-1/2}$.*

### D.1. Discrete reduced GD recursion on $\mathcal{M}$

We consider the explicit gradient descent update with learning rate $\eta > 0$,

$$W_{k+1} = W_k - \eta \, \nabla_W \mathcal{L}(W_k), \qquad M_k = W_k W_k^\top.$$

Recall that $\nabla_W \mathcal{L}(W) = G(M)W$ with a symmetric matrix $G(M)$. Hence

$$W_{k+1} = (I - \eta G_k)W_k, \qquad G_k := G(M_k),$$

and therefore

$$M_{k+1} = W_{k+1}W_{k+1}^\top = (I - \eta G_k)\, M_k \,(I - \eta G_k). \tag{56}$$

On the invariant manifold $\mathcal{M}$, $M_k$ commutes with $G_k$ (Lemma 3.1), so (56) simplifies to

$$M_{k+1} = (I - \eta G_k)^2 M_k,$$

i.e. each coordinate in the $\mathcal{M}$-eigendecomposition is multiplied by a squared scalar factor. Writing

$$M_k = a_k \, w_\star w_\star^\top + b_k \, vv^\top + c_k \, P_\perp, \qquad r_k := a_k + (1+\lambda)b_k + (d-2)c_k,$$

we obtain the following recursion

$$
\begin{aligned}
a_{k+1} &= a_k \Big(1 + 4\eta\,(3 - 2a_k - r_k)\Big)^2, \\
b_{k+1} &= b_k \Big(1 + 4\eta(1+\lambda)\big((1-r_k) - 2(1+\lambda)b_k\big)\Big)^2, \\
c_{k+1} &= c_k \Big(1 + 4\eta\big((1-r_k) - 2c_k\big)\Big)^2, \\
r_k &= a_k + (1+\lambda)b_k + (d-2)c_k.
\end{aligned}
\tag{57}
$$

Equivalently, in terms of the spike/bulk masses $(a_k, B_k, C_k)$ where

$$B_k := (1+\lambda)b_k, \qquad C_k := (d-2)c_k, \qquad r_k = a_k + B_k + C_k,$$

define

$$K_k := \frac{1 - a_k - C_k}{3} \qquad \Longleftrightarrow \qquad K_k - B_k = \frac{(1 - r_k) - 2B_k}{3}.$$

Then (57) becomes

$$
\begin{aligned}
a_{k+1} &= a_k \Big(1 + 4\eta\,(3 - 2a_k - r_k)\Big)^2, \\
K_k &= \frac{1 - a_k - C_k}{3}, \\
r_k &= a_k + B_k + C_k, \\
B_{k+1} &= B_k \Big(1 + 12\eta(1+\lambda)\,(K_k - B_k)\Big)^2, \\
C_{k+1} &= C_k \Big(1 + 4\eta(1 - r_k) - \frac{8\eta}{d-2}C_k\Big)^2.
\end{aligned}
\tag{58}
$$

**Discrete turning condition.** From (58),

$$\frac{B_{k+1}}{B_k} = \Big(1 + 12\eta(1+\lambda)(K_k - B_k)\Big)^2.$$

Under Assumption C and the invariant region from Lemma D.4 (generalization of the barrier Lemma for GF) we have $K_k \in [-1/3, 1/3]$ and $B_k \in [0, 1/3]$, hence $|K_k - B_k| \le 2/3$ and so

$$\left|12\eta(1+\lambda)(K_k - B_k)\right| \le 8\eta(1+\lambda) \le \frac{1}{2}.$$

In particular the bracket is positive and lies in $[1/2, 3/2]$, and therefore

$$B_{k+1} \le B_k \quad \Longleftrightarrow \quad K_k - B_k \le 0 \quad \Longleftrightarrow \quad B_k \ge K_k \quad \Longleftrightarrow \quad b_{k+1} \le b_k. \tag{59}$$

So the continuous turning condition $\dot{b} \le 0$ is replaced by the (equivalent) discrete condition $b_{k+1} \le b_k$.

## D.2. Learning-rate scaling

The recursion (58) still contains a fast component: $B_k$ evolves on the $\eta(1 + \lambda)$ scale. Indeed,

$$\frac{B_{k+1}}{B_k} = \left(1 + 12\eta(1 + \lambda)(K_k - B_k)\right)^2,$$

so the natural stability scale is $\eta(1 + \lambda) = O(1)$ (i.e. $\eta = O(1/\lambda)$).

**Assumption C** (Learning rate)**.** We assume

$$0 < \eta \leq \frac{1}{16(1 + \lambda)}. \tag{60}$$

*Remark* D.3. The bound (60) ensures (on the forward-invariant region of Lemma D.4) that the key multiplicative factor

$$1 + 12\eta(1 + \lambda)(K_k - B_k)$$

stays in $[1/2, 3/2]$. This prevents "overshoot" (where the bracket would cross $-1$ and squaring would reverse the intended monotonicity) and makes the turning condition (59) exactly equivalent to $B_k \geq K_k$.

## D.3. Uniform discrete barriers

**Lemma D.4.** *Assume $d \geq 4$ and the learning rate satisfies*

$$\eta \leq \min\left\{\frac{1}{24}, \frac{1}{16(1 + \lambda)}\right\}. \tag{61}$$

*Assume further that the initialization satisfies*

$$(a_0, B_0, C_0) \in \mathcal{R}, \qquad \mathcal{R} := [0, 1] \times \left[0, \frac{1}{3}\right] \times [0, 1].$$

*Then for all $k \geq 0$,*

$$0 \leq a_k \leq 1, \qquad 0 \leq B_k \leq \frac{1}{3}, \qquad 0 \leq C_k \leq 1, \qquad \text{and hence} \qquad 0 \leq r_k := a_k + B_k + C_k \leq \frac{7}{3}.$$

*Proof.* We show that $\mathcal{R}$ is forward invariant for the map $(a_k, B_k, C_k) \mapsto (a_{k+1}, B_{k+1}, C_{k+1})$ defined by (58). Fix $k \geq 0$ and assume $(a_k, B_k, C_k) \in \mathcal{R}$. Write $r_k = a_k + B_k + C_k$ and $K_k = (1 - a_k - C_k)/3$. On $\mathcal{R}$ we have $0 \leq r_k \leq 7/3$ and $K_k \leq 1/3$.

**Step 1: Positivity.** All three updates in (58) are of the form $x_{k+1} = x_k(\cdots)^2$, hence $a_{k+1}, B_{k+1}, C_{k+1} \geq 0$ whenever $a_k, B_k, C_k \geq 0$.

**Step 2: Upper barrier $a_k \leq 1$.** Using $r_k \geq a_k$,

$$3 - 2a_k - r_k \leq 3 - 3a_k = 3(1 - a_k),$$

so

$$a_{k+1} = a_k\left(1 + 4\eta(3 - 2a_k - r_k)\right)^2 \leq a_k\left(1 + 12\eta(1 - a_k)\right)^2.$$

Let $\alpha := 12\eta \leq 1/2$ (since $\eta \leq 1/24$) and define $f_\alpha(x) := x(1 + \alpha(1 - x))^2$ on $[0, 1]$. A direct derivative computation gives

$$f'_\alpha(x) = (\alpha x - \alpha - 1)(3\alpha x - \alpha - 1) \geq 0 \quad \text{on } [0, 1] \quad (\text{for } \alpha \leq 1/2),$$

hence $f_\alpha$ is increasing on $[0, 1]$ and $\max_{[0,1]} f_\alpha = f_\alpha(1) = 1$. Therefore $a_k \in [0, 1] \Rightarrow a_{k+1} \leq 1$.

**Step 3: Upper barrier $C_k \leq 1$.** Using $r_k \geq C_k$ and dropping the negative term,

$$1 + 4\eta(1 - r_k) - \frac{8\eta}{d - 2}C_k \leq 1 + 4\eta(1 - C_k).$$

Thus

$$C_{k+1} \leq C_k\Big(1 + 4\eta(1 - C_k)\Big)^2 = f_\beta(C_k), \qquad \beta := 4\eta \leq \frac{1}{6} < \frac{1}{2}.$$

By the same monotonicity argument as Step 2 (with $\beta \leq 1/2$), $f_\beta([0,1]) \subseteq [0,1]$, so $C_k \leq 1 \Rightarrow C_{k+1} \leq 1$.

**Step 4: Upper barrier $B_k \leq 1/3$.** Since $K_k \leq 1/3$, we have $K_k - B_k \leq \frac{1}{3} - B_k$ and hence

$$B_{k+1} = B_k\Big(1 + 12\eta(1 + \lambda)(K_k - B_k)\Big)^2 \leq B_k\Big(1 + 12\eta(1 + \lambda)\Big(\frac{1}{3} - B_k\Big)\Big)^2.$$

Let $y_k := 3B_k \in [0,1]$ and $\gamma := 4\eta(1 + \lambda) \leq 1/4 < 1/2$. Then the above becomes

$$y_{k+1} \leq y_k\Big(1 + \gamma(1 - y_k)\Big)^2 = f_\gamma(y_k),$$

and as in Step 2, $f_\gamma([0,1]) \subseteq [0,1]$ for $\gamma \leq 1/2$. Hence $y_k \leq 1 \Rightarrow y_{k+1} \leq 1$, i.e. $B_{k+1} \leq 1/3$.

Thus $(a_{k+1}, B_{k+1}, C_{k+1}) \in \mathcal{R}$, so $\mathcal{R}$ is forward invariant. Finally, $0 \leq r_k \leq 7/3$ follows from $(a_k, B_k, C_k) \in \mathcal{R}$. □

### D.4. Phase Ia: spike-driven escape to $r \geq \rho$

**Lemma D.5.** *Under the assumptions of Theorem D.1, the following holds.*

*(i) Geometric growth before escape. For all $k < N_1$,*

$$a_0\Big(1 + 12\eta(1 - \rho)\Big)^{2k} \leq a_k \leq a_0\Big(1 + 12\eta\Big)^{2k}, \tag{62}$$

$$b_0\Big(1 + 4\eta(1 + \lambda)(1 - 3\rho)\Big)^{2k} \leq b_k \leq b_0\Big(1 + 4\eta(1 + \lambda)\Big)^{2k}, \tag{63}$$

$$c_0\Big(1 + 4\eta(1 - \rho) - 8\eta\rho/(d - 2)\Big)^{2k} \leq c_k \leq c_0\Big(1 + 4\eta\Big)^{2k}. \tag{64}$$

*(ii) Escape time scale. The escape index satisfies*

$$N_1 = O\Big(\frac{1}{\eta\lambda}\log d\Big).$$

*(iii) Smallness of signal and bulk at escape. At the escape time,*

$$a_{N_1} \lesssim \frac{\rho_0}{d + \lambda}, \qquad C_{N_1} = (d - 2)c_{N_1} \lesssim \rho_0,$$

*while simultaneously*

$$a_{N_1} \geq a_0 = \mu, \qquad C_{N_1} \geq C_0 = (d - 2)\mu = \Theta(\rho_0).$$

*(iv) Spike dominance at escape. One has*

$$r_{N_1} = \Theta(\rho), \qquad B_{N_1} = r_{N_1} - a_{N_1} - C_{N_1} = \Theta(\rho).$$

*Proof.* By definition of $N_1$, we have

$$r_k = a_k + B_k + C_k \leq \rho \qquad \text{for all } k < N_1.$$

**Step 1: Geometric growth before escape.** For $k < N_1$, we have $a_k, B_k, C_k \leq r_k \leq \rho$. Apply (58).

*Signal component $a_k$.* From (58),

$$\frac{a_{k+1}}{a_k} = \Big(1 + 4\eta(3 - 2a_k - r_k)\Big)^2.$$

Using $a_k, r_k \leq \rho$ gives

$$3(1 - \rho) \; \leq \; 3 - 2a_k - r_k \; \leq \; 3,$$

hence

$$\left(1 + 12\eta(1 - \rho)\right)^2 \; \leq \; \frac{a_{k+1}}{a_k} \; \leq \; \left(1 + 12\eta\right)^2,$$

which yields the bounds for $a_k$.

*Spike component $b_k$.* Using $B_k = (1 + \lambda)b_k$, the $B$-update in (58) gives

$$\frac{B_{k+1}}{B_k} = \left(1 + 12\eta(1 + \lambda)(K_k - B_k)\right)^2 = \left(1 + 4\eta(1 + \lambda)\left((1 - r_k) - 2B_k\right)\right)^2.$$

For $k < N_1$, $r_k \leq \rho$ and $B_k \leq \rho$, so

$$1 - 3\rho \; \leq \; (1 - r_k) - 2B_k \; \leq \; 1,$$

hence

$$\left(1 + 4\eta(1 + \lambda)(1 - 3\rho)\right)^2 \; \leq \; \frac{B_{k+1}}{B_k} \; \leq \; \left(1 + 4\eta(1 + \lambda)\right)^2,$$

and the same bounds hold for $b_k$.

*Bulk component $c_k$.* From (57),

$$\frac{c_{k+1}}{c_k} = \left(1 + 4\eta(1 - r_k) - 8\eta c_k\right)^2.$$

Using $r_k \leq \rho$ and $c_k \leq r_k/(d - 2) \leq \rho/(d - 2)$ yields

$$1 + 4\eta(1 - \rho) - 8\eta\frac{\rho}{d - 2} \; \leq \; 1 + 4\eta(1 - r_k) - 8\eta c_k \; \leq \; 1 + 4\eta,$$

which gives the stated bounds for $c_k$.

**Step 2: Escape time scale.** From Step 1, for $k < N_1$,

$$B_k \geq B_0\left(1 + 4\eta(1 + \lambda)(1 - 3\rho)\right)^{2k}.$$

Since $r_{N_1 - 1} \leq \rho$, we must have $B_{N_1 - 1} \leq \rho$, hence

$$B_0\left(1 + 4\eta(1 + \lambda)(1 - 3\rho)\right)^{2(N_1 - 1)} \leq \rho.$$

Taking logs and using $\log(1 + x) \gtrsim x$ for $x \leq 1/4$ (here $4\eta(1 + \lambda) \leq 1/4$), we obtain

$$N_1 \; \lesssim \; \frac{1}{\eta(1 + \lambda)} \log\left(\frac{\rho}{B_0}\right).$$

Recalling $B_0 = (1 + \lambda)\mu \asymp (1 + \lambda)\rho_0/(d + \lambda)$, we have $\log(\rho/B_0) = \Theta(\log d)$, which proves (ii).

**Step 3: Smallness of $a$ and $C$ at escape.** From Step 1,

$$a_{N_1 - 1} \leq a_0(1 + 12\eta)^{2(N_1 - 1)}, \qquad C_{N_1 - 1} \leq C_0(1 + 4\eta)^{2(N_1 - 1)}.$$

By Step 2, $\eta N_1 = O((\log d)/\lambda)$, and since $\lambda \gtrsim \log d$,

$$(1 + O(\eta))^{N_1} = O(1).$$

Thus

$$a_{N_1 - 1} \lesssim a_0 \asymp \frac{\rho_0}{d + \lambda}, \qquad C_{N_1 - 1} \lesssim C_0 \asymp \rho_0.$$

A one-step update yields the same bounds at $k = N_1$, proving the upper bounds in (iii).

For the lower bounds, note that for $k < N_1$ we have $r_k \leq \rho$ and $a_k \leq \rho$, hence $3 - 2a_k - r_k \geq 3(1-\rho) > 0$, so $a_{k+1} \geq a_k$ and $a_{N_1} \geq a_0$. Similarly, for $k < N_1$,

$$1 + 4\eta(1 - r_k) - \frac{8\eta}{d-2}C_k \;\geq\; 1 + 4\eta(1-\rho) - \frac{8\eta}{d-2}\rho \;>\; 1,$$

for $d$ large enough, so $C_{k+1} \geq C_k$ and $C_{N_1} \geq C_0$.

**Step 4: Spike dominance at escape.** By definition $r_{N_1} > \rho$. On the other hand, for $k < N_1$ we have $B_k \leq \rho$ and $K_k \leq 1/3$, so

$$\frac{B_{k+1}}{B_k} \leq \left(1 + 12\eta(1+\lambda) \cdot \frac{1}{3}\right)^2 \leq \left(1 + \frac{1}{4}\right)^2 < 2,$$

and similarly $\frac{a_{k+1}}{a_k} \leq (1 + 12\eta)^2 < 2$ and $\frac{C_{k+1}}{C_k} \leq (1 + 4\eta)^2 < 2$ (using $\eta \leq 1/24$). Therefore $r_{k+1} \leq 2r_k$ for $k < N_1$, and hence $r_{N_1} \leq 2\rho$. Thus $r_{N_1} = \Theta(\rho)$.

Finally, since $a_{N_1}, C_{N_1} = o(\rho)$ by Step 3, we conclude

$$B_{N_1} = r_{N_1} - a_{N_1} - C_{N_1} = \Theta(\rho),$$

which proves (iv). □

## D.5. Phase Ib: fast spike saturation to the turning index

Define

$$K_k := \frac{1 - a_k - C_k}{3}, \qquad E_k := K_k - B_k.$$

Phase Ib corresponds to indices $k \in [N_1, N_2]$, where $B_k$ increases and reaches the turning condition $b_{k+1} \leq b_k$.

**Proposition D.6.** *Under the assumptions of Theorem D.1, the following holds.*

1. *Phase duration.*
$$N_2 - N_1 = O\left(\frac{1}{\eta\lambda} \log \frac{1}{\rho_0}\right) = O\left(\frac{1}{\eta\lambda} \log\log d\right). \tag{65}$$

2. *Stability of slow components.*
$$a_{N_2} = a_{N_1}(1 + o(1)), \qquad C_{N_2} = C_{N_1}(1 + o(1)). \tag{66}$$

3. *Spike saturation level.*
$$B_{N_2} = \frac{1}{3}(1 + o(1)), \qquad r_{N_2} = \frac{1}{3}(1 + o(1)). \tag{67}$$

*Proof.* By definition of $N_2$, we have $E_k > 0$ for all $k \in [N_1, N_2)$ and $E_{N_2} \leq 0$.

**Step 1: monotonicity of $B_k$ before turning and a uniform lower bound.** From (58),

$$\frac{B_{k+1}}{B_k} = \left(1 + 12\eta(1+\lambda)E_k\right)^2.$$

Thus for $k < N_2$ we have $E_k > 0$ and hence $B_{k+1} > B_k$, i.e. $B_k$ is strictly increasing on $k \in [N_1, N_2]$. By Phase Ia, $B_{N_1} = \Theta(\rho)$, hence for $d$ large enough,
$$B_k \geq \rho/2, \qquad k \in [N_1, N_2]. \tag{68}$$

Also $B_k \leq 1/3$ for all $k$ by Lemma D.4.

**Step 2: bootstrap region $r_k \leq 1/2$ and monotonicity of $a_k, C_k, K_k$.** Define

$$\bar{N} := \sup\left\{n \in [N_1, N_2] : r_j \leq \frac{1}{2} \text{ for all } j \in [N_1, n]\right\}.$$

This is well-defined since $r_{N_1} = \Theta(\rho) < 1/2$ for $\rho < 1/12$. On the interval $j \in [N_1, \bar{N}]$ we have $a_j \leq r_j \leq 1/2$ and $1 - r_j \geq 1/2$.

From the $a$-update in (58), for $j \in [N_1, \bar{N}]$,

$$\frac{a_{j+1}}{a_j} = \left(1 + 4\eta(3 - 2a_j - r_j)\right)^2 \geq \left(1 + 4\eta \cdot \frac{3}{2}\right)^2 > 1,$$

so $a_j$ is nondecreasing on $[N_1, \bar{N}]$.

For $C_j$, using $c_j = C_j/(d-2) \leq r_j/(d-2) \leq \frac{1}{2(d-2)}$,

$$1 + 4\eta(1 - r_j) - \frac{8\eta}{d-2}C_j \geq 1 + 2\eta - \frac{4\eta}{d-2} \geq 1 + \eta \quad \text{(for } d \text{ large enough)},$$

hence $C_{j+1} \geq C_j$ and $C_j$ is nondecreasing on $[N_1, \bar{N}]$. Therefore $K_j = (1 - a_j - C_j)/3$ is nonincreasing on $[N_1, \bar{N}]$.

Finally, by Phase Ia we have $C_{N_1} = \Theta(\rho_0)$, hence

$$C_j \geq C_{N_1} \geq c\,\rho_0, \qquad j \in [N_1, \bar{N}], \tag{69}$$

for some absolute $c > 0$.

**Step 3: exact recursion for $E_k$ and forcing to cross $0$ (correct).** Using $E_k = K_k - B_k$ and the $B$-update in (58),

$$B_{k+1} = B_k\left(1 + 12\eta(1 + \lambda)E_k\right)^2,$$

we obtain the exact identity

$$E_{k+1} = (1 - 24\eta(1 + \lambda)B_k)E_k - (K_k - K_{k+1}) - 144\eta^2(1 + \lambda)^2 B_k E_k^2, \qquad k \geq 0, \tag{70}$$

since $K_k - K_{k+1} = \frac{(a_{k+1} - a_k) + (C_{k+1} - C_k)}{3}$.

On $k \in [N_1, \bar{N}]$ we have $K_{k+1} \leq K_k$. Moreover, from Step 2,

$$K_k - K_{k+1} = \frac{(a_{k+1} - a_k) + (C_{k+1} - C_k)}{3} \geq \frac{C_{k+1} - C_k}{3}.$$

Using $C_{k+1} = C_k(1 + u_k)^2$ with $u_k := 4\eta(1 - r_k) - \frac{8\eta}{d-2}C_k \geq \eta$ on $[N_1, \bar{N}]$ for $d$ large, we get $C_{k+1} - C_k = C_k(2u_k + u_k^2) \geq 2\eta C_k$. Hence

$$K_k - K_{k+1} \geq \frac{2\eta}{3}C_k \geq c'\eta\rho_0, \qquad k \in [N_1, \bar{N}],$$

using (69), for some absolute $c' > 0$.

Also by (68),

$$24\eta(1 + \lambda)B_k \geq 12\rho\,\eta(1 + \lambda) =: \alpha, \qquad k \in [N_1, \bar{N}],$$

and $\alpha \in (0, 1)$ since $\eta \leq 1/(16(1 + \lambda))$ and $\rho < 1/12$. Finally, the last term in (70) is *negative*, so for all $k \in [N_1, \bar{N}]$ with $k < N_2$ (so $E_k > 0$),

$$E_{k+1} \leq (1 - \alpha)E_k - \beta, \qquad \beta := c'\eta\rho_0. \tag{71}$$

The remainder of the contraction/iteration argument is unchanged and yields (65).

**Step 4: closing the bootstrap ($\bar{N} = N_2$).** Let $m_\star$ be the smallest integer such that $E_{N_1 + m_\star} \leq 0$ from Step 3. We show $r_k \leq 1/2$ for all $k \in [N_1, N_1 + m_\star]$. Using crude upper bounds from (58):

$$a_{k+1} \leq a_k(1 + 12\eta)^2, \qquad C_{k+1} \leq C_k(1 + 4\eta)^2,$$

we obtain for $k \leq N_1 + m_\star$,

$$a_k + C_k \leq (a_{N_1} + C_{N_1})(1 + 12\eta)^{2m_\star}.$$

Since $a_{N_1} + C_{N_1} \lesssim \rho_0$ (Phase Ia) and $\eta m_\star = O(\log((1+\lambda)/\rho_0)/\lambda) = o(1)$ under $\lambda \gtrsim \log d$, we have $(1 + 12\eta)^{2m_\star} = 1 + o(1)$ and thus $a_k + C_k = o(1)$ uniformly on this range. Together with $B_k \leq 1/3$, this yields $r_k \leq 1/3 + o(1) < 1/2$ for all $k \in [N_1, N_1 + m_\star]$. Hence $\bar{N} \geq N_1 + m_\star \geq N_2$, so in fact $\bar{N} = N_2$.

**Step 5: stability of slow components.** Let $m := N_2 - N_1$. Then $m = O((\eta\lambda)^{-1} \log(1/\rho_0))$ and

$$\frac{a_{N_2}}{a_{N_1}} \leq (1 + 12\eta)^{2m} = \exp(O(\eta m)) = \exp\left(O\left(\frac{\log(1/\rho_0)}{\lambda}\right)\right) = 1 + o(1),$$

and similarly $C_{N_2} = C_{N_1}(1 + o(1))$. This proves (66).

**Step 6: turning location.** By definition of $N_2$, $B_{N_2} \geq K_{N_2} = (1 - a_{N_2} - C_{N_2})/3$, hence

$$B_{N_2} \geq \frac{1}{3} - \frac{a_{N_2} + C_{N_2}}{3} = \frac{1}{3}(1 + o(1)),$$

since $a_{N_2} + C_{N_2} = o(1)$ by Step 5 and Phase Ia. On the other hand $B_{N_2} \leq 1/3$ by Lemma D.4. Thus $B_{N_2} = \frac{1}{3}(1 + o(1))$, and then $r_{N_2} = a_{N_2} + B_{N_2} + C_{N_2} = \frac{1}{3}(1 + o(1))$, proving (67). $\square$

### D.6. Phase IIa: bulk inflation to $r \geq 1 - \varepsilon$

**Proposition D.7.** *Under the assumptions of Theorem D.1, the following holds.*

1. **Monotonicity.** *For every $k \in [N_2, N_3)$ (so $r_k \leq 1 - \varepsilon$) one has*

$$a_{k+1} \geq a_k, \qquad C_{k+1} \geq C_k, \qquad K_{k+1} \leq K_k,$$

   *and moreover $B_k \geq K_k$ and $B_{k+1} \leq B_k$.*

2. **Uniform lower bound on $K$ and $B$.** *For every $k \in [N_2, N_3)$,*

$$K_k \geq \frac{\varepsilon}{2}, \qquad \text{and hence} \qquad B_k \geq \frac{\varepsilon}{2}. \tag{72}$$

3. **Tracking of $D_k := B_k - K_k$.** *Let $D_k := B_k - K_k \geq 0$. Then for every $k \in [N_2, N_3)$,*

$$0 \leq D_k \leq (1 - \alpha)^{k-N_2} D_{N_2} + C_\star \frac{\eta}{\alpha}, \qquad \alpha := 6\varepsilon\eta(1 + \lambda), \tag{73}$$

   *for an absolute constant $C_\star > 0$. In particular, using $D_{N_2} = O(a_{N_2} + C_{N_2}) = O(\rho_0)$ from Phase Ib,*

$$\sup_{k \in [N_2, N_3)} D_k = O(\rho_0) + O\left(\frac{1}{\varepsilon(1 + \lambda)}\right) = o(1).$$

4. **Phase duration.**

$$N_3 - N_2 = O\left(\frac{1}{\eta} \log \frac{1}{\varepsilon\rho_0}\right) = O\left(\frac{1}{\eta} \log\log d\right) \quad (\varepsilon \text{ fixed}). \tag{74}$$

5. **Exit values of the summary statistics.** *At $k = N_3$,*

$$a_{N_3} = d^{-1+o(1)}, \qquad C_{N_3} = 1 - \Theta(\varepsilon), \qquad B_{N_3} = \Theta(\varepsilon), \qquad r_{N_3} = 1 - \Theta(\varepsilon).$$

*Proof.* Fix $k \in [N_2, N_3)$, so $r_k \leq 1 - \varepsilon$.

**Step 1: monotonicity of $a_k$ and $C_k$, hence $K_k$ decreases.** Since $a_k \leq 1$ and $r_k \leq 1 - \varepsilon$,

$$3 - 2a_k - r_k \geq 3 - 2 - (1 - \varepsilon) = \varepsilon \geq 0,$$

so (58) gives $a_{k+1} \geq a_k$.

For $C_k$, use the lower bound (valid for all $u$):
$$(1 + u)^2 \geq 1 + 2u.$$

With $u := 4\eta(1 - r_k) - \frac{8\eta}{d-2}C_k$, we get from (58)

$$C_{k+1} \geq C_k\left(1 + 8\eta(1 - r_k) - \frac{16\eta}{d - 2}C_k\right),$$

hence

$$C_{k+1} - C_k \geq 8\eta(1 - r_k)C_k - \frac{16\eta}{d-2}C_k^2 \geq 8\eta\varepsilon\, C_k - \frac{16\eta}{d-2}C_k.$$

For $d$ large enough (e.g. $d \geq 2 + 2/\varepsilon$) the coefficient is nonnegative, hence $C_{k+1} \geq C_k$. Therefore

$$K_{k+1} - K_k = -\frac{(a_{k+1} - a_k) + (C_{k+1} - C_k)}{3} \leq 0,$$

so $K_k$ is nonincreasing on $[N_2, N_3)$.

**Step 2: $B \geq K$ persists and $B$ is nonincreasing.** At time $N_2$ we have $B_{N_2} \geq K_{N_2}$ by definition of $N_2$. Define $D_k := B_k - K_k$. From the $B$-update in (58) and $K_k - B_k = -D_k$,

$$\frac{B_{k+1}}{B_k} = \left(1 - 12\eta(1 + \lambda)D_k\right)^2.$$

By Lemma D.4 we have $B_k \leq 1/3$, $K_k \in [-1/3, 1/3]$, hence $D_k \leq 2/3$. Together with $\eta \leq 1/(16(1 + \lambda))$ this implies $12\eta(1 + \lambda)D_k \leq 1/2$, so the bracket is positive and at most 1, hence $B_{k+1} \leq B_k$. Moreover, one can write the exact identity

$$D_{k+1} = B_{k+1} - K_{k+1} = (1 - 24\eta(1 + \lambda)B_k)D_k + (K_k - K_{k+1}) + 144\eta^2(1 + \lambda)^2 B_k D_k^2,$$

which shows $D_k \geq 0 \Rightarrow D_{k+1} \geq 0$ since $K_k - K_{k+1} \geq 0$ and the last term is nonnegative. By induction $D_k \geq 0$ for all $k \in [N_2, N_3)$, i.e. $B_k \geq K_k$.

**Step 3: lower bounds on $K_k$ and $B_k$.** Using $B_k \geq K_k$ and $r_k = a_k + B_k + C_k \leq 1 - \varepsilon$,

$$1 - \varepsilon \;\geq\; a_k + C_k + B_k \;\geq\; a_k + C_k + K_k = a_k + C_k + \frac{1 - a_k - C_k}{3} = \frac{1 + 2(a_k + C_k)}{3}.$$

Hence $1 - a_k - C_k \geq \frac{3}{2}\varepsilon$, so

$$K_k = \frac{1 - a_k - C_k}{3} \geq \frac{\varepsilon}{2}.$$

Since $B_k \geq K_k$, this proves (72).

**Step 4: tracking bound for $D_k = B_k - K_k$.** From Step 2, $x_k := 12\eta(1 + \lambda)D_k \in [0, 1/2]$ on this interval. For $x \in [0, 1]$ we have

$$(1 - x)^2 \leq 1 - x.$$

Therefore

$$B_{k+1} = B_k(1 - x_k)^2 \leq B_k(1 - x_k) = B_k\left(1 - 12\eta(1 + \lambda)D_k\right).$$

Subtracting $K_{k+1}$ and using $D_k = B_k - K_k$ gives the one-step inequality

$$D_{k+1} \leq (1 - 12\eta(1 + \lambda)B_k)D_k + (K_k - K_{k+1}).$$

By (72), $B_k \geq \varepsilon/2$, hence

$$12\eta(1 + \lambda)B_k \geq 6\varepsilon\,\eta(1 + \lambda) =: \alpha.$$

Also, using $\eta \leq 1/24$ and $a_k, C_k \leq 1$, one has the crude bound $K_k - K_{k+1} \leq C_\star \eta$ for an absolute constant $C_\star > 0$ (e.g. $C_\star = 40/3$ works). Thus

$$D_{k+1} \leq (1 - \alpha)D_k + C_\star \eta,$$

and iterating this affine contraction yields (73).

**Step 5: logistic lower comparison for $C_k$ and the bound on $N_3 - N_2$.**

Fix $k \in [N_2, N_3)$, so that by definition $r_k \leq 1 - \varepsilon$. Recall that the correct discrete update for $C_k$ on $\mathcal{M}$ is

$$C_{k+1} = C_k \left(1 + 4\eta(1 - r_k) - \frac{8\eta}{d-2}C_k\right)^2.$$

Using the inequality $(1 + u)^2 \geq 1 + 2u$ valid for all $u \geq -1$, we obtain

$$C_{k+1} - C_k \;\geq\; 8\eta(1 - r_k)C_k - \frac{16\eta}{d-2}C_k^2. \tag{75}$$

**Step 5.1: expressing $1 - r_k$ and isolating the main drift.** Recall that $B_k = K_k + D_k$ with $D_k := B_k - K_k \geq 0$, and

$$K_k = \frac{1 - a_k - C_k}{3}.$$

Therefore

$$r_k = a_k + B_k + C_k = a_k + C_k + K_k + D_k = \frac{1 + 2a_k + 2C_k}{3} + D_k,$$

and hence

$$1 - r_k = \frac{2}{3}(1 - a_k - C_k) - D_k. \tag{76}$$

Substituting (76) into (75) yields

$$C_{k+1} - C_k \;\geq\; \frac{16\eta}{3}(1 - a_k - C_k)C_k \;-\; 8\eta D_k C_k \;-\; \frac{16\eta}{d-2}C_k^2. \tag{77}$$

**Step 5.2: uniform lower bounds from Phase IIa.** From Step 3 of Phase IIa we have, for all $k \in [N_2, N_3)$,

$$1 - a_k - C_k \;\geq\; \frac{3}{2}\varepsilon. \tag{78}$$

From Step 4 of Phase IIa we also have the uniform tracking bound

$$\sup_{k \in [N_2, N_3)} D_k \;\leq\; \delta, \tag{79}$$

where $\delta > 0$ can be taken arbitrarily small by choosing $d$ large enough (since $D_k = O(\rho_0) + O((\varepsilon(1 + \lambda))^{-1}) = o(1)$).

Finally, since $0 \leq C_k \leq 1$ on the invariant region, we have

$$\frac{16}{d-2}C_k \leq \delta \qquad \text{for all } k \in [N_2, N_3), \tag{80}$$

again for $d$ large enough.

**Step 5.3: deriving a logistic lower bound.** Applying (78)–(80) to (77), we obtain

$$C_{k+1} - C_k \;\geq\; 8\varepsilon\,\eta\,C_k \;-\; 8\delta\,\eta\,C_k \;-\; \delta\,\eta\,C_k.$$

Choosing $\delta \leq \varepsilon/4$, this simplifies to

$$C_{k+1} - C_k \;\geq\; 4\varepsilon\,\eta\,C_k.$$

Moreover, since $1 - a_k - C_k \leq 1 - C_k$, we also have from (77) and (79) that

$$C_{k+1} - C_k \;\geq\; \frac{16\eta}{3}(1 - C_k)C_k \;-\; O(\delta)\eta\,C_k.$$

Combining the two bounds yields the uniform logistic inequality

$$C_{k+1} \;\geq\; C_k + \gamma\,\eta\,C_k(1 - C_k), \tag{81}$$

for some absolute constant $\gamma > 0$ (e.g. $\gamma = \varepsilon$), and for all $k \in [N_2, N_3)$.

**Step 5.4: odds–ratio growth and duration of Phase IIa.** Define the odds ratio $Z_k := \frac{C_k}{1 - C_k}$. From (81) one checks directly that

$$Z_{k+1} \ \geq \ (1 + \gamma\eta)\, Z_k.$$

Iterating,

$$Z_{N_2 + m} \geq Z_{N_2}(1 + \gamma\eta)^m.$$

From Phase Ib we have $C_{N_2} = \Theta(\rho_0)$, hence $Z_{N_2} = \Theta(\rho_0)$. To reach $r_k > 1 - \varepsilon$ it suffices that

$$C_k \geq 1 - \tfrac{3}{2}\varepsilon,$$

since then

$$r_k = \frac{1 + 2a_k + 2C_k}{3} + D_k \geq 1 - \varepsilon.$$

Equivalently, we require $Z_k \gtrsim 1/(\varepsilon)$. Therefore it suffices that

$$(1 + \gamma\eta)^m \ \gtrsim \ \frac{1}{\varepsilon\rho_0},$$

which yields

$$m = O\Big(\frac{1}{\eta} \log \frac{1}{\varepsilon\rho_0}\Big).$$

This proves (74).

**Step 6: exit location.** At $k = N_3$ we have $r_{N_3} > 1 - \varepsilon$ by definition of $N_3$, while for all $k \in [N_2, N_3)$ we have $r_k \leq 1 - \varepsilon$.

*Step 6.1: bound on $a_{N_3}$.* From Phase Ib we have $a_{N_2} \lesssim \rho_0/(d + \lambda)$. Moreover, from the $a$-update in the corrected recursion,

$$a_{k+1} = a_k\Big(1 + 4\eta(3 - 2a_k - r_k)\Big)^2 \leq a_k(1 + 12\eta)^2,$$

since $3 - 2a_k - r_k \leq 3$. Iterating for $m := N_3 - N_2$ steps gives

$$a_{N_3} \leq a_{N_2}(1 + 12\eta)^{2m} \leq a_{N_2} \exp\big(24\eta m\big).$$

By Step 5 we already proved $m = O\big(\eta^{-1} \log \frac{1}{\varepsilon\rho_0}\big)$, hence

$$a_{N_3} \leq \frac{\rho_0}{d + \lambda} \cdot \Big(\frac{1}{\varepsilon\rho_0}\Big)^{O(1)} = d^{-1 + o(1)},$$

since $\rho_0 = \log^{-c_0} d$ and $\varepsilon$ is fixed. In particular $a_{N_3} = o(1)$.

*Step 6.2: $C_{N_3} = 1 - \Theta(\varepsilon)$.* Recall the exact identity

$$r_k = a_k + B_k + C_k = a_k + C_k + K_k + D_k = \frac{1 + 2a_k + 2C_k}{3} + D_k, \qquad D_k := B_k - K_k \geq 0.$$

By Step 4 we have $\sup_{k \in [N_2, N_3)} D_k = o(1)$, and the same bound also holds at $k = N_3$ (because the recursion and barriers keep all quantities in a compact region and the estimate for $D_k$ is deterministic on that interval). Thus $D_{N_3} = o(1)$.

Since $r_{N_3} > 1 - \varepsilon$, we obtain

$$1 - \varepsilon < r_{N_3} = \frac{1 + 2a_{N_3} + 2C_{N_3}}{3} + D_{N_3} \leq \frac{1 + 2a_{N_3} + 2C_{N_3}}{3} + o(1),$$

hence

$$C_{N_3} \ \geq \ 1 - \frac{3}{2}\varepsilon - \frac{a_{N_3}}{1} - o(1) = 1 - \frac{3}{2}\varepsilon - o(1),$$

using $a_{N_3} = o(1)$ from Step 6.1.

Conversely, since $r_{N_3-1} \leq 1 - \varepsilon$ and $D_{N_3-1} \geq 0$,

$$1 - \varepsilon \geq r_{N_3-1} = \frac{1 + 2a_{N_3-1} + 2C_{N_3-1}}{3} + D_{N_3-1} \geq \frac{1 + 2C_{N_3-1}}{3},$$

which implies

$$C_{N_3-1} \leq 1 - \frac{3}{2}\varepsilon.$$

Since $C_k$ is nondecreasing on $[N_2, N_3)$ (Step 1 of Phase IIa), we have $C_{N_3-1} \leq C_{N_3} \leq 1$, and therefore

$$C_{N_3} = 1 - \Theta(\varepsilon).$$

*Step 6.3: $B_{N_3} = \Theta(\varepsilon)$ and $r_{N_3} = 1 - \Theta(\varepsilon)$.* From $r_{N_3} = a_{N_3} + B_{N_3} + C_{N_3}$ and Step 6.1–6.2,

$$B_{N_3} = r_{N_3} - a_{N_3} - C_{N_3} = \big(1 - \Theta(\varepsilon)\big) - o(1) - \big(1 - \Theta(\varepsilon)\big) = \Theta(\varepsilon).$$

Finally, since $C_{N_3} = 1 - \Theta(\varepsilon)$ and $a_{N_3} = o(1)$, we have

$$r_{N_3} = a_{N_3} + B_{N_3} + C_{N_3} = 1 - \Theta(\varepsilon),$$

which completes the exit characterization.

$\square$

### D.7. Phase IIb: signal amplification and alignment

We now analyze the post-inflation regime $k \geq N_3$, where the bulk is already saturated ($C_k = 1 - \Theta(\varepsilon)$) and the spike has turned ($B_k$ is nonincreasing). Recall the stopping index

$$N_4 := \inf\{k \geq N_3 : a_k \geq \delta\}.$$

**Proposition D.8.** *Under the assumptions of Theorem D.1, the following holds.*

1. ***Monotonicity and contraction of nuisance.*** *For all $k \in [N_3, N_4)$ we have*

$$a_{k+1} \geq a_k, \qquad B_{k+1} \leq B_k.$$

   *Moreover, the bulk deficit $1 - C_k$ contracts geometrically: there exists an absolute constant $c_0 > 0$ (depending only on $\varepsilon$) such that*

$$1 - C_{k+1} \leq (1 - c_0\eta)\,(1 - C_k), \qquad B_{k+1} \leq (1 - c_0\eta(1 + \lambda))\,B_k, \qquad k \in [N_3, N_4).$$

   *Consequently, $B_k$ decays geometrically on the $\eta^{-1}\lambda^{-1}$ scale and $1 - C_k$ decays geometrically on the $\eta^{-1}$ scale.*

2. ***Signal amplification time.*** *The hitting time satisfies*

$$N_4 - N_3 = O\Big(\frac{1}{\eta} \log \frac{\delta}{a_{N_3}}\Big) = O\Big(\frac{1}{\eta} \log d\Big).$$

3. ***Alignment becomes constant and then converges to $1$.*** *For all $k \geq N_4$,*

$$\mathrm{Align}(k) = \frac{a_k}{\sqrt{a_k^2 + b_k^2 + (d-2)c_k^2}} = 1 - o(1).$$

*Proof.* Fix $k \geq N_3$. We use the discrete recursion on $\mathcal{M}$:

$$a_{k+1} = a_k\Big(1 + 4\eta(3 - 2a_k - r_k)\Big)^2, \qquad B_{k+1} = B_k\Big(1 + 12\eta(1 + \lambda)(K_k - B_k)\Big)^2,$$

$$C_{k+1} = C_k\left(1 + 4\eta(1 - r_k) - \frac{8\eta}{d-2}C_k\right)^2, \qquad K_k = \frac{1 - a_k - C_k}{3}, \qquad r_k = a_k + B_k + C_k.$$

From Phase IIa we have at $k = N_3$:

$$a_{N_3} = o(1), \qquad C_{N_3} = 1 - \Theta(\varepsilon), \qquad B_{N_3} = \Theta(\varepsilon), \qquad r_{N_3} = 1 - \Theta(\varepsilon).$$

In particular,

$$K_{N_3} = \frac{1 - a_{N_3} - C_{N_3}}{3} = \Theta(\varepsilon) > 0.$$

**Step 1: contraction of nuisance terms.**

*Bulk component.* Write $\delta_k := 1 - C_k$. For $k \geq N_3$, we have $r_k \geq 1 - \Theta(\varepsilon)$ and $C_k = 1 - \delta_k$ with $\delta_k = O(\varepsilon)$. Expanding the $C$-update yields

$$C_{k+1} = (1 - \delta_k)\left(1 - 4\eta\delta_k + O(\eta\varepsilon)\right)^2 = 1 - \delta_k(1 - c\eta) + O(\eta\delta_k^2),$$

for some absolute $c > 0$. Since $\delta_k = O(\varepsilon)$, the quadratic term is negligible, and therefore

$$\delta_{k+1} \leq (1 - c_0\eta)\,\delta_k,$$

which proves geometric contraction of $1 - C_k$.

*Spike-orthogonal component.* Since $B_{N_3} = \Theta(\varepsilon)$ and $K_{N_3} = \Theta(\varepsilon)$, and both $a_k$ and $C_k$ increase thereafter, we have $K_k \leq K_{N_3}$ for all $k \in [N_3, N_4]$. Hence

$$K_k - B_k \leq -c\varepsilon$$

for some absolute $c > 0$, and therefore

$$B_{k+1} \leq B_k\left(1 - c_0\eta(1 + \lambda)\right),$$

proving geometric decay of $B_k$.

**Step 2: signal amplification.** For $k \in [N_3, N_4)$ we have $a_k \leq \delta$ and $r_k \leq 1 + o(1)$, hence

$$3 - 2a_k - r_k \geq 1$$

for $\delta$ small enough. Thus

$$a_{k+1} \geq a_k(1 + 4\eta)^2, \qquad a_{N_3+m} \geq a_{N_3}(1 + 4\eta)^{2m}.$$

The hitting time $N_4$ therefore satisfies

$$N_4 - N_3 \leq \frac{1}{8\eta}\log\frac{\delta}{a_{N_3}} = O(\eta^{-1}\log d).$$

**Step 3: alignment convergence.** For $k \geq N_4$, we have $a_k \geq \delta$, while $B_k = o(1)$ and $1 - C_k = o(1)$ by Step 2. Since $b_k = B_k/(1 + \lambda)$ and $(d-2)c_k^2 = C_k^2/(d-2)$,

$$\|M_k\|_F^2 = a_k^2 + \frac{B_k^2}{(1 + \lambda)^2} + \frac{C_k^2}{d - 2} = a_k^2(1 + o(1)).$$

Therefore

$$\mathrm{Align}(k) = \frac{a_k}{\|M_k\|_F} = 1 - o(1),$$

and since $a_k$ is nondecreasing while the nuisance terms decay, $\mathrm{Align}(k)$ increases monotonically to 1. $\qquad\square$

## E. Discrete-time population dynamics of SpecGD on $\mathcal{M}$

We analyze the *discrete-time population dynamics* of SpecGD restricted to the invariant manifold $\mathcal{M}$.

### E.1. Diagonalization on $\mathcal{M}$ and the exact scalar recursion

On $\mathcal{M}$, $G(M)$ is diagonal in the orthogonal decomposition $\mathrm{span}\{w_\star\} \oplus \mathrm{span}\{v\} \oplus \mathrm{range}(P_\perp)$. If $M = a\, w_\star w_\star^\top + b\, vv^\top + c\, P_\perp$ and $r = \mathrm{Tr}(MQ)$, then the eigenvalues of $G(M)$ are

$$g_{w_\star} = 8(a-1) + 4(r-1) = 4\,(r+2a-3), \qquad g_v = 8(1+\lambda)^2 b + 4(1+\lambda)(r-1) = 4(1+\lambda)\,(r+2(1+\lambda)b-1), \quad (82)$$

$$g_\perp = 8c + 4(r-1) = 4\,(r+2c-1). \qquad (83)$$

Hence the update directions are determined by the signs of the three affine expressions

$$r_k + 2a_k - 3, \qquad r_k + 2(1+\lambda)b_k - 1, \qquad r_k + 2c_k - 1,$$

exactly as in the continuous-time analysis.

**Lemma E.1.** *Assume $M_k \in \mathcal{M}$ and write $a_k = \alpha_k^2$, $b_k = \beta_k^2$, $c_k = \gamma_k^2$ with $\alpha_k, \beta_k, \gamma_k \geq 0$. Let $(g_{w_\star,k}, g_{v,k}, g_{\perp,k})$ be* (82)–(83) *evaluated at $(a_k, b_k, c_k, r_k)$, where $r_k = \mathrm{Tr}(M_k Q)$.*

*Then $M_{k+1} \in \mathcal{M}$ and the square-root variables obey the* exact *recursion*

$$\alpha_{k+1} = \big|\alpha_k - \eta\, \mathrm{Sign}(g_{w_\star,k})\big|, \qquad \beta_{k+1} = \big|\beta_k - \eta\, \mathrm{Sign}(g_{v,k})\big|, \qquad \gamma_{k+1} = \big|\gamma_k - \eta\, \mathrm{Sign}(g_{\perp,k})\big|. \qquad (84)$$

*Consequently,*

$$a_{k+1} = \big(\alpha_k - \eta\, \mathrm{Sign}(g_{w_\star,k})\big)^2, \quad b_{k+1} = \big(\beta_k - \eta\, \mathrm{Sign}(g_{v,k})\big)^2, \quad c_{k+1} = \big(\gamma_k - \eta\, \mathrm{Sign}(g_{\perp,k})\big)^2. \qquad (85)$$

*Proof.* Fix $k$ and write an SVD of $W_k$ in the orthonormal basis $U := [w_\star, v, U_\perp]$:

$$W_k = U \Sigma_k V_k^\top, \qquad \Sigma_k = \mathrm{diag}(\alpha_k, \beta_k, \gamma_k, \dots), \qquad \alpha_k, \beta_k, \gamma_k \geq 0.$$

Then $M_k = W_k W_k^\top = U \Sigma_k^2 U^\top$ has the form in $\mathcal{M}$ with coefficients $(a_k, b_k, c_k) = (\alpha_k^2, \beta_k^2, \gamma_k^2)$.

By (82)–(83), $G(M_k)$ is diagonal in the same basis $U$, hence

$$G(M_k)W_k = U\ \mathrm{diag}(g_{w_\star,k}\alpha_k,\ g_{v,k}\beta_k,\ g_{\perp,k}\gamma_k, \dots)\, V_k^\top.$$

For a diagonal matrix, the polar factor replaces each *nonzero* diagonal entry by its sign and leaves zero entries equal to $0$. Thus

$$U_k = \mathrm{polar}(G(M_k)W_k) = U\ \mathrm{diag}(\mathrm{Sign}(g_{w_\star,k}\alpha_k),\ \mathrm{Sign}(g_{v,k}\beta_k),\ \mathrm{Sign}(g_{\perp,k}\gamma_k), \dots)\, V_k^\top.$$

Substituting into $W_{k+1} = W_k - \eta U_k$ gives

$$W_{k+1} = U\ \mathrm{diag}(\alpha_k - \eta\,\mathrm{Sign}(g_{w_\star,k}\alpha_k),\ \beta_k - \eta\,\mathrm{Sign}(g_{v,k}\beta_k),\ \gamma_k - \eta\,\mathrm{Sign}(g_{\perp,k}\gamma_k), \dots)\, V_k^\top.$$

Taking singular values yields (84), and squaring yields (85). $\qquad\qquad\square$

**Corollary E.2** (Discrete invariance of $\mathcal{M}$)**.** *If $M_0 \in \mathcal{M}$, then $M_k \in \mathcal{M}$ for all $k \geq 0$.*

*Remark* E.3 (Discrete barrier overshoot)**.** The continuous-time dynamics evolve continuously across the sign changes of $r + 2a - 3$, $r + 2(1+\lambda)b - 1$, and $r + 2c - 1$. In discrete time, (84) shows that a single step can overshoot a threshold, flip the sign, and step back when $\eta$ is too large.

**Learning rate choice.** Contrary to the continuous-time analysis, the choice of the learning rate is critical. We choose $\eta$ as follows

$$\eta = \frac{\kappa}{\sqrt{d+\lambda}}, \qquad (86)$$

where $\kappa > 0$ is a small enough constant.

Note that experimentally, Figure 2 suggests that SpecGD could still work with a larger learning rate. But the analysis is beyond the scope of this work.

**E.2. Discrete phase structure: initial isotropic growth, then uniform regulation of $b_k, c_k$**

Define the first index at which the *spike-direction* sign can change:

$$N_1' := \inf\Big\{k \geq 0 : \ r_k + 2(1+\lambda)b_k \geq 1\Big\}.$$

(Compare with $T_1'$ in the continuous-time analysis.)

**Lemma E.4.** *For every $k < N_1'$ one has*

$$g_{w_\star,k} < 0, \qquad g_{v,k} < 0, \qquad g_{\perp,k} < 0,$$

*and hence (since $\alpha_k, \beta_k, \gamma_k > 0$ along this segment)*

$$\alpha_{k+1} = \alpha_k + \eta, \qquad \beta_{k+1} = \beta_k + \eta, \qquad \gamma_{k+1} = \gamma_k + \eta.$$

*In particular, for all $k \leq N_1'$,*

$$\alpha_k = \beta_k = \gamma_k = \sqrt{\mu} + k\eta, \qquad a_k = b_k = c_k = (\sqrt{\mu} + k\eta)^2.$$

*Consequently,*

$$r_k = (d+\lambda)\alpha_k^2, \qquad r_k + 2(1+\lambda)b_k = (d+3\lambda+2)\alpha_k^2, \qquad r_k + 2c_k = (d+\lambda+2)\alpha_k^2, \qquad r_k + 2a_k = (d+\lambda+2)\alpha_k^2. \tag{87}$$

*Moreover*

$$N_1' = O\left(\frac{1}{\eta\sqrt{d+\lambda}}\right) = O(1). \tag{88}$$

*Proof.* We argue by induction as in the continuous-time proof. At $k = 0$, isotropy gives $\alpha_0 = \beta_0 = \gamma_0 = \sqrt{\mu}$. Assume $\alpha_k = \beta_k = \gamma_k$ for some $k$ and that $k < N_1'$, i.e. $r_k + 2(1+\lambda)b_k < 1$. Since (87) holds, we have $(d+3\lambda+2)\alpha_k^2 < 1$. Then $(d+\lambda+2)\alpha_k^2 < 1$ and $(d+\lambda+2)\alpha_k^2 < 3$, so by (82)–(83) we have $g_{v,k} < 0, g_{\perp,k} < 0, g_{w_\star,k} < 0$. Since $\alpha_k, \beta_k, \gamma_k > 0$ along this segment, (84) yields $\alpha_{k+1} = \alpha_k + \eta, \beta_{k+1} = \beta_k + \eta, \gamma_{k+1} = \gamma_k + \eta$, which proves the first claims.

The formulas (87) follow by substituting $a = b = c = \alpha^2$ into the definition of $r$ and the three threshold expressions. Finally, (88) follows by solving $r_k + 2(1+\lambda)b_k \geq 1$. $\qquad\square$

**E.3. Uniform trapping of the nuisance coordinates $b_k$ and $c_k$**

**Lemma E.5.** *Assume $\eta = \kappa/\sqrt{d+\lambda}$ for small enough $\kappa$ and $d$ large enough. Define*

$$C_b(\kappa) := \left(\kappa + \frac{1}{\sqrt{3}}\right)^2, \qquad C_c(\kappa) := (1+\kappa)^2. \tag{89}$$

*Then for all $k \geq 0$,*

$$(1+\lambda)b_k \leq C_b(\kappa), \qquad (d-2)c_k \leq C_c(\kappa). \tag{90}$$

*Proof.* We bound $\beta_k = \sqrt{b_k}$ and $\gamma_k = \sqrt{c_k}$ using (84). Note that by (82)–(83),

$$g_{v,k} < 0 \iff r_k + 2(1+\lambda)b_k < 1, \qquad g_{\perp,k} < 0 \iff r_k + 2c_k < 1.$$

**Step 1: bound on $\beta_k$.** Define $\overline{\beta} := \frac{1}{\sqrt{3(1+\lambda)}} + \eta$. If $g_{v,k} \geq 0$, then $\mathrm{Sign}(g_{v,k}) \in \{0, +1\}$ and $\beta_{k+1} = |\beta_k - \eta\,\mathrm{Sign}(g_{v,k})| \leq \max\{\beta_k, \eta\}$. If $g_{v,k} < 0$, then $r_k + 2(1+\lambda)b_k < 1$, i.e.

$$a_k + 3(1+\lambda)b_k + (d-2)c_k < 1,$$

so $3(1+\lambda)b_k < 1$ and $\beta_k < \frac{1}{\sqrt{3(1+\lambda)}}$, hence $\beta_{k+1} = \beta_k + \eta \leq \overline{\beta}$. Since $\beta_0 = \sqrt{\mu} \ll 1/\sqrt{1+\lambda}$, we obtain $\beta_k \leq \overline{\beta}$ for all $k$.

Therefore

$$(1+\lambda)b_k = (1+\lambda)\beta_k^2 \leq (1+\lambda)\bar{\beta}^2 = \left(\frac{1}{\sqrt{3}} + \eta\sqrt{1+\lambda}\right)^2 \leq \left(\frac{1}{\sqrt{3}} + \kappa\right)^2 = C_b(\kappa),$$

using $\eta = \kappa/\sqrt{d+\lambda}$ and $\sqrt{1+\lambda} \leq \sqrt{d+\lambda}$.

**Step 2: bound on $\gamma_k$.** Define $\bar{\gamma} := \frac{1}{\sqrt{d}} + \eta$. If $g_{\perp,k} \geq 0$, then $\mathrm{Sign}(g_{\perp,k}) \in \{0, +1\}$ and $\gamma_{k+1} \leq \max\{\gamma_k, \eta\}$. If $g_{\perp,k} < 0$, then $r_k + 2c_k < 1$, i.e.

$$a_k + (1+\lambda)b_k + d\,c_k < 1,$$

so $d\,c_k < 1$ and $\gamma_k < 1/\sqrt{d}$, hence $\gamma_{k+1} = \gamma_k + \eta \leq \bar{\gamma}$. Since $\gamma_0 = \sqrt{\mu} \ll 1$, we obtain $\gamma_k \leq \bar{\gamma}$ for all $k$.

Finally, using $d/(d-2) \leq 3$ for $d \geq 3$ and $\eta\sqrt{d} \leq \kappa$,

$$(d-2)c_k = (d-2)\gamma_k^2 \leq (d-2)\bar{\gamma}^2 \leq \left(\sqrt{\tfrac{d-2}{d}} + \eta\sqrt{d-2}\right)^2 \leq (1+\kappa)^2 = C_c(\kappa).$$

This proves (90). □

### E.4. Deterministic signal growth to a constant level

**Proposition E.6.** *Assume $\eta = \kappa/\sqrt{d+\lambda}$ and let $C_b(\kappa), C_c(\kappa)$ be as in (89). Define*

$$C_{\mathrm{noise}}(\kappa) := C_b(\kappa) + C_c(\kappa), \qquad A_0(\kappa) := 1 - \frac{C_{\mathrm{noise}}(\kappa)}{3}. \tag{91}$$

*Assume $A_0(\kappa) > 0$ (ie. $\kappa$ is small enough). Let the hitting time*

$$N_2' := \inf\{k \geq 0 : a_k \geq A_0(\kappa)\}. \tag{92}$$

*Then:*

1. *For every $k < N_2'$, one has $g_{w_\star,k} < 0$ and hence*

$$\alpha_{k+1} = \alpha_k + \eta \qquad \text{for all } k < N_2'. \tag{93}$$

2. *Consequently,*

$$N_2' = O(\eta^{-1}). \tag{94}$$

3. *Moreover, for all $k \geq N_2'$,*

$$a_k \geq A(\kappa, \eta) := \left(\sqrt{A_0(\kappa)} - \eta\right)^2. \tag{95}$$

*Proof.* By Lemma E.5, for all $k$,

$$(1+\lambda)b_k \leq C_b(\kappa), \qquad (d-2)c_k \leq C_c(\kappa),$$

hence using $r_k = a_k + (1+\lambda)b_k + (d-2)c_k$,

$$r_k \leq a_k + C_{\mathrm{noise}}(\kappa).$$

Therefore

$$r_k + 2a_k \leq 3a_k + C_{\mathrm{noise}}(\kappa).$$

If $a_k < A_0(\kappa) = 1 - C_{\mathrm{noise}}(\kappa)/3$, then $r_k + 2a_k < 3$, so by (82) we have $g_{w_\star,k} = 4(r_k + 2a_k - 3) < 0$. Since $\alpha_k > 0$ for $k < N_2'$ (it is increasing by (93) from $\alpha_0 = \sqrt{\mu} > 0$), (84) yields $\alpha_{k+1} = \alpha_k + \eta$, proving item (1). Item (2) follows immediately.

For item (3), we prove the discrete barrier

$$\alpha_k \geq \sqrt{A_0(\kappa)} - \eta \qquad \text{for all } k \geq N_2'.$$

The base case holds since $\alpha_{N_2'} \geq \sqrt{A_0(\kappa)}$ by definition of $N_2'$. If $\alpha_k \geq \sqrt{A_0(\kappa)}$, then $\alpha_{k+1} \geq \alpha_k - \eta \geq \sqrt{A_0(\kappa)} - \eta$. If instead $\alpha_k \in [\sqrt{A_0(\kappa)} - \eta, \sqrt{A_0(\kappa)})$, then $a_k < A_0(\kappa)$, hence $g_{w_\star,k} < 0$ and (84) gives $\alpha_{k+1} = \alpha_k + \eta \geq \sqrt{A_0(\kappa)}$. This closes the induction, and squaring yields (95). □

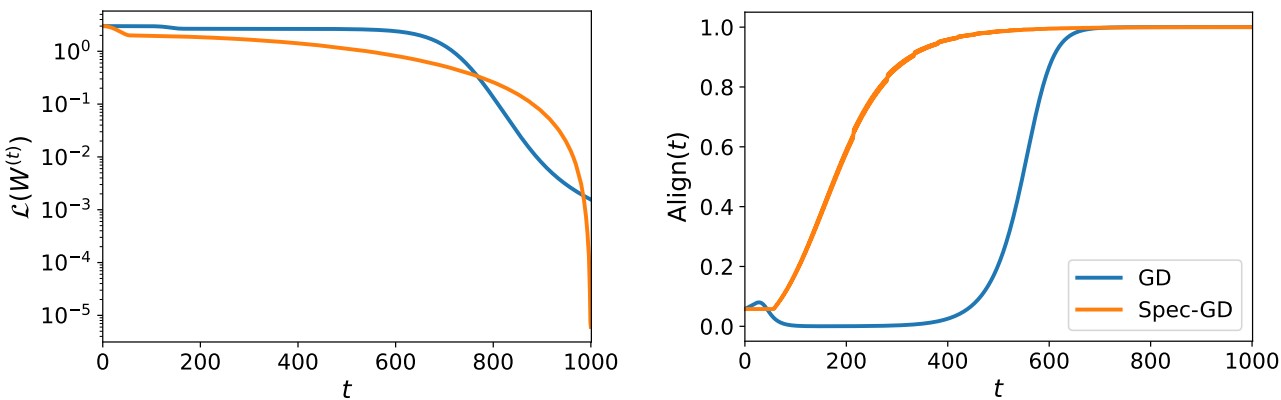

*Figure 5.* Population training dynamics. **Left:** Population loss $\mathcal{L}(W^{(t)})$. **Right:** Alignment $\mathrm{Align}(t)$.

### E.5. Alignment guarantee

**Proposition E.7.** *Under the assumptions of Theorem 4.1 we have for all $k \geq N_2'$,*

$$1 - \mathrm{Align}(k) \leq \frac{1}{2A(\kappa,\eta)^2}\left(\frac{C_b(\kappa)^2}{(1+\lambda)^2} + \frac{C_c(\kappa)^2}{d-2}\right), \tag{96}$$

*where $C_b(\kappa), C_c(\kappa)$ are defined in (89) and $A(\kappa,\eta)$ in (95).*

*Proof.* Fix $k \geq N_2'$. By Proposition E.6, $a_k \geq A(\kappa,\eta) > 0$. We have

$$\mathrm{Align}(k) = \frac{1}{\sqrt{1+X_k}}, \qquad X_k := \frac{b_k^2 + (d-2)c_k^2}{a_k^2} \geq 0.$$

Since $1 - (1+x)^{-1/2} \leq x/2$ for $x \geq 0$,

$$1 - \mathrm{Align}(k) \leq \frac{X_k}{2} = \frac{b_k^2 + (d-2)c_k^2}{2a_k^2} \leq \frac{b_k^2 + (d-2)c_k^2}{2A(\kappa,\eta)^2}.$$

Applying Lemma E.5 gives

$$b_k^2 \leq \frac{C_b(\kappa)^2}{(1+\lambda)^2}, \qquad (d-2)c_k^2 \leq \frac{C_c(\kappa)^2}{d-2},$$

which yields (96). $\qquad\qquad\square$

## F. Additional experiments

**Population dynamics.** We illustrate the population-level dynamics of GD and SpecGD on the spiked phase-retrieval model. We use dimension $d = m = 300$, spike strength $\lambda = 10$, learning rate $\eta = 10^{-3}$, and an initialization on the Stiefel manifold $M_0 = \theta^2 I_d$ with initial scaling $\theta^2 = \rho_0/(d+\lambda)$, $\rho_0 = 10^{-2}$. Figure 5 reports the evolution of the population loss $\mathcal{L}(W^{(t)})$ and the Frobenius alignment $\mathrm{Align}(t)$. Under standard GD, the alignment initially *decreases* during the early iterations, reflecting the uniform growth of all spectral components before the signal direction becomes dominant. Only after this phase does the alignment start to increase and eventually approach one. In contrast, SpecGD exhibits a much faster improvement of alignment, without a pronounced initial decay, and reaches the high-alignment regime in significantly fewer iterations. While both algorithms eventually recover an almost perfectly aligned solution, SpecGD decreases the population loss much more rapidly than GD and achieves a substantially lower value within the same training horizon.

**Influence of the learning rate.** We consider the experimental setting described in the main text. As shown in Figure 6, an overly aggressive learning rate causes the loss associated with GD to diverge, whereas the loss for SpecGD remains stable and well controlled. Figure 7 further shows that, under GD, the spike component reaches a large magnitude, after which the contraction mechanism on the nuisance components no longer operates effectively.

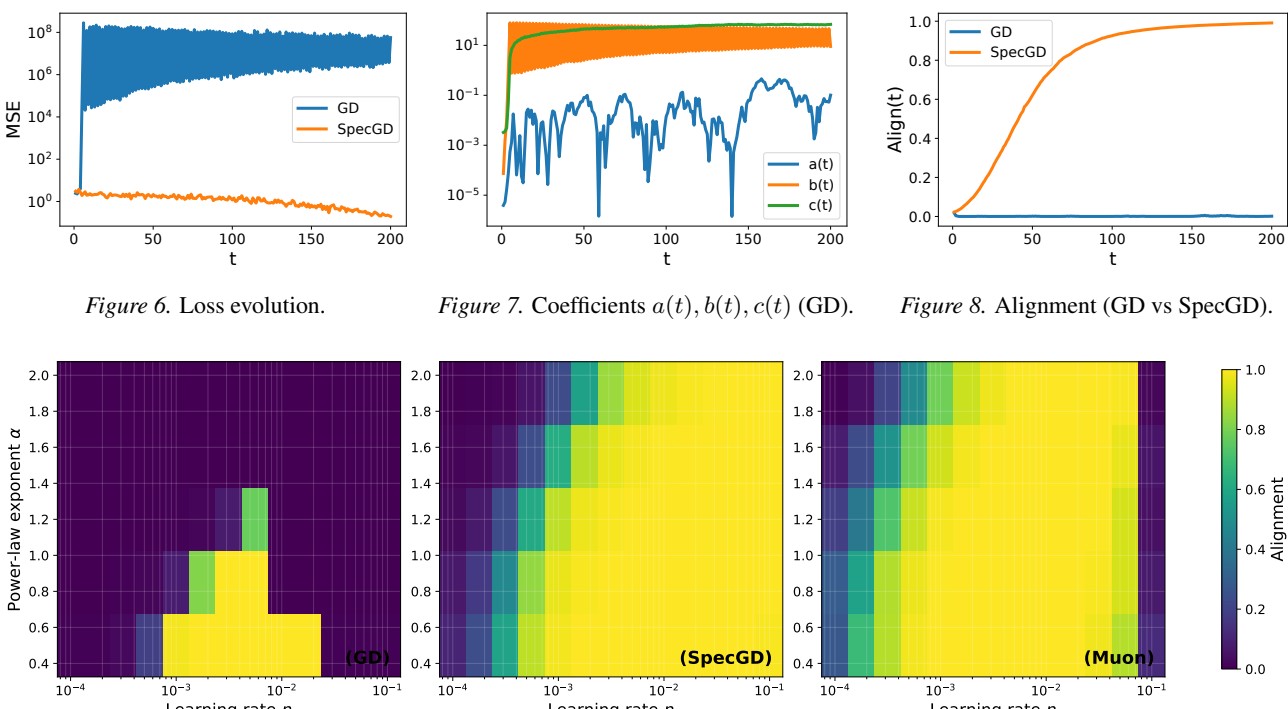

*Figure 6.* Loss evolution.    *Figure 7.* Coefficients $a(t), b(t), c(t)$ (GD).    *Figure 8.* Alignment (GD vs SpecGD).

*Figure 9.* Comparison of the performances of GD, SpecGD, and Muon across different learning rates and power-law exponents, $d = 300$.

**Power-law covariance.**    We now consider a more challenging anisotropic setting in which the input covariance matrix $Q$ has a power-law spectrum. Specifically, we take $Q = U \operatorname{diag}(\lambda_1, \ldots, \lambda_d) U^\top$ with $\lambda_i \propto i^{-\alpha}$ for $\alpha = 2$, normalized so that $\operatorname{Tr}(Q) = d$. The teacher vector $w_\star$ is chosen so that $Q^{1/2} w_\star$ aligns with one of the smallest-eigenvalue directions of $Q$, and is normalized to satisfy $\|Q^{1/2} w_\star\|_2 = 1$. We use Gaussian random initialization for $W$, dimension $d = 300$, width $m = 100$, learning rate $\eta = 10^{-3}$, batch size 500, and run the algorithms for 1000 iterations.

We additionally study the robustness of the phenomenon under power-law anisotropy by varying the exponent $\alpha$. Figure 9 reports the final performance obtained by GD, SpecGD, and Muon across different learning rates and values of $\alpha$. As anisotropy increases, GD deteriorates significantly, whereas SpecGD and Muon remain substantially more stable.

**MNIST dataset.**    To investigate whether anisotropy-driven misalignment persists beyond the phase-retrieval setting, we train a two-layer ReLU network on a binary MNIST task (digits 4 vs. 9). Each image is flattened into a standardized vector $x \in \mathbb{R}^{784}$. We introduce a controlled rank-one nuisance anisotropy by perturbing the inputs according to

$$x' = x + gu, \qquad g \sim \mathcal{N}(0, \tau^2),$$

where $u \in \mathbb{R}^{784}$ is a fixed unit vector orthogonal to the difference of class means, shared across all samples, and $\tau > 0$ controls the spike strength. The scalar coefficient $g$ is sampled independently for each example. This construction induces a rank-one perturbation of the input covariance while remaining label-independent by construction. We compare standard gradient descent (GD) with spectral gradient descent (SpecGD) and Muon in the full-batch regime, and measure test accuracy. The corresponding experimental results are shown in Figure 10. GD struggles to separate the perturbed images, whereas SpecGD and Muon remain significantly more stable and display qualitatively similar behavior, supporting the hypothesis that spectral normalization of the updates mitigates the effect of anisotropic nuisance directions.

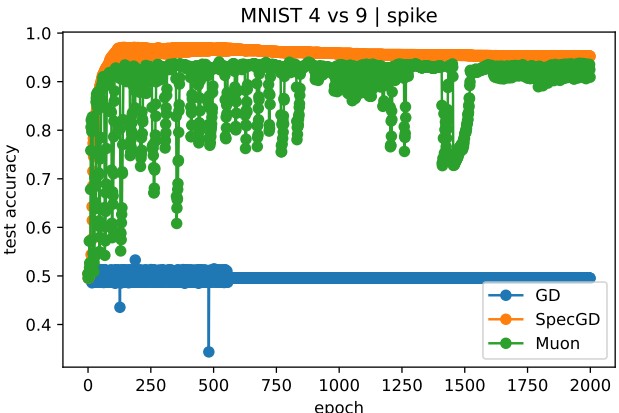

*Figure 10.* Test accuracy of the model trained by different optimizers on the MNIST spiked-covariance experiment for $\tau = 200$.

