# OpenReview forum: "Spectral Gradient Descent Mitigates Anisotropy-Driven Misalignment: A Case Study in Phase Retrieval"
_ICML.cc/2026/Conference — ICML 2026 regular_

### Official Review · Reviewer_pvuc · 2026-03-12

**Soundness:** 4
**Presentation:** 3
**Significance:** 3
**Originality:** 3
**Overall Recommendation:** 4
**Confidence:** 4

**Summary:**

This paper provides a theoretical analysis of spectral gradient descent (SpecGD) methods, exemplified by the Muon optimizer, through the lens of phase retrieval under anisotropic Gaussian inputs. The authors derive an exact reduction to a three-dimensional invariant manifold and characterize a two-stage learning dynamic: a growth stage where signal, spike, and bulk coefficients evolve simultaneously, followed by an alignment stage where the signal aligns with the true direction. The key finding is that SpecGD mitigates variance-induced misalignment by removing spike amplification effects that plague standard gradient descent under strong anisotropy.

**Compliance With Llm Reviewing Policy:**

Affirmed.

**Final Justification:**

This is a good paper. I will keep this as my final rating, good luck.

**Key Questions For Authors:**

1. Can you provide theoretical or empirical evidence that the spike suppression mechanism identified in phase retrieval extends to multi-layer neural networks with realistic data distributions?

2. Under what conditions on real-world data covariance structures would you expect SpecGD to provide significant advantages over standard adaptive optimizers like Adam?

3. In your experiments beyond the spiked model (power-law spectra), what is the quantitative relationship between the degree of anisotropy and the performance gap between SpecGD and GD?

**Limitations:**

Yes. The authors acknowledge the stylized nature of their model and discuss extensions to more general covariance structures in the experiments section.

**Strengths And Weaknesses:**

**Strengths:**
1. Rigorous theoretical analysis with exact dynamical reduction to interpretable low-dimensional manifolds
2. Clear mechanistic explanation of why SpecGD outperforms GD under anisotropic data through spike suppression
3. Well-structured presentation connecting optimization geometry to statistical learning dynamics

**Weaknesses:**
1. Highly stylized setting (phase retrieval with spiked covariance) limits direct applicability to deep learning
2. Gap between theoretical model and practical neural network training remains substantial
3. Limited practical guidance for practitioners on when to prefer SpecGD over standard optimizers

---

> ### Author Rebuttal · Authors · 2026-03-29
>
> We thank the reviewer for the positive evaluation and for raising important questions regarding the scope and practical relevance of our results.
>
> 1. **On the stylized setting and extension beyond phase retrieval,**
>
>       We agree that the phase retrieval model is stylized. Its purpose is to provide a minimal nonlinear setting where anisotropy and feature interaction can be analyzed exactly. Importantly, the mechanism we identify: a variance-induced misalignment and its suppression by spectral normalization, does not rely on the specific form of phase retrieval. What is essential is: anisotropic data covariance, and misalignment between high-variance directions and the task-relevant signal. We provide additional experimental results that show the phenomenon analyzed can also be observed when training more general two layer neural networks on MNIST data perturbed by a spike (see also the answer to reviewer Bx7Q): https://ibb.co/ycR1dxzp.
>
> 2. **When should practitioners prefer SpecGD?**
>
>     Our results suggest that SpecGD (and Muon-like methods) are particularly advantageous in regimes where: the data exhibits strong anisotropy (e.g., heavy-tailed or power-law spectra), and dominant variance directions are not aligned with the task signal. In contrast, in near-isotropic settings or when leading components are informative, the advantage is expected to be smaller.
>
> 3. **Quantitative relationship between anisotropy and performance gap.**
>
>     Under power law covariance matrix, we experimentally show that anisotropy critically affects SGD while Muon/SpecGD are less sensitive: https://ibb.co/N6qPYXvF.

---

> > ### Author Rebuttal · Reviewer_pvuc · 2026-04-02
> >
> > Thank you for the rebuttal. The response helps clarify the intended scope of the paper and gives a more concrete qualitative explanation of when the identified mechanism may matter in practice, especially in strongly anisotropic settings where dominant variance directions are misaligned with the task-relevant signal. I also appreciate the discussion connecting the single-spike analysis to broader anisotropic regimes.
> >
> > My concerns are nevertheless only partially resolved. The main reason is that the paper remains centered on a stylized phase-retrieval setting, and the written rebuttal still provides limited self-contained evidence that the spike-suppression mechanism extends to realistic multi-layer neural network training beyond this model. In addition, the guidance on when practitioners should prefer SpecGD over standard optimizers is helpful but remains mostly qualitative, and the quantitative relationship between anisotropy strength and the performance gap is still not fully characterized.
> >
> > Overall, I continue to view the paper as technically strong and the theoretical analysis as interesting. However, I also continue to see the main limitation as the gap between the clean theoretical results and more realistic neural-network settings, so my overall assessment is broadly unchanged.
> >
> > As a follow-up, could the author clarify what quantitative trend they expect between anisotropy strength and the GD-vs-SpecGD performance gap in the broader non-spiked settings discussed in the rebuttal?

---

> > > ### Author Response · Authors · 2026-04-04
> > >
> > > Thank you for your additional feedback. A precise quantitative characterization of how the power-law exponent an affects the performance gap between GD and SpecGD would require a dedicated analysis and is outside the scope of this work. That said, our theoretical results in the spiked model suggest a clear mechanism that extends qualitatively to the power-law setting. When the signal is aligned with the tail of the covariance spectrum, and the power exponent $\alpha$ increases, the performance gap between GD and SpecGD is expected to increase. GD would first learn the easy but uninformative spectrum directions, while SpecGD will learn faster the signal direction since it is less biased toward reducing the variance and learns directions more evenly. This is consistent with Figure 8 and the additional heat map experiments.

---

### Official Review · Reviewer_ydvD · 2026-03-12

**Soundness:** 4
**Presentation:** 4
**Significance:** 3
**Originality:** 3
**Overall Recommendation:** 5
**Confidence:** 3

**Summary:**

The authors study the gradient descent dynamics on the population loss of a wide quadratically activated two-layer network learning data where the labels generated by a generalized linear model with quadratic activation with additive gaussian noise. The input comes from a gaussian distribution with a spiked covariance, where the spike in the data is orthogonal to the vector used to generate the labels. The gradient descent dynamics is first relaxed to gradient flow, which gives concrete measures on how well the data and the label spikes are learned. Then, these dynamics are discretized again. The dynamics depends on two phases: first the (uninformative) data spike is learned, then this spike gets unlearned and the informative label spike is recovered. This algorithm is compared with a different algorithmic choice, which the authors call spectral GD, where the gradient gets filtered at each step to become independent of its spectrum. This other choice appears to perform better than GD at recovering the informative task.

**Compliance With Llm Reviewing Policy:**

Affirmed.

**Final Justification:**

The authors addressed my concerns. I am happy to keep my already positive score.

**Key Questions For Authors:**

1. In section 5.4 you mention that to obtain the updates for discrete step updates one cannot just discretize the flow. Can you give an intuitive explanation of it? It would be nice to give some intuition on it directly in that section.
2. Is there an intuitive argument for $T_1$ and $T_2$ in the GD case to scale exactly as the spectral algorithm times $\log(d)$?
3. In theorem 2 statement (ii): do you mean to also say that $k<T_2$?
4. In section 2 the labels are generated with additive gaussian noise, which is then essentially ignored (and in fact in appendix D there is the additional assumption $\sigma = 0$. Does it mean additive noise plays no role?
5. You claim that spectral GD can be considered a proxy for Muon. Would you see the same phenomenology if you were to do some numerical experiments with it in the spirit of section 6?

**Limitations:**

Yes

**Strengths And Weaknesses:**

The paper is very clearly written and easy to follow. Although the data model considered is quite simple, I think this type of simplified setting is useful for isolating the mechanisms through which different optimization algorithms behave. In this case, the model provides a clear illustration of how the performance of gradient descent can be affected by anisotropy in the data. The choice of architecture is also appropriate for this type of theoretical study: two-layer networks with quadratic activations have been widely studied in recent theoretical work and provide a tractable setting where meaningful analytical results can be obtained.

From a technical standpoint, the work appears sound. The assumptions are clearly stated and seem reasonable given the goal of studying optimization dynamics in a controlled synthetic model. The theoretical arguments are well structured and the proofs appear correct, with the main claims supported by the analysis presented in the paper. The experimental section is consistent with the theory: the experiments reproduce the predicted behavior in the synthetic setting and therefore serve as a useful validation of the analytical results. That said, it could strengthen the paper to include additional experiments on practical algorithms that inspired spectral GD (for example Muon), which would help clarify how the insights obtained in the simplified model translate to more realistic optimization settings.

In terms of originality, the analysis of SpecGD in this framework appears to be new. While the overall modeling choices are now becoming increasingly common in theoretical studies, the paper provides a new phenomenological perspectives. The contribution is therefore more about the implications of the method rather giving new theoretical analysis, but the insights are nonetheless interesting.

---

> ### Author Rebuttal · Authors · 2026-03-29
>
> We thanks the reviewer for his positive assessment on our work and useful comments.
>
> 1. One of the key issue is the barrier argument used in continuous time. Continuity is essential to show that the flow cannot go beyond some barrier. But in discrete time, the gradient jump can go above the barrier and a more intricate argument is required. This is also the key step giving the restrictions on the learning rate.
> 2. At initialization the alignment is of order $\sqrt d^{-1}$. Under GD update we obtain a multiplicative exponential grow, so $O(log d)$ time is required to reach $O(1)$ alignment. On the other hand SpecGD leads to an additive quadratic grow and only $O(1)$ time is required.
> 3. For $k≤T_2$ it is difficult to obtain a precise characterization of the growth of $a, b$ and $c$ due to the coupling effect of the neural mass $r$. This is why we can only give a close form expression of these parameters during Phase I. However, we can still qualitatively show alignment and quantify the required time.
> 4. We analyzed population gradient and adding additive noise would lead to the same dynamics in this setting.
> 5. Yes, we expect the same qualitative phenomenology to hold for Muon. We performed additional experiments that support this claim a) on general power law covariance with various anisotropic exponent $\alpha$ and different learning rate: 　https://ibb.co/N6qPYXvF and b) MNIST dataset perturbed by a spike: https://ibb.co/ycR1dxzp.

---

> > ### Author Rebuttal · Reviewer_ydvD · 2026-04-03
> >
> > The authors have addressed my concerns. I am happy to keep my positive score.

---

### Official Review · Reviewer_Bx7Q · 2026-03-12

**Soundness:** 3
**Presentation:** 3
**Significance:** 2
**Originality:** 3
**Overall Recommendation:** 4
**Confidence:** 4

**Summary:**

This paper studies why spectral gradient updates (SpecGD), viewed as an idealized proxy for the Muon optimizer, may perform better than standard gradient descent in a particular setting of the nonlinear phase retrieval problem where the input follows a spiked Gaussian distribution. The main contribution is theoretical: the authors show that gradient descent suffers from an anisotropy-driven misalignment effect at the early stage of optimization, while SpecGD does not. The paper also provides numerical experiments suggesting that this qualitative phenomenon extends to power law covariance matrices.

**Compliance With Llm Reviewing Policy:**

Affirmed.

**Final Justification:**

The authors have adequately addressed my concerns, and I will maintain my positive score.

**Key Questions For Authors:**

1. As mentioned in the paper, recent work has already studied SpecGD as a canonical abstraction of Muon in linear, bilinear, and deep-linear settings, with the general conclusion that SpecGD learns spectral components more evenly than GD. Could the authors clarify more explicitly what conceptual insight is uniquely enabled by the present nonlinear phase-retrieval setting? In particular, how does the nonlinear structure change the understanding beyond what was already known in the linear or deep-linear cases?

2. The motivation for studying phase retrieval as a proxy for neural network training is somewhat weak. Although the authors argue that the phase retrieval problem can be interpreted as training a two-layer quadratic network with fixed second-layer weights, this connection is used mainly as motivation. It would strengthen the paper if the authors could include an experiment with an actual neural network on a real dataset to demonstrate whether the same anisotropy-driven misalignment mechanism persists in more realistic training settings.

**Limitations:**

Yes

**Strengths And Weaknesses:**

Strengths:
The paper identifies a specific failure mode of gradient descent under anisotropy in the nonlinear phase retrieval setting and develops a clean analysis explaining why SpecGD can perform better in this regime. The mathematical analysis provides useful theoretical insight that helps improve understanding of the problem.

Weaknesses:
The analysis is developed within the phase retrieval setting and may be somewhat limited to this specific setup. The behavioral connection to deep networks is not fully explored. In this phase retrieval formulation, the objective is quadratic in M=WW^T
 and therefore fourth order in W. Since W is full rank, global convergence is perhaps not entirely surprising in this setting. The experiments are supportive but somewhat limited relative to the breadth of the motivation.

The paper frames the model as equivalent to training a two-layer quadratic network with fixed second-layer weights; however, this viewpoint serves mainly as motivation rather than as a fully developed empirical connection.

---

> ### Author Rebuttal · Authors · 2026-03-29
>
> We thank the reviewer for the thoughtful feedback.
>
> 1. **What is new beyond linear / deep-linear SpecGD analyses?**
>
>      The fact that SpecGD learns spectral component more evenly is already present in the analysis of Ma et al. (2026). But their comparison with other algorithms such as GD involve different choices for the learning rate. Our comparison is more direct since we use the same fixed learning rate for both GD and SpecGD. However, our contribution goes beyond this by identifying a qualitatively new phenomenon: variance-induced misalignment. Early GD dynamics can be dominated by high-variance directions that are completely uninformative for the task, leading to delayed (and sometimes degraded) alignment with the signal. This mechanism is specific to anisotropic settings that hasn’t been studied previously to the best of our knowledge. The non-linearity is less important than anisotropy, but provides a natural setting where convex optimization methods cannot be used and gradient based method are natural. Additionally, the nonlinear phase-retrieval structure introduces a key feature absent in prior work:
>      a) the dynamics are coupled through the network mass $r(t)$;
>      b) different learning phases (growth and alignment) driven by different mechanisms;
>      c) competition between signal and noise directions.
>
> 2. **On global convergence being “unsurprising”.**
>
>      We would like to clarify that our focus is not on global convergence per se (while non-convex, the loss landscape is relatively benign, only one saddle point), but on the finite-time dynamics and representation trajectory. Our contribution is not that convergence happens, but how and when meaningful alignment occurs for GD and SpecGD. In particular we show that the two algorithms while converging (for a sufficiently small learning rate), have different behaviors.
>
> 3. **On connection to neural networks.**
>
>     Quadratic neural networks offer a tractable model to analyze the dynamics of neural networks. Our model encompasses several key property, such as non-linearity, non-convex learning dynamic with complex coupling between direction and feature learning. To show that the insight provided beyond this model, we performed a new experiment on MNIST dataset with a standard two layer neural network with ReLU activation function. The results show that training this model with SGD can be misled by uninformative spike, while SpecGD and Muon remain robust to this perturbation. The result can be visualized here: https://ibb.co/ycR1dxzp.

---

> > ### Author Rebuttal · Reviewer_Bx7Q · 2026-04-01
> >
> > Thank you for the clarifications.

---

### Official Review · Reviewer_btAn · 2026-03-13

**Soundness:** 3
**Presentation:** 3
**Significance:** 3
**Originality:** 3
**Overall Recommendation:** 5
**Confidence:** 3

**Summary:**

The paper provides a theoretical analysis of spectral gradient descent in the context of non-convex optimization. In specific, it uses a phase retrieval problem as a toy model to investigate how data anisotropy (modeled as spiked covariance) results in misalignment in standard euclidiean-type gradient descent, such as Adam.
SpecGD using its sign-based updates in an adaptive basis supresses the noise amplification while GD suffers from variance-induced misalignment, where the spike direction is amplified during early stage of training and the signal recovery is delayed.
The paper derive a reduced ODE dynamics to track the weight alignment with respect to the true weight. It proves while the euclidean GDs are easily captured by dominant noise directions, specGD can handle it well, ensuring learning rate invariance and robust convergence.

**Compliance With Llm Reviewing Policy:**

Affirmed.

**Final Justification:**

The authors argues that while their model is stylized, it rigorously isolates a phenomenon that persists in more complex and practical settings. the new experiments and clarifications in dimensional scaling strengthen the paper. I believe the insights regarding the SpecGD's learning rate invariance against misalignment is a clear contribution to the field. I am favoring of accepting the paper.

**Key Questions For Authors:**

1. I am curious if how multiple spikes of varying strengths would change the learning dynamics of SpecGD.
2. More practical setups: can you explain any potential gaps/issues arised on the results and explanations based on the current setup d is very large like 10000 in S^d-1 and also in complex-valued domain (maybe one S for real, the other for imaginary)?
3. since Muon was the motivation, can the authors explain what theoretical results they expect to change versus remain qualitatively intact when the practical features of Muon implementation (Newton-Schulz iteration and momentum) are introduced?

**Limitations:**

no potential negative societal impact.

**Strengths And Weaknesses:**

+ geometric insight in anisotropy: the paper demonstrates the failure of GD in phase retrieval is not just about the local optima, but also about the geometric bias toward noise direction.

+ learning rate invariance results (fig. 2) would be a good insight for the use of GD-based optimization in phase retrieval practically, where different models associated with different physical systems can vary learning rate.

+ reduction to 3-dimensional invariant manifold is a nice interpretable dynamics analyis.

As a researcher working in the intersection between ML and physics, I see the weakness of papers as follows:
- limited assumption on spike: the input distribution assumes a simple dominant spike unlike real-world scenario, where multiple, complex spectrum of singular values can exist, deteriorating the optimization performance.

- small dimensions d=300. scaling behavior would be insightful.

- limited baselines. perhaps different adaptive methods can be also explored in a more practical view point.

---

> ### Author Rebuttal · Authors · 2026-03-29
>
> We thank the reviewer for his positive assessment and insightful questions.
>
> We agree that the analyzed model is stylized. Its purpose is to isolate and rigorously characterize a specific mechanism: under anisotropic data, gradient descent can be misled by high-variance but uninformative directions, whereas SpecGD remains largely insensitive to such variance distortions. Importantly, this mechanism does not rely on the presence of a single spike, but rather on the existence of anisotropic directions misaligned with the signal.
>
> 1) We expect the same qualitative behavior to persist with multiple spikes of varying strengths. In the case of orthogonal spikes, the analysis should extend to a higher-dimensional dynamical system (one coordinate per spike), where GD sequentially amplifies dominant nuisance directions, while SpecGD remains largely insensitive to their magnitudes.
> When spikes are correlated, the dynamics become more intricate due to interactions between eigenspaces, and extending the current analysis would require new techniques. We view this as a natural and important direction for future work.
> Note that our power-law experiments, which effectively correspond to a multi-spike spectrum, already provide empirical evidence that the phenomenon persists beyond the single-spike setting.
>
> 2) The separation between GD and SpecGD becomes more pronounced as the dimension increases. In particular, the discrete analysis shows that: GD requires a time of order $log(d)$ to escape the spike-dominated regime, whereas SpecGD enters the alignment phase in $O(1)$ time.
>
> 3) We performed additional experiments on a) the power-law covariance model with the target signal lying in the space spanned by the eigenvectors of Q associated with the smallest eigenvalues and b) MNIST data perturbed by a spike (adding a few other spikes doesn’t change the qualitative result). Instead of a quadratic net, we trained a two-layer neural network. Our experiments confirm that Muon has a similar behavior as SpecGD, just a little more unstable due to the polar decomposition approximation. The main gain is on the computational time required to compute the gradient. The corresponding plots of the additional experiments can be visualized here: https://ibb.co/N6qPYXvF and https://ibb.co/ycR1dxzp.

---

> > ### Author Rebuttal · Reviewer_btAn · 2026-04-05
> >
> > The authors have addressed my concerns.

---

### Decision · Program_Chairs · 2026-04-30

**Decision:**

Accept (regular)

**Comment:**

The paper studies the spectral GD on a stylized overparameterized phase retrieval setting under anisotropic data covariance. The key take away is that spectral GD allows faster convergence where the signal, spike and bulk components progress at similar rates, while GD is considerably slowed down by the anisotropic spike. The observation is consistent and complementary with several existing case studies of spectral GD, and reviewers generally find the paper well-written and contribute to the understanding of spectral GD. The main weakness of the paper is on the simplified setting, and the limitation in the choice of learning rates (especially for spectral GD), and that the spike direction is orthogonal to the signal (which cannot be satisfied in practice, when the same model is used to sense different ground truth w_*).